# Ultrastrong MXene composite fibers through static-dynamic densification for wireless electronic textiles

Tianzhu Zhou[1,2,3], Jia Yan [4], Can Cao [5], Qiang He [1], Wulong Li [1], Long Chen [1], Chao Wu [6], Yuqi Feng [7], Denvid Lau [7], Qunfeng Cheng[2,3,4,8] ✉ & Lei Wei [1] ✉

Inherent transverse wrinkles and resulting voids between MXene ($Ti_3C_2T_x$) nanosheets hinder the preservation of their intrinsic mechanical and electrical properties in macroscopic fibers. Here, we demonstrate a controllable and continuous method for kilometer-scale fabrication of ultrastrong MXene composite fibers by utilizing static filling with short carbon nanotubes combined with dynamic thermal drawing using polylactic acid to bridge MXene nanosheets through hydrogen bonds. The resulting composite fibers achieve a record tensile strength of ~941.5 MPa and an electrical conductivity of ~3899.0 S cm$^{-1}$, with an even higher conductivity of ~12,836.4 S cm$^{-1}$ for the inner MXene fiber. This static-dynamic densification strategy significantly reduces voids with a low porosity of ~4.2% and enhances the nanosheet orientation factor to ~0.945. The embroidered smart textiles enable long-range, battery-free wireless health monitoring, body-coupled remote drone operation, and assisted communication with sustained mechanical durability. This versatile strategy offers a general pathway to fabricate high-performance functional fibers.

Fiber-based smart electronic textiles are essential in various daily wearable applications, including health monitoring and management, smart displays, artificial muscles, and artificial intelligence integration[1-8]. Intensive research has focused on conductive polymer-based, carbon-based, and metal-based materials, driven by the pursuit of fibers that offer superior mechanical and electrical properties, as well as high flexibility and ease of deformation[9,10]. Two-dimensional (2D) MXene ($Ti_3C_2T_x$) nanosheets emerge as a promising class of 2D materials for developing high-performance fibers, owing to their high

tensile strength of ~17.3 GPa and electrical conductivity (~$2.4 \times 10^4$ S cm$^{-1}$)[11-13]. Various methods have been employed to fabricate MXene fibers from these nanosheets, including wet and dry spinning, coating, electrospinning, biscrolling technique, and thermal drawing[14-17]. However, the intrinsic electrical and mechanical properties of MXene nanosheets have not yet been fully achieved in their macroscopic fibrous forms, often compromised by the presence of voids and wrinkles due to weak interfacial interactions between the nanosheets during the assembly process.

[1]School of Electrical and Electronic Engineering, Nanyang Technological University, Singapore, Singapore. [2]State Key Laboratory of Bioinspired Interfacial Materials Science, School of Nano Science and Technology, Suzhou Institute for Advanced Research, University of Science and Technology of China, Suzhou, Jiangsu, China. [3]School of Chemistry and Materials Science, University of Science and Technology of China, Hefei, Anhui, China. [4]School of Chemistry, Key Laboratory of Bio-inspired Smart Interfacial Science and Technology of the Ministry of Education, Beihang University, Beijing, China. [5]School of Materials Science and Engineering, Nanyang Technological University, Singapore, Singapore. [6]UKCRIC Advanced Infrastructure Materials Laboratory, Department of Civil and Environmental Engineering, Imperial College London, London, UK. [7]Department of Architecture and Civil Engineering, City University of Hong Kong, Hong Kong, China. [8]Institute of Energy Materials Science (IEMS), University of Shanghai for Science and Technology, Shanghai, China. ✉e-mail: chengqf@ustc.edu.cn; wei.lei@ntu.edu.sg

Increasing efforts have been devoted to enhancing the performance of MXene fibers by promoting the axial alignment of MXene nanosheets, primarily using wet spinning[18]. Most studies frequently utilize cross-linkers, combined with the pre-stretching step, to effectively align the MXene nanosheets and eliminate wrinkles, thus improving the overall properties of fibers[19]. Meanwhile, although MXene fibers have been fabricated using various coagulation processes with different coagulation baths[14,18], these strategies also primarily aim to promote the axial alignment of MXene nanosheets during wet spinning. However, there remains a noticeable gap between the intrinsic mechanical and electrical properties of MXene nanosheets and the realized properties of macroscopic fibers, an area ripe for further exploration[20]. This discrepancy often stems from the loose assembly of MXene nanosheets in the transverse direction, leading to the formation of transverse wrinkles due to the weak interfacial interactions and the effect of capillary contraction[21,22], similar to those observed with graphene nanosheets[23]. The occurrence of transverse wrinkles during the wet spinning process can lead to increased voids within the fibers by the wet spinning, diminishing their density and degrading both mechanical and electrical properties. Therefore, developing a controlled process for fabricating high-performance MXene fibers is essential to fully realize the potential of MXene nanosheets. Additionally, smart textiles made from these fibers must offer robust mechanical and electrical performance to effectively be integrated with our bodies[24]. These textiles are repeatedly subjected to physical stresses, such as moving and deformations, particularly under harsh conditions or when in contact with human skin[25]. They are also susceptible to damage from regular cleaning and maintenance, such as washing and drying processes, which can lead to performance degradation and limit their long-term usability[3,26]. Therefore, an effective solution could be the development of fibers with an encapsulation layer, which would further enhance the mechanical properties of the fibers, thereby enhancing the durability of smart textiles.

Here, we demonstrated a controllable strategy using short one-dimensional (1D) carboxylated multiwalled carbon nanotubes (CNTs) to statically fill the voids resulting from transverse wrinkles, bridging MXene nanosheets by hydrogen bonds during the wet spinning. Subsequently, the thermal drawing was employed to further dynamically reduce the voids through drawing stresses, enabling the continuous and controlled fabrication of ultrastrong MXene-CNTs-polylactic acid (PLA) (designated as MCP) composite fibers with an in situ generated encapsulation layer of PLA. Thanks to the static-dynamic densification strategy, the porosity of the MCP fiber was significantly reduced to ~4.2%, while the orientation factor of the MXene nanosheets was effectively enhanced to ~0.945. With the enhancement of interfacial interactions within fibers due to the formation of hydrogen bonds between CNTs and MXene nanosheets, as well as between PLA and MXene nanosheets, the densified fiber achieved an ultrahigh tensile strength of ~941.5 MPa and toughness of ~147.9 MJ m$^{-3}$, accompanied by an electrical conductivity of ~3899.0 S cm$^{-1}$ with an even higher conductivity of ~12,836.4 S cm$^{-1}$ for the inner MXene fiber. The smart textiles crafted from the resulting ultrastrong MCP fibers showed exceptional electromagnetic performance and mechanical durability. A long-range, battery-free wireless human health monitoring system was developed for the health monitoring of temperature, pulse pressure, and UV intensity, and featured data storage to further evaluate daily activities. Meanwhile, the smart wireless textile, utilizing body-coupled ultrastrong MXene composite fibers, achieves remote drone control and facilitates assisted communication. The synergistic combination of the static filling method and dynamic thermal drawing, along with interfacial interactions, offers more opportunities to achieve high-performance fibers using various nanostructured functional materials.

## Results

### Fabrication of ultrastrong MXene composite fibers

Etching transformed the initial Ti$_3$AlC$_2$ into accordion-like Ti$_3$C$_2$T$_x$, resulting in ~11.1 μm lateral size and ~1.5 nm thick MXene nanosheets after vibration exfoliation, confirmed by the scanning electron microscope (SEM) and atomic force microscope (AFM) images (Supplementary Figs. 1 and 2). These nanosheets, showing defect-free hexagonal crystallinity, were further verified by high-resolution transmission electron microscopy (HR-TEM) images and selected area electron diffraction patterns (Supplementary Fig. 3), along with the X-ray diffraction (XRD) patterns with the (00l) diffractions (Supplementary Fig. 4)[14,27]. Due to weak interfacial interactions between MXene nanosheets in the transverse direction, significant transverse wrinkles occur, leading to increased voids in the resulting macroscopic pure MXene fibers (designated as MX) with high porosity, as illustrated in Supplementary Fig. 2. Unlike 2D nanosheet structures, the 1D nanofiber structure can effectively prevent the formation of transverse wrinkles under shear stress during wet spinning[23,28]. CNTs, as 1D nanofibers, are considered as ideal building blocks for high-performance macroscopic fibers, owing to their exceptional intrinsic strength and high electrical conductivity[28]. Thus, we developed a method using short carboxylated CNTs with a diameter of 5.6 ± 2.3 nm (mean ± standard deviation, unless otherwise stated below) and an optimized length of ~0.46 μm (Supplementary Figs. 5 and 6) to bridge MXene nanosheets through hydrogen bonds, statically filling the voids resulting from transverse wrinkles during the wet spinning, as shown in Fig. 1a and Supplementary Movie 1. This method effectively reduced the porosity and simultaneously enhanced the axial alignment of MXene nanosheets in the resulting MXene-CNTs (designated as MC) composite fibers. Then, leveraging the dynamic thermal drawing induced stresses to further minimize voids and promote the alignment, ultrastrong MCP fibers were fabricated with low porosity and high orientation factor (Supplementary Movie 2). Meanwhile, this process generated an in situ PLA encapsulation layer, bridging MXene nanosheets through hydrogen bonds.

First, AFM images of MXene nanosheets were used to examine the formation of transverse wrinkles, as shown in Supplementary Fig. 2. These samples were prepared using high-speed coating spray at 500 rpm, simulating high shear stress in wet spinning. Although the samples of MXene nanosheets were obtained using high spray speed, the transverse wrinkles still occurred with several layers. Clearly, the measured height of the transverse wrinkles in MXene nanosheets increased from ~4.2 nm to ~33.6 nm as the number of layers in the wrinkles increased from 2 to 5 layers. This increase, with a difference from ~1.2 to ~26.1 nm, was substantially higher than that in the calculated thickness of ~3.0 to ~7.5 nm based on the height of a single nanosheet being 1.5 nm. The results confirmed that increased voids were generated when the transverse wrinkles occurred. Consequently, when the macroscopic MX fibers were fabricated from pure MXene nanosheets using wet spinning, the presence of the transverse wrinkles because of weak interfacial interactions led to more voids (Supplementary Fig. 2), further confirmed by SEM and HR-TEM images of the cross-sections. Additionally, finite element analysis (FEA) was conducted to further elucidate how transverse wrinkles lead to the formation of voids. Based on the rheological properties of the pure MXene spinning dispersion with a concentration of 40 mg mL$^{-1}$, the viscosity-shear rate curve demonstrated the shear-thinning behavior, indicating that the solution was non-Newtonian (Supplementary Fig. 7). As the shear rate ($\dot{\gamma}$) increased, the randomly distributed MXene nanosheets in spinning dispersion progressively aligned more closely with the axial direction of the shear stress. Then, this non-Newtonian solution was fitted by the power-law model $\eta = k\dot{\gamma}^{n-1}$ for the FEA simulation during the wet spinning[29], where the flow power-law index ($n$) is 0.01. According to the shear stress calculated from the FEA simulation, the shear stress in the axial center of the spinning tube ($Y$-

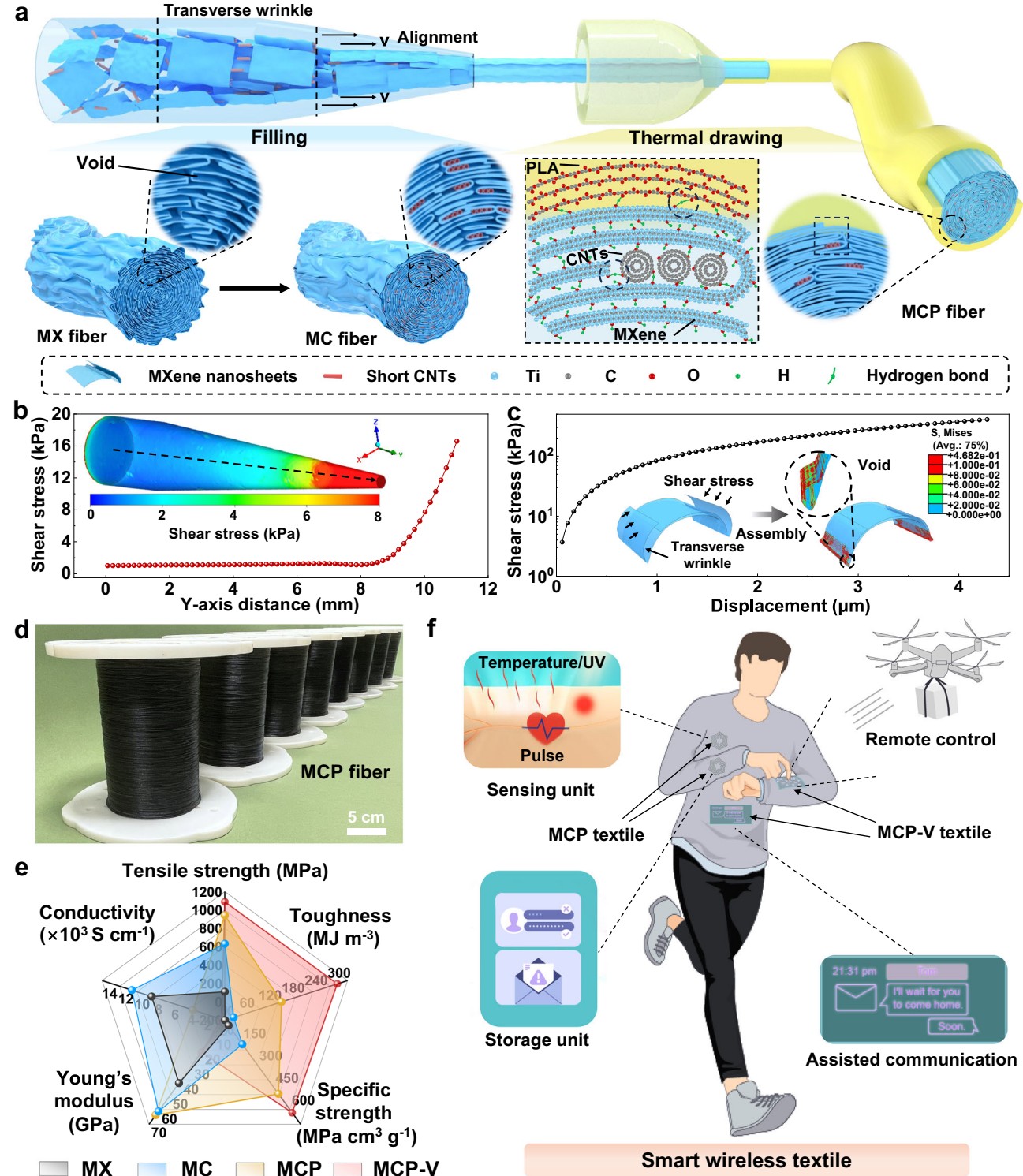

**Fig. 1 | Fabrication flow to achieve ultrastrong MXene composite fibers.**
**a** Illustration of the fabrication of MXene-carbon nanotubes (CNTs)-polylactic acid (PLA) (MCP) fiber by the combination of static filling method and dynamic thermal drawing with an in situ generated encapsulation layer through the hydrogen bonds between CNTs and MXene nanosheets, and PLA and MXene nanosheets, while the pure MXene fiber (MX) with lots of voids generated from transverse wrinkles during the wet spinning. **b** The shear stress distribution in the axial center of the spinning tube along the Y-axis, according to finite element analysis (FEA) simulation. **c** The required stress was to compact the transverse wrinkles nanosheets sufficiently to reduce voids, obtained from the dynamic FEA simulation. **d** Photograph of the kilometer-scale MCP fiber, extending to a length of thousands of meters. **e** Star plot of tensile strength, toughness, specific strength, Young's modulus, and electrical conductivity of MXene composite fibers, including MX, MXene-CNTs (MC), MCP, and MCP-V. **f** Illustration of the smart wireless textiles incorporating various spiral inductors textile based on MCP fiber, featuring sensing, and storage units, along with MCP-vinyl silicone-acetoxy silicone resin (VSASR)-ZnS-Cu$^{2+}$ (MCP-V) textiles for remote control of the drone and assisted communication.

axis) reached up to ~16.7 kPa, while the shear stress increased sharply to ~80 kPa along the $Z$-axis from the center to the edges (Fig. 1b and Supplementary Fig. 8). However, the formation of transverse wrinkles in MXene nanosheets during the wet spinning requires over 408 kPa of the stress, as indicated by dynamic FEA calculations, to compact the nanosheets sufficiently and reduce voids (Fig. 1c). Thus, the shear stress generated during the wet spinning assembly was insufficient to significantly reduce the voids caused by transverse wrinkles in the MX fiber, resulting in high porosity.

Here, the MXene-CNTs spinning dispersion with a high concentration of 40 mg mL$^{-1}$ (Supplementary Fig. 9) was chosen for the wet spinning due to a high axial alignment of nanosheets with a low n in wet spinning under shear stress (Supplementary Fig. 7). Then, using the proposed static filling method combined with the dynamic thermal drawing, ultrastrong dense MCP fibers with the length of thousands of meters were successfully fabricated, as shown in Fig. 1d. Notably, the achieved MCP fiber outperformed both MX and MC fibers, exhibiting a combination of high tensile strength of ~941.5 MPa, toughness of ~147.9 MJ m$^{-3}$, electrical conductivity of ~3899.0 S cm$^{-1}$, with an even higher electrical conductivity of ~12,836.4 S cm$^{-1}$ of the inner MC fiber, specific strength of ~477.9 MPa cm$^3$ g$^{-1}$, and high Young's modulus of ~61.4 GPa (Fig. 1e). Moreover, thanks to the high tensile strength and conductivity of MCP fibers, a long-range, battery-free wireless human health monitoring system was successfully developed using these spiral inductor patterns embroidered by MXene composite fibers, comprising a wireless sensing unit and a data storage unit, as shown in Fig. 1f. Simultaneously, it effectively enabled the monitoring of the human health by tracking temperature, pulse pressure, and UV intensity, collecting data and transferring it to the data storage unit on the wrist for further analysis and evaluation. When the MCP fiber was further coated with an outer layer of vinyl silicone-acetoxy silicone resin (VSASR) and ZnS-Cu$^{2+}$, the resulting MCP-VSASR-ZnS-Cu$^{2+}$ (designated as MCP-V) fiber demonstrated the highest tensile strength of ~1088.5 MPa, toughness of ~293.5 MJ m$^{-3}$, and electrical conductivity of ~395.5 S cm$^{-1}$, with an even higher electrical conductivity of 13,567.4 S cm$^{-1}$ of the inner MC fiber (Fig. 1e). Moreover, the schematic showed that this advanced smart wireless textile based on the MCP-V fiber achieved remote drone control and supported assisted communication (Fig. 1f).

## Interfacial interactions and morphological characterization of MXene composite fibers

Systematic characterizations on the resulting MXene composite fibers were performed to understand the interfacial interactions. Fourier transform infrared spectroscopy (FTIR) spectra revealed a shift in the wavenumber for the -OH group of MXene nanosheets from ~3472.1 cm$^{-1}$ in MX fiber to ~3464.4 cm$^{-1}$ in MC fiber (Supplementary Fig. 10), indicating the formation of hydrogen bonds between CNTs and MXene nanosheets[19]. Additionally, the wavenumber for the -OH group in MCP fiber sharply shifted down to ~3453.2 cm$^{-1}$ compared to that of MC, accompanied by a shift of ~4.7 cm$^{-1}$ in the carbonyl functional group (-C = O) of PLA from ~1744.2 to 1739.5 cm$^{-1}$, proving the hydrogen bonds between PLA and MXene nanosheets. Furthermore, X-ray photoelectron spectroscopy (XPS) spectra indicated that upon CNTs were filled into the voids among MXene nanosheets, the binding energy of Ti-O 2$p_{2/3}$ and Ti-O 2$p_{1/2}$ in MC fiber shifted to 459.16 eV and 463.7 eV from 459.0 and 463.5 eV in MX fiber (Supplementary Fig. 11), further confirming the formation of hydrogen bonds between CNTs and MXene nanosheets. Simultaneously, the binding energy of Ti-O in the MCP fiber shifted up to 459.4 eV and 464.1 eV compared to those in the MC fiber, verifying the hydrogen bonds between PLA and MXene nanosheets. These findings suggested that the short CNTs statically filled the voids caused by the transverse wrinkles through hydrogen bonds to effectively fabricate the scaled-up MC fiber (Supplementary Fig. 12). Subsequently, the PLA further bridged the inner MXene

nanosheets by hydrogen bonds during the dynamic thermal drawing process, achieving dense ultrastrong MCP fibers with the length of thousands of meters, including various colors (Supplementary Fig. 12). These MCP fibers are capable of bearing a load of 1.5 kg and withstand several loading cycles, as demonstrated in Supplementary Movie 3.

Figure 2a illustrates that the MX fiber exhibited numerous voids in its cross-sectional SEM, obtained by focused ion beam (FIB). Also, as demonstrated in the HR-TEM images of the cross-section of the fiber in Fig. 2b, a large number of transverse wrinkles were formed with numerous voids due to the weak interfacial interactions among MXene nanosheets. However, after short CNTs were used to statically fill the voids among MXene nanosheets, the MC fiber with the optional weight percentage of 3% CNTs (designated as MC-3%) was successfully fabricated with a diameter of ~102.0 μm, exhibiting a denser structure than the MX fiber (Supplementary Fig. 13). Then, due to dynamic thermal drawing stresses, an even denser MCP fiber with the diameter ~158.6 μm was fabricated by further reducing the voids with an in situ generated encapsulation layer of PLA during the thermal drawing process, as evidenced by the SEM image of cross-section in Fig. 2c. Moreover, as depicted in Fig. 2d, several stacked CNTs filled the voids among MXene nanosheets within the MCP fiber, resulting a denser structure. The voids in the obtained MXene composite fibers were further evaluated through the three-dimensional (3D) reconstruction using nano-computed tomography (nano-CT). Figure 2e, f and Supplementary Fig. 13 illustrate that the MX fibers contained numerous voids (red color) among MXene nanosheets (blue and its transparent colors) within the fibers (Supplementary Movie 4), whereas the MC fibers exhibited significantly fewer voids using the static filling method. By contrast, few voids within the denser microstructure were observed in MCP fibers with the encapsulation layer of PLA (yellow and its transparent colors) (Supplementary Movie 5), aligning with the findings from SEM and TEM images. Meanwhile, despite applying pre-stretching during the wet spinning process, a few wrinkles were visible in the axial section of the MX fiber, as shown in Fig. 2g. This is attributed to the weak interfacial interactions between MXene nanosheets along the fiber axis. However, when employing static filling method combined with dynamic thermal drawing through hydrogen bonds, the alignment of MXene nanosheets within fibers was significantly enhanced, as shown in Fig. 2h, further confirmed by the axial section of the 3D reconstruction with few voids (Fig. 2i). Furthermore, the calculated volume of voids in MX fiber is ~0.7 × 10$^8$ nm$^3$ with the length distributed at ~950 nm in the axial direction, as shown in Supplementary Fig. 13. However, the resulting MCP fibers exhibited lower volumes of voids of ~0.1 × 10$^7$ nm$^3$ with lengths shorter than 250 nm. These results indicate that the short CNTs effectively filled the voids of MXene fibers, and the MC fibers were further densified to minimize the residual voids under dynamic drawing stresses induced by the thermal drawing.

The length of CNTs is a key factor in this static filling method. CNTs of varying lengths, from ~0.19 to ~13.91 μm, were used as the filler to fabricate MC fibers with a fixed weight percentage of ~1% (Supplementary Fig. 14). As the CNTs length increased to ~0.46 μm, the porosity of the MC fibers decreased from ~18.8% to ~15.8%, compared to the ~22.7% porosity of the MX fiber. Meanwhile, the orientation factor ($f$) of MC fibers improved from ~0.842 to ~0.848, compared to that of MX with ~0.831 (Supplementary Figs. 15–17, and Supplementary Table 1). This improvement was attributed to the fact that shorter CNTs filled the voids more effectively and enhanced the alignment of MXene nanosheets through hydrogen bonds. However, due to their shorter length (~0.19 μm), the interfacial interactions were weaker when filling the voids between the nanosheets, as compared to longer CNTs (~0.46 μm). In contrast, longer CNTs tended to increase the porosity, as their lengths exceeded those of the voids. As a result, the MC fibers, using CNTs with an optimized length of ~0.46 μm, exhibited a high tensile strength up to 284.3 ± 4.7 MPa and electrical

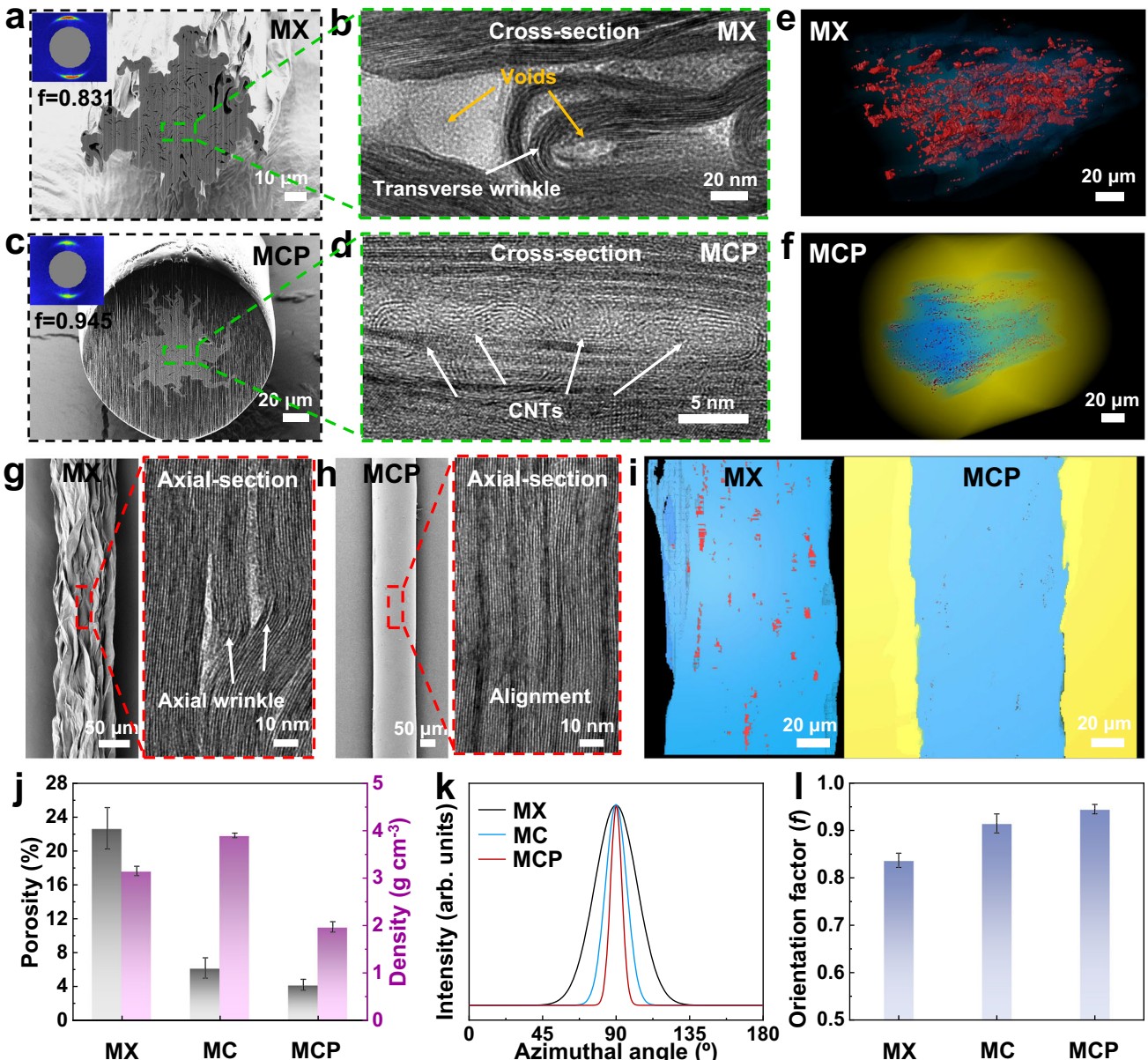

**Fig. 2 | Morphological characterization of MXene composite fibers.** Cross-section scanning electron microscope (SEM) image (**a**) and high-resolution transmission electron microscopy (HR-TEM) image (**b**) for MX fiber. Cross-section SEM image (**c**) and HR-TEM image (**d**) for the MCP fiber. Three-dimensional (3D) reconstruction of the whole fibers along the axial direction with voids in MX (**e**) and MCP (**f**) fibers by nano-computed tomography (nano-CT), while the blue transparent color for MXene nanosheets in MX fiber, and that for MXene nanosheets and CNTs in MCP fiber, the red color for voids, and the yellow transparent color for the PLA. HR-TEM images of the axial section for MX (**g**) and MCP (**h**) fibers. **i** 3D reconstruction of the axial section with voids in MX and MCP fibers by nano-CT, while the blue color for MXene nanosheets in MX fiber, and that for MXene nanosheets and CNTs in MCP fiber, the red color for voids, and the yellow color for the PLA. **j** Bar plot of porosity and density of the MXene composite fibers. **k** Plots of azimuthal angle of the MXene composite fibers. **l** The *f* of the MXene composite fibers. All error bars show mean ± standard deviation.

conductivity of $9597.3 \pm 40.7$ S cm$^{-1}$, compared to $109.3 \pm 2.2$ MPa and $8944.1 \pm 85.8$ S cm$^{-1}$ for the MX fiber (Supplementary Figs. 18 and 19, and Supplementary Tables 2 and 3), thanks to the effective filling by the shorter CNTs[30]. Furthermore, various weight percentages of short CNTs with an optimized length of ~0.46 μm, ranging from 1% to 5%, were used as the filler to fabricate the MC fibers (denoted as MC-1% to MC-5%). As depicted in Fig. 2j, Supplementary Fig. 20, and Supplementary Tables 4 and 5, the porosity of MC fiber (MC-3%) sharply reduced to ~6.2% when using 3 wt% CNTs (as calculated by thermogravimetric analysis in Supplementary Fig. 21), which was lower than that of MX fibers. Meanwhile, the MC fiber has a high density of $3.90$ g cm$^{-3}$ compared to $3.15$ g cm$^{-3}$ for the MX fiber. Moreover, the *f* of

the obtained MC fibers increased to ~0.915 because of the strong hydrogen bonds between CNTs and MXene nanosheets, surpassing that of the MX fiber (Fig. 2k, l, and Supplementary Fig. 22). However, using more than 3 wt% CNTs as the fillers led to an increase in the porosity and a decrease in the alignment of the MXene nanosheets. This was further evidenced by the SEM images with the denser cross-sections for the MC fiber with 3 wt% CNTs, as illustrated in Supplementary Figs. 23 and 24. The dynamic thermal drawing process requires enough ultimate tensile force for MC fibers to fabricate an encapsulation layer. In this work, the MC-180 fiber fabricated with a 180 μm nozzle (Supplementary Figs. 25–29 and Supplementary Tables 6–8) shows only an ultimate tensile force of ~1.36 N, which

cannot meet the dynamic thermal drawing process. Thus, we further increase the spinning nozzle diameter from 180 to 380 μm, and then the achieved MC-380 fiber with a diameter of ~102.0 μm reaches the ultimate tensile force of ~4.87 N, which is successfully fabricated into MCP fibers with a diameter of ~158.6 μm. These MCP fibers have an ultimate tensile force of ~18.59 N, which effectively meets the load-bearing requirements for practical applications, such as embroidery techniques. Additionally, their alignment with operating radio frequencies (e.g., 13.56 MHz) due to their suitable diameter makes them well-suited for wearable wireless textiles.

Next, dynamic and continuous thermal drawing was developed to further densify the MC fiber, incorporating an in situ generated PLA encapsulation layer through hydrogen bonds, as shown in Supplementary Fig. 30. The MC fiber was fed into a hollow PLA preform and thermally drawn at a central temperature of 210 °C. At this moment, some voids within the inner MC fiber were generated with the porosity increasing to ~14.1% due to the removal of intercalated water and partial loss of hydroxyl surface terminations at elevated temperatures[31], testified by in situ XRD patterns obtained during fiber heating from 25 °C to 210 °C in Supplementary Fig. 31 and Supplementary Table 9. To evaluate the mechanical behavior during the thermal drawing process, an FEA model was constructed featuring the inner MC fiber and the outer PLA layer, forming a contact with each other when the draw-down ratio ($\tau$) was 47, as shown in Supplementary Fig. 30. Both dynamic transverse and axial stresses resulted from the thermal drawing process were applied to the inner MC fiber. Furthermore, an increase in the $\tau$ from 47 to 71 raised the transverse stress to 17.3 MPa, further reducing the residual voids, while the axial stress increased to 34.5 MPa to enhance the axial alignment of MXene nanosheets and PLA chains. Here, the MCP fibers were fabricated through dynamic thermal drawing using various $\tau$ of 55, 63, 67, and 71, and were labeled as MCP-55, MCP-63, MCP-67, and MCP-71, respectively. When the $\tau$ reached 71, the porosity of the MCP fiber (MCP-71) was effectively reduced to ~4.2% because of the strong transverse stress induced by the thermal drawing (Fig. 2i, Supplementary Fig. 32, and Supplementary Tables 10 and 11). Moreover, the MCP fiber has a lower density due to an in situ generated encapsulation PLA layer, compared to the MX and MC fibers. Simultaneously, the increased dynamic axial stress significantly improved the $f$ of inner MC fiber of MCP fibers to ~0.945 with denser microstructures, as well as that of the PLA chains (Fig. 2k, l, Supplementary Figs. 33–35, and Supplementary Table 12). These were also confirmed by the cross-section SEM images of MCP fibers with the diameter decreased from ~188.1 to ~158.6 μm, as shown in Supplementary Fig. 36 and Supplementary Table 13.

## Mechanical properties of MXene composite fibers

As a result of filling CNTs into MXene nanosheets through hydrogen bonds in the wet spinning, the MC fibers with 3 wt% CNTs demonstrated superior tensile strength of 631.8 ± 4.5 MPa with the toughness of 23.2 ± 1.2 MJ m$^{-3}$ and conductivity of 11,364.8 ± 65.8 S cm$^{-1}$, compared to the tensile stress of 109.3 ± 2.2 MPa for MX fiber with the toughness of 0.4 ± 0.02 MJ m$^{-3}$ and conductivity of 8,944.1 ± 85.8 S cm$^{-1}$ (Fig. 3a, Supplementary Figs. 37 and 38, and Supplementary Tables 14 and 15). However, as the weight percentage of CNTs increased, both the tensile strength and toughness of the MC fibers decreased, along with a reduction in electrical conductivity. This reduction is attributed to the degraded stress-transfer efficiency and electron transfer pathway due to excessive CNTs aggregated between MXene nanosheets with high porosity[32]. When the draw-down ratio increased during dynamic thermal drawing, the tensile strength and toughness of MCP fibers were further enhanced from 418.3 ± 5.9 to 941.5 ± 5.9 MPa and from 180.4 ± 6.9 to 147.9 ± 4.2 MJ m$^{-3}$, respectively (Supplementary Figs. 39 and 40, and Supplementary Table 16). These properties were 9 times and 411 times greater than those of MX fibers. Additionally, the MCP fibers had remarkable electrical conductivity,

reaching up to 3899.0 S cm$^{-1}$ with an even higher electrical conductivity of ~12,836.4 S cm$^{-1}$ of the inner MXene fiber in Supplementary Table 17. The aforementioned enhancements can be attributed to the further reduction in porosity and the significant improvement in the alignment of fibers. As illustrated in Fig. 3b and Supplementary Table 18, the obtained MCP fiber (MCP$_{core}$ for the inner MXene fiber in whole composite MCP fiber) achieved the record tensile strength and high electrical conductivity, toughness, and specific strength compared with the reported MXene-based fibers through different methods, including wet and dry spinning, coating, scrolling, wet spinning/thermal drawing, and thermal drawing[14–16,18–20,25,33–54]. Meanwhile, our MXene composite fibers also exhibited superior performance in toughness, surpassing those of the graphene-based fibers, carbon fibers, some CNT fibers, liquid metal fiber, Ag-coated Kevlar fiber, Ag-coated Nylon fiber, and metal wires through different preparation methods, as shown in Supplementary Table 19.

Furthermore, the real-time resistance-strain curves of the MX, MC, and MCP fibers, as shown in Fig. 3c. The MCP fiber exhibits a ~2% change in resistance prior to the fracture during a single stretch intact throughout this deformation range, compared to ~69% for the MX fiber and ~26% for the MC fiber. The results indicate that the MXene component within the PLA matrix remains intact throughout this deformation range, and the MXene structure inside the PLA remains completely undamaged up to the point of failure, suggesting that MXene and PLA undergo co-fracture. Meanwhile, the cyclic loading behavior of the prepared MXene composite fibers was assessed under various loading stresses (Supplementary Fig. 41). Under a loading stress of 500 MPa, the MC fiber maintained an electrical conductivity retention rate of 68.4% after 4000 cycles, whereas the MX fiber failed after just 200 cycles. In contrast, the MCP-71 fiber maintained 85.6% after 4000 cycles. Under a loading stress of 250 MPa, the MCP-71 fiber could retain a high electrical conductivity rate of 98.8% after 4000 cycles (Supplementary Table 20). These high-performance metrics are attributed to the dense microstructure of the MCP fiber with the encapsulation layer, which is enhanced by the static filling and dynamic thermal drawing processes through hydrogen bonds.

## Fracture mechanism of MXene composite fibers

Density functional theory (DFT) calculation was conducted to analyze the interfacial interactions within MCP fibers across four different interfaces, including MXene-MXene, CNTs-CNTs, CNTs-MXene, and PLA-MXene (Supplementary Fig. 42). The absorbed energy ($E_a$) and electron transfer number ($ET$) were evaluated to assess the bond strength at these interfaces[55]. Figure 3d illustrates a trend that bond strength increased with the rising $E_a$ and $ET$ between the surfaces. According to these calculations, the PLA-MXene and CNTs-MXene interfaces exhibit relatively high $E_a$ of −7.15 eV and −6.82 eV, respectively. These values are significantly higher than those for CNTs-CNTs (−1.46 eV) and MXene-MXene (−1.35 eV) interactions. Additionally, the $ET$ indicated the interaction strengths follow a similar sequence of MXene-MXene with a value of 0.026 e <CNTs-CNTs, with that of 0.032 e <CNTs-MXene, with that of 0.855 e for hydrogen bond <PLA-MXene with that of 1.037 e for hydrogen bond. The results indicate that the hydrogen bond interactions at the PLA-MXene and CNT-MXene interfaces are stronger than those at the MXene-MXene and CNT-CNT interfaces.

Therefore, the possible fracture mechanism of the MCP fiber is depicted in Fig. 3e and Supplementary Fig. 43. Initially, during stretching, neighboring MXene nanosheets slide relative to each other because of the lowest $E_a$ and $ET$ values for MXene-MXene, alongside the straightening of molecular chains in the encapsulation PLA layer. As the tensile load increases, the hydrogen bonds between short CNTs and MXene nanosheets, exhibiting higher $E_a$ and $ET$ values for CNTs-MXene, gradually begin to break, leading to increased energy dissipation. Subsequently, further sliding takes place between MXene

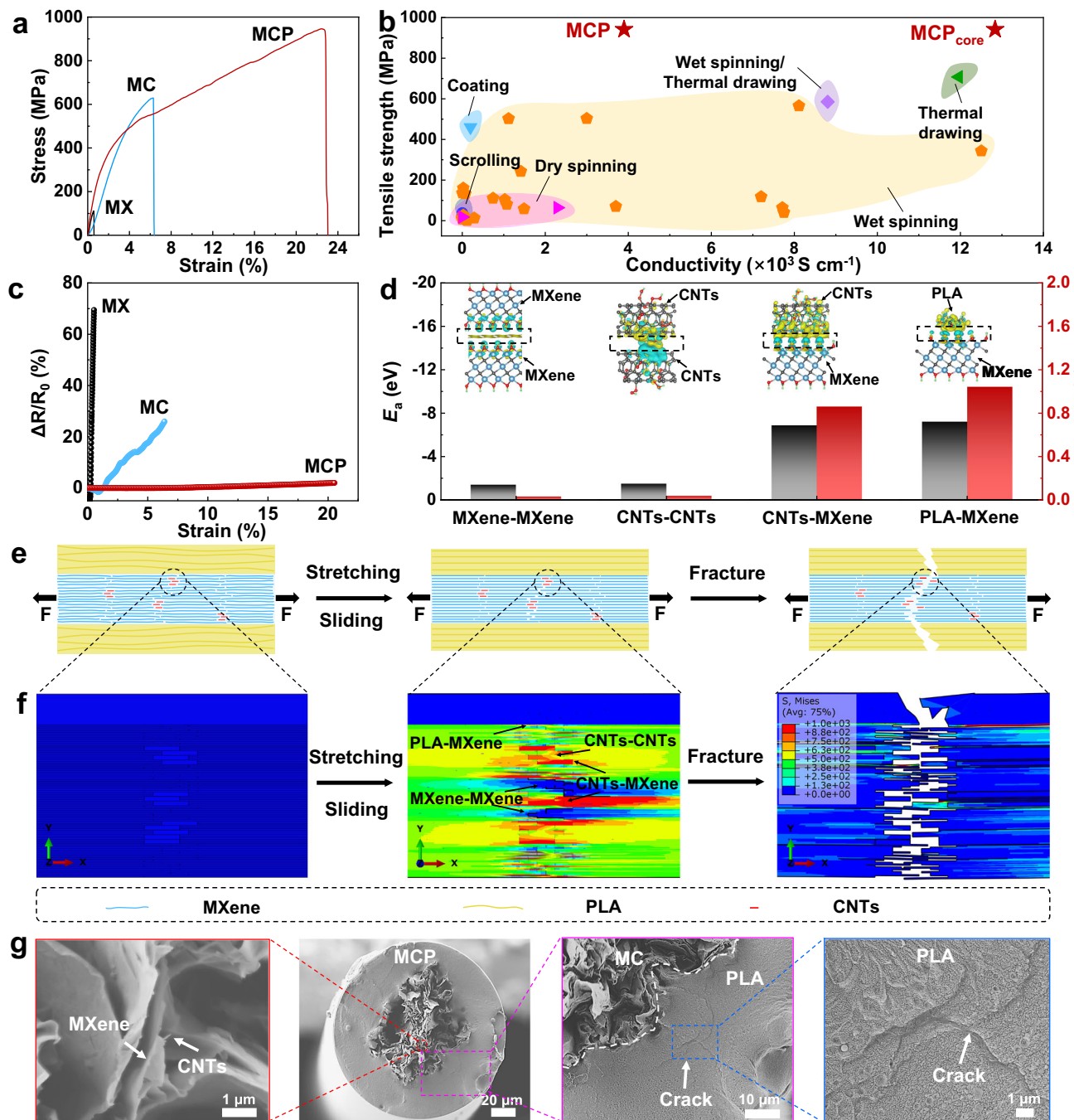

**Fig. 3 | Mechanical properties and fracture mechanism of MXene composite fiber. a** Stress–strain curves of the obtained MXene composite fibers. **b** Comparison of tensile strength and conductivity between the MCP fiber and the previously reported MXene-based fibers. **c** Real-time resistance-strain curves of MXene composite fibers. **d** Density functional theory (DFT) calculation of absorbed energy ($E_a$) and electron transfer number ($ET$) for four different interfaces, including MXene-MXene, CNTs-CNTs, CNTs-MXene, and PLA-MXene. **e** Fracture schematic diagram of the MCP fiber, as well as sliding between the MXene nanosheets and failure of the strong hydrogen bonds of CNTs-MXene and PLA-MXene interfaces. **f** FEA of the local fracture process of the MCP fiber. **g** Fracture morphology of the MCP fiber with the pull out of short CNTs and the crack of the encapsulated PLA layer.

nanosheets as well as among CNTs. Furthermore, the fracture of the hydrogen bonds between the encapsulation PLA layer and MXene nanosheets with the highest $E_a$ and $ET$ for PLA-MXene initiates and progressively straightens along the loading direction. Upon continued stretching, these hydrogen bonds are completely disrupted. The pulled-out CNTs indicate that the interfacial interactions between the short CNTs and MXene have been completely broken, which enhances energy dissipation and contributes to improved tensile strength. This whole process of the break of interfacial bonds effectively enhances

stress-transfer efficiency and substantially increases energy dissipation. Moreover, as demonstrated in Supplementary Fig. 43, an FEA model for a segment of the MCP fiber was developed to further reveal the fracture mechanism of the fiber. This model incorporated four different cohesive elements, and each represents a distinct interface, along with MXene nanosheets having a thickness of 1.5 nm and short CNTs with a diameter of 4.5 nm. As shown in Fig. 3f, Supplementary Fig. 44, and Supplementary Movie 6, the simulation results indicate that sliding occurs between MXene nanosheets upon loading. As the

stretching continues, initial cracking in the MCP fiber is triggered by the failure at the interface between short CNTs and MXene nanosheets, leading to the breakdown of the hydrogen bonds. Subsequently, the hydrogen bonds between the PLA layer and MXene nanosheets breakdown, resulting in the complete fracture of the entire model. The simulated results closely align with those from the DFT calculations.

In addition, as shown in Supplementary Fig. 45, the MX fiber exhibited flat fracture edges on the MXene nanosheets due to the weak interfacial interactions between them, according to the fracture morphologies observed in the front view. For the case that short CNTs filled the voids within MX fiber to fabricate the MC fiber through hydrogen bonds, CNTs were pulled out from the layers of MXene nanosheets, accompanied by curled fracture edges of MXene nanosheets to effectively enhance the stress-transfer efficiency between CNTs and MXene nanosheet, as shown in Supplementary Fig. 45. By contrast, the MCP fiber, fabricated from MC fiber by thermal drawing through hydrogen bonds, not only demonstrated the pull out of CNTs from the MXene layers with the obviously curled fracture edges but also exhibited cracks in the encapsulation layer of PLA, as shown in Fig. 3g and Supplementary Fig. 45. This indicates that the hydrogen bonds between the CNTs and MXene nanosheet, as well as between the PLA and MXene nanosheet in denser MCP fiber, substantially improved stress-transfer efficiency, thereby enhancing the tensile strength of the fibers.

## Electromagnetic performance and mechanical durability of smart textiles

Smart electronic textiles offer a compelling way to integrate digital technology with the human body, potentially eliminating the intrusiveness and risks associated with implanted or epidermal electronics[56], all while maintaining wearer comfort and not interfering with daily activities. Different from conventional electronics that are rigid and flat, these textiles are crafted from flexible fibers designed to withstand the mechanical, thermal, and chemical stresses of daily wear and laundering[25]. Our MCP fiber, featuring ultrahigh tensile strength and excellent electrical conductivity with in situ generation of encapsulation PLA layer, is promising for use in smart electronic textiles. Utilizing digitally controlled embroidery techniques, we precisely integrated computationally optimized electromagnetic patterns into existing textiles, as demonstrated in Supplementary Fig. 46 and Supplementary Movie 7. The textile, embroidered with a spiral inductor of a flower pattern and optimized for a 1-mm gap and 8 turns (Fig. 4a) (designated as MCP textile), enabled effective wireless transfer at the frequency of 13.56 MHz. Other rectangle and circle patterns are shown in Supplementary Fig. 46. The resulting smart MCP textile demonstrated high stability, maintaining their performance when flatted, bent, twisted, or stretched without any damage, as shown in Supplementary Fig. 47. For comparison, we also embroidered commercial copper wire (Cu) (commonly used in wireless communications) with a diameter of 100 μm and polyimide-encapsulation copper wire (E-Cu) with an inner conductive Cu wire diameter of 100 μm, into the textiles using the same flower spiral inductor pattern, designated as Cu textile and E-Cu textile in Supplementary Fig. 48, respectively. Figure 4b suggests that the MCP textile with the flower spiral inductor pattern offered the high electromagnetic performance of the scattering parameters (S-parameters, $S_{11}$) values of −25.0 dB at the frequency band of 13.56 MHz, compared to −24.2 dB for Cu textile and −24.6 dB for E-Cu textile. Additionally, the MCP textiles achieved a quality factor (Q) higher than 10 across all textiles at the frequency band of 13.56 MHz (Supplementary Fig. 49), indicating their potential for wireless systems.

To ensure suitability for daily wear, the mechanical durability of the smart textile with the flower spiral inductor pattern was rigorously evaluated. Figure 4c demonstrates that the MCP textile maintained a stable $S_{11}$ of −25 dB and high electrical conductivity across bending

angles ranging from 0° to 150°, confirming its ability to function normally at various bending angles[57]. As shown in Supplementary Fig. 49, thanks to the excellent tensile strength and toughness of MCP fiber, when they were embroidered into the textile, the smart textile exhibited a minimal conductivity relative change of -0.3% after $9 \times 10^4$ cycles of 180° bending, as well as the stable $S_{11}$ parameters. In contrast, the Cu textile fractured after only 300 cycles of bending, while the E-Cu textile lasted 1450 cycles. Moreover, after $3 \times 10^4$ twisting cycles of 360°, the MCP textile retained a high conductivity of 12,772.2 S cm$^{-1}$ of the inner MXene fiber (99.5% retention), while the Cu and E-Cu textiles broke after just 150 and 650 cycles, respectively. After $5 \times 10^4$ cycles of stretching under 20% strain, the MCP textile also maintained a conductivity of -12,797.9 S cm$^{-1}$ of the inner MXene fiber (99.7% retention), far surpassing the Cu textile, which fractured after $6 \times 10^3$ cycles, and the E-Cu textile after $1.5 \times 10^4$ cycles. Furthermore, as shown in Fig. 4d and Supplementary Fig. 50, after 90 cycles of washing, the MCP fiber in the textile performed the conductivity of -12,810.7 S cm$^{-1}$ of the inner MXene fiber (99.8% retention), demonstrating superior resistance to mechanical stresses compared to the Cu and E-Cu textiles, which fractured after 4 and 20 cycles of washing. This difference in durability is attributed to the higher tensile strength and toughness of the MCP fiber with denser microstructure, larger than that of -260.3 MPa and -41.53 MJ m$^{-3}$ for Cu wire, and that of -177.0 MPa and -65.74 MJ m$^{-3}$ for E-Cu wire, as illustrated in Supplementary Fig. 49. The results verify that the smart MCP textile is an ideal candidate for wireless communication applications, offering high mechanical durability for daily use.

Moreover, as shown in Supplementary Fig. 51, the MCP spiral inductor demonstrates a high power-transfer efficiency (η) of -51% at a specific distance between the inductor and receiver. While η decreases with increasing distance, it remains stable, even after enduring $9 \times 10^4$ bending cycles at a 180°, $3 \times 10^4$ twisting cycles, $5 \times 10^4$ stretching cycles under 20% strain, and $1 \times 10^4$ poking cycles with displacement ranging from 0 mm to 20 mm. Furthermore, the inductance (L) maintains a high value of -3 μH at 13.56 MHz even after the same mechanical durability tests. Additionally, the measured resistance ($R$s), L, and Q of the MCP spiral inductor textile remain consistent under 0–20% strain at 13.56 MHz, even after $5 \times 10^4$ stretching cycles. This excellent performance is primarily attributed to the high tensile strength and toughness of the dense MCP fiber with high electrical conductivity. The smart textile demonstrates exceptional bending durability, outperforming materials like liquid metal fibers and metallic wire fibers for wireless applications, as detailed in Supplementary Table 21. Additionally, despite embedding MCP fiber, the moisture vapor transmission rate (MVTR) of the MCP textiles remained comparable to that of the pure textile with $1633.21 \pm 50.88$ g m$^{-2}$·24 h, consistent with that of $1629.98 \pm 66.42$ g m$^{-2}$·24 h for the rectangle textile, $1630.50 \pm 46.65$ g m$^{-2}$·24 h for the circle textile, and $1628.91 \pm 58.40$ g m$^{-2}$·24 h for the flower textile in Supplementary Fig. 52. These results indicated minimal variation of MVTR values and demonstrated that the textile's breathability was not compromised by the addition of MCP fibers. However, the commercial rigid printed circuit board (C-PCB) used in wireless communications, with the low MVTR of $26.52 \pm 4.88$ g m$^{-2}$·24 h, fails to meet practical application requirements for smart wearable textiles, further highlighting the superior performance of MCP textiles in daily applications.

## Battery-free wireless human health monitoring system

A wireless power unit was first developed, as demonstrated in Supplementary Fig. 53 and Supplementary Movie 8, which effectively powered light-emitting devices by near field communication (NFC). We assessed the consistency of the wireless power transfer by measuring the rectified voltage at two types of receivers (R2.0 and R3.2), each operating at -2.0 V and -3.2 V, when a human equipped with the wireless power unit and receivers at various walking speeds (v). The received voltage remained stable at around 2.0 V and 3.2 V for both

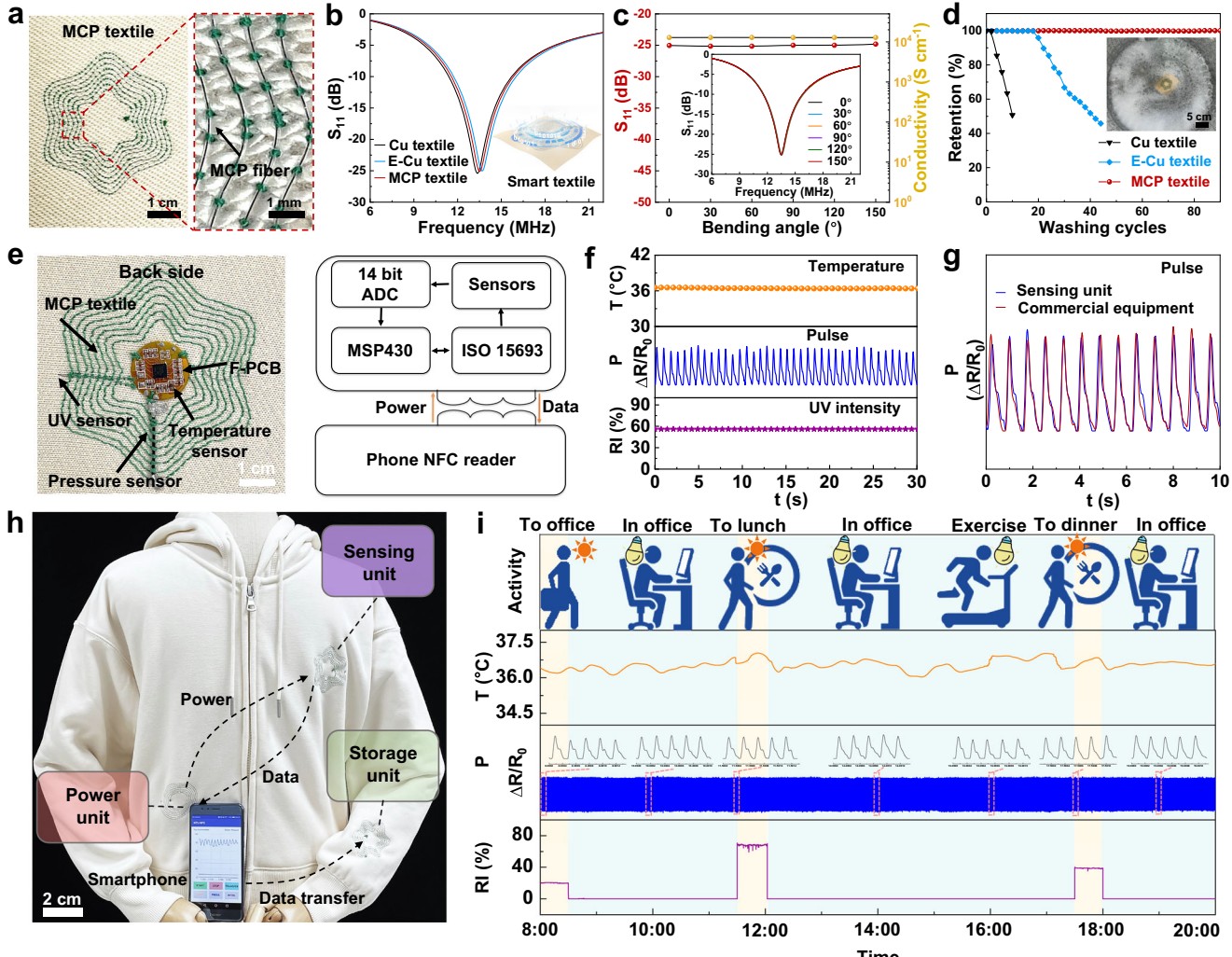

**Fig. 4 | Electromagnetic performance and mechanical durability of smart textiles based on MCP fibers. a** Smart textile embroidered with a flower spiral inductor pattern using the MCP fiber through the embroidery machine method, featuring a gap space of 1 mm and 8 turns. **b** The S-parameters ($S_{11}$) of the smart textiles, embroidered with flower patterns using various fibers, including copper wire (Cu), polyimide-encapsulation copper wire (E-Cu), and MCP fibers. **c** The $S_{11}$ and conductivity of the smart textile with a flower pattern at the bending angles from 0° to 150°. **d** The electrical conductivity retention of the MCP fiber in the smart textile during 90 washing cycles. **e** Photography of the back side of the wireless sensing unit and its block diagram. **f** The stability of the sensing unit, including monitoring human being's temperature ($T$), pulse pressure ($P$), and relative UV intensity (RI) over a 30-s period. **g** Comparison of the $P$ measured by using the sensing unit and commercial equipment. **h** A long-range, battery-free wireless human health monitoring system embedded in a hoodie, which consisted of a wireless power unit, wireless sensing unit, and storage unit. **i** Actual measurement data of the human health monitoring system to monitor the $T$, $P$, and RI over a 12-h period between 8:00 AM and 8:00 PM with different activities.

types of receivers, with $v$ ranging from 0 to 10.5 km h⁻¹. Furthermore, the unit maintained a consistent voltage output for all receivers during various motions over half-hour intervals, such as static, stooping, squatting, walking, and even running. This demonstrated the high stability of the wireless power unit, sufficient to support a wireless system for daily activities. Additionally, our wireless power unit consistently powers the LED light even when subjected to bending angles from 0° to 150°, stretching strains ranging from 0% to 20%, and poking distances from 0 to 20 mm (Supplementary Fig. 54 and Supplementary Movie 9). This performance enables our wireless power unit to be a promising candidate for practical applications. Next, a wireless sensing unit based on a flexible PCB (F-PCB) with a diameter of 1.7 cm, was fixed in the center of the flower spiral inductor pattern, to monitor temperature ($T$), pulse pressure ($P$), and relative UV intensity (RI), as shown in Fig. 4e. Data from the sensing unit was transferred to a database in the data storage unit for storage and further analysis. Figure 4f and Supplementary Movie 10 demonstrate that this wireless sensing unit consistently and stably monitored the $T$, $P$, and RT over 30 s, indicating

its practical application compared with commercial equipment (Fig. 4g). Finally, users can access and evaluate these data through the wireless storage unit by NFC touching through the smartphone application or other devices, as shown in Supplementary Figs. 55 and 56, and Supplementary Movie 11. Meanwhile, this storage unit still functioned normally in water for 30 days and under a pressing load ranging from 0 to 120 kg, as evidenced in Supplementary Fig. 56 and Supplementary Movie 12.

Lastly, a long-range, battery-free wireless human health monitoring system using these spiral inductor patterns was achieved through NFC at the frequency of 13.56 MHz[56], comprising a wireless power unit, a wireless sensing unit, and a data storage unit, as shown in Fig. 4h and Supplementary Fig. 56. When a smartphone came in contact with the circle spiral inductor pattern on the hoodie, it powered the sensing unit embedded in the flower spiral inductor pattern on the chest. This step enabled long-range monitoring the human health, tracking $T$, $P$, and RI. Simultaneously, the smartphone with the customized application recorded data through wireless communication. The data was

then transferred to the data storage unit located in the flower spiral inductor pattern on the wrist for further assessment. Our system effectively functioned at distances exceeding 50 cm, overcoming the limitations of NFC, which typically operates within a range of less than 10 cm. With its high electromagnetic performance and remarkable mechanical durability, this system can effectively monitor our various daily activities such as walking, working, dining out, and exercising between 8:00 AM and 8:00 PM over the 12-h period, as shown in Fig. 4i. Notably, $T$ increased during outdoor activities such as commuting to the office, going to lunch, exercising, and dining out, due to the generated body heat by these activities. Meanwhile, the $P$ had slight fluctuations in the pulse signal during these activities (inset for $P$ data). Additionally, RI rose during outdoor daily activities and varied throughout the duration, peaking at ~58% between 11:30 AM and 12:00 PM. Additionally, the entire wireless system operated effectively across temperatures ranging from −50 °C to 100 °C (Supplementary Fig. 57), further testifying to the suitability of this system for daily applications even under extreme environments.

### Battery-free, body-coupled interaction textile

We applied the static filling method and dynamic thermal drawing to fabricate robust MCP-V fibers by coating MC-PLA-BaTiO$_3$ (denoted as MCP$_{Ba}$) fibers, fabricated from MC fibers and PLA-BaTiO$_3$ via thermal drawing (as illustrated in Supplementary Figs. 58 and 59). The resulting MCP-V fibers exhibited a three-layer structure, consisting of the MC fiber core, a PLA-BaTiO$_3$ layer, and an outer VSASR-ZnS-Cu$^{2+}$ coating, as shown in Supplementary Figs. 60 and 61. Consequently, the MCP-V fibers had a high orientation factor of 0.951 and low porosity (~4.1%) after coating (Supplementary Fig. 62 and Supplementary Table 22), resulting in a high tensile strength of 1088.5 ± 10.1 MPa, toughness of 293.5 ± 10.5 MJ m$^{-3}$, and excellent electrical conductivity of 13,567.4 ± 210.5 S cm$^{-1}$ of the inner MXene fiber (Supplementary Table 23). Additionally, the fibers exhibited a high specific strength of 622.0 ± 5.8 MPa cm$^3$ g$^{-1}$, and a low density of ~1.75 g cm$^{-3}$. Hundreds of meters of MCP-V fibers were successfully fabricated, as shown in Supplementary Fig. 63.

As illustrated in Supplementary Fig. 64 and Fig. 5a, MCP-V fibers with different colors are integrated with a wireless receiver and a transmitter to achieve body-coupled sensing and display functionalities[1]. It is composed of three functional layers (Supplementary Fig. 59): the antenna core (MC fibers, blue) for generating an alternating electromagnetic field, the dielectric layer (BaTiO$_3$ mixed PLA, yellow) for storing coupled energy, and the optical layer (ZnS-Cu$^{2+}$ mixed VSASR, transparent) for visualizing electric fields. The human body, serving as the opposing pole, helps confine the ambient electromagnetic field. When the electric field intensity at the interface capacitance exceeds the air breakdown threshold, a localized plasma discharge occurs, producing an additional electric displacement term. This disrupts the equilibrium of confined charge pairs, enabling wireless transmission of tactile sensing signals. Upon contact with a finger, the MCP-V fiber's interface capacitance efficiently captures ambient electromagnetic energy, causing the fiber to emit light (Supplementary Movie 13). Moreover, the MCP-V fiber is capable of emitting high-luminance light (~110 cd cm$^{-2}$) and producing electrical signals with an amplitude of −20 dBm at ~13.56 MHz (Supplementary Fig. 64), all without batteries or additional components.

Additionally, thanks to their high tensile strength, toughness, and functionality for body-coupled wireless optical and electrical signal transmission, our MCP-V fibers can be embroidered onto textiles to create various patterns and letters that emit light with different colors for assisted communication, as demonstrated in Fig. 5b, c, Supplementary Fig. 65, and Supplementary Movie 14. The fiber maintains a stable red light emission even under stretching, bending, and poking, as shown in Fig. 5d. Remarkably, after $9 \times 10^4$ bending cycles at 180°, $5 \times 10^4$ stretching cycles at 18% strain, and $2 \times 10^4$ poking cycles at a

distance of 30 mm, the MCP-V fiber textile retains its high-luminance light output and stable electrical signals (Supplementary Fig. 66 and Supplementary Movie 15). This high durability is attributed to the dense structure of fibers, fabricated by combining static filling and dynamic thermal drawing. Furthermore, the resulting body-coupled MCP-V fiber textile demonstrates high-performance wireless electrical signal transmission, enabling it to function as a battery-free remote control for electronic devices, such as drones. As shown in Fig. 5e–g and Supplementary Fig. 67, the textile enables precise and reliable remote control of drone operations, including takeoff, landing, ascent, descent, forward and backward movement, and even load/unload actions (Supplementary Movie 16). Moreover, the MCP-V fiber textile can control drone operations without any interference over a distance of ~5 km, from destination A to destination C with outdoor settings (Fig. 5h). In summary, our fibers with high tensile strength and toughness, achieved through advanced fabrication techniques of static filling and dynamic thermal drawing, are highly suitable for smart textiles to engage daily applications.

## Discussion

This work demonstrated a controllable strategy through the combination of static filling method and dynamic thermal drawing to reduce the voids caused by the transverse wrinkles between MXene nanosheets, achieving dense MXene composite fibers with an in situ generated encapsulation layer. The resulting fiber performed ultrahigh tensile strength and toughness, accompanied by high electrical conductivity. Using this static-dynamic densification strategy, the porosity of the MCP fiber was substantially reduced, together with the effective enhancement of the alignment of the MXene nanosheets. The interfacial interactions within fibers were promoted because of the formation of strong hydrogen bonds, further enhancing the stress-transfer efficiency and improving the mechanical properties. Additionally, the smart textile crafted from the MCP fiber showed high electromagnetic performance and durability even under complex deformations. The long-range battery-free wireless human health monitoring system offered a substantially stable human health monitor, including temperature, pulse pressure, and UV intensity, during daily activities. Furthermore, the smart wireless textile prepared from the ultrastrong body-coupled MXene composite fiber achieved remote drone operation and enabled assisted communication. Therefore, the demonstrated static-dynamic densification strategy would be broadly applied to fabricate high-performance fibers with significantly reduced voids based on a wide range of nanostructured functional materials, fundamentally enhancing their properties to meet diverse requirements.

## Methods

### Materials

Ti$_3$AlC$_2$ powders with a particle size of approximately 400 mesh were purchased from Jilin 11 Technology Co., Ltd. Lithium fluoride was obtained from Sigma-Aldrich Co., Ltd. Polylactic acid (PLA) was produced by Kai Feng Co., Ltd. Carboxylated multiwalled CNTs were supplied by Xian Feng Co., Ltd. BaTiO$_3$ was obtained from Sigma-Aldrich Co., Ltd. Vinyl silicone resin (Sylgard 184) was sourced from Dow Corning. Acetoxy silicone resin (SI 595) was purchased from LOCTITE Co., Ltd. ZnS-Cu$^{2+}$ phosphor powder was acquired from Shanghai Keyan Phosphor Technology, China. Fluorescent dyes were also obtained from Sigma-Aldrich Co., Ltd.

### Characterization

SEM images were captured utilizing a ZEISS Crossbeam 540 FIB-SEM at 5 kV. HR-TEM images were obtained with a Grand ARM instrument (ACTEM JEOL JEM-ARM300F) operating at 300 kV from the lamella obtained through FIB-SEM. AFM images were acquired by a Park NX10 AFM. FTIR spectra were recorded at room temperature using Diamond

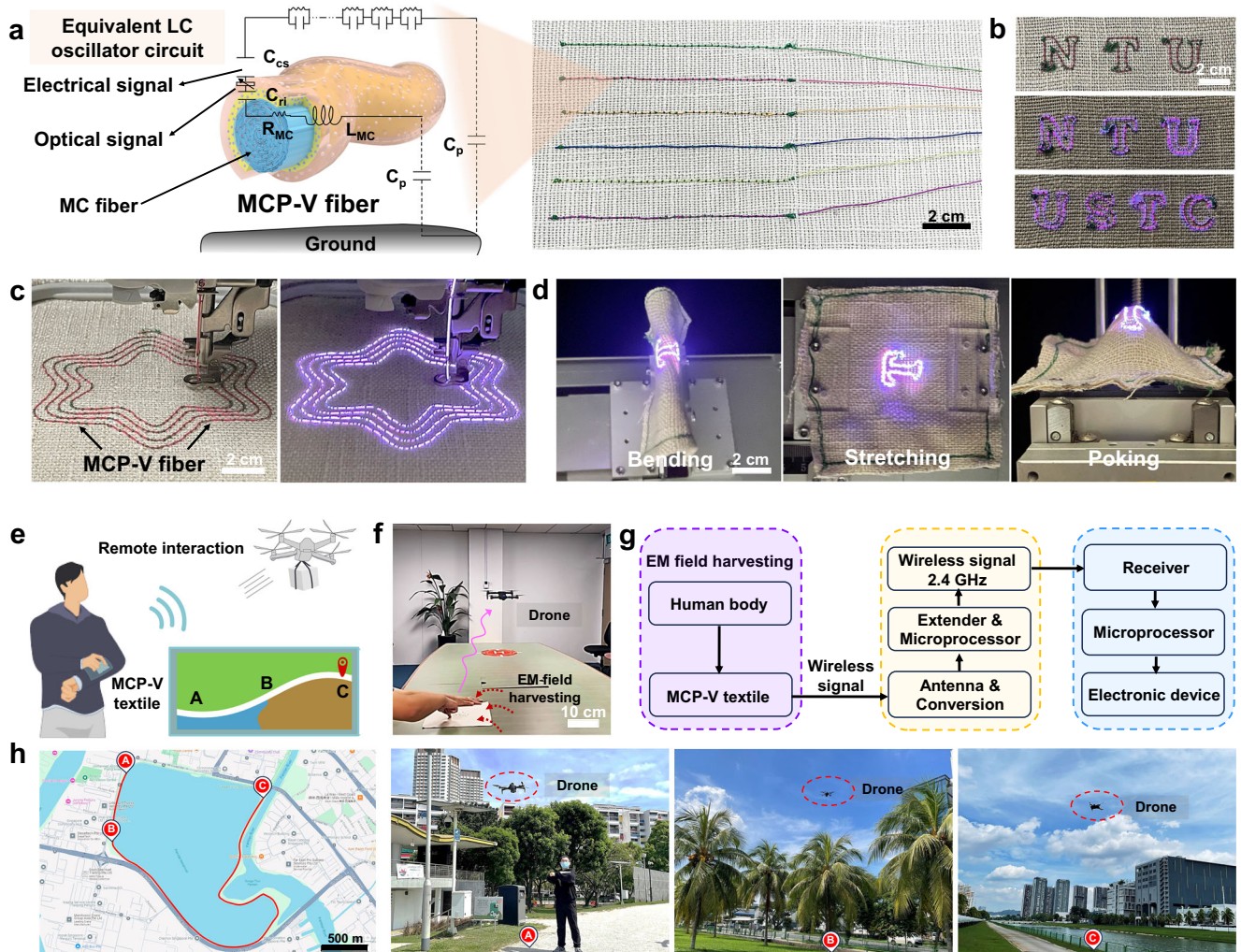

**Fig. 5 | Applications of the MCP-V textile. a** Schematic diagram illustrating the mechanism for wireless optical and electrical signal generation and transmission using a single MCP-V fiber with various colors under body-coupled electromagnetic (EM) fields. Where $C_{cs}$ represents the contact interface capacitance during local discharge at the surface, $C_{ri}$ is the radial interface capacitance, $R_{MC}$ is the resistance, and $L_{MC}$ is the axial fiber inductance provided by the MC fiber. Additionally, $C_p$ is the parasitic capacitance with the ground formed by the human body and the MCP-V fiber. **b** The dyed MCP-V fibers emit a red glow when exposed to body-coupled EM fields for assisted communication. **c** Embroider patterns on textiles using MCP-V fiber. **d** Durability of the textile incorporating MCP-V fiber under bending, stretching, and poking conditions. **e** A conceptual illustration shows how MCP-V textiles can be used to control a drone remotely. **f** Photograph showcasing MCP-V textiles being used to control a drone remotely. **g** Block diagram illustrating the interaction between MCP-V textiles and a drone, depicting signal input from the textile interface, processing of commands, wireless transmission to the drone, and execution of drone actions in response to the received signals. **h** The flight path of the drone controlled by the MCP-V textile, accompanied by real-life images.

ATR by an FTIR Frontier (PerkinElmer). TGA spectra were collected under a nitrogen atmosphere with a Q500 SDT thermal analyzer at a heating rate of 10 °C min⁻¹ between 35 °C and 800 °C. XRD patterns were conducted using Cu-K radiation from an XRD Bruker D8 Advance. Rheological analysis of the MXene-CNTs spinning solutions was performed with a rheometer of TA HR10 under both steady shear and dynamic oscillatory conditions. Additionally, the viscoelastic properties of the spinning dispersion were characterized by measuring the storage and loss modulus over a frequency range of 0.1–100 rad s⁻¹, maintaining a strain amplitude of 0.1% with a 1 mm gap at 25 °C during frequency sweep tests. Polarized optical microscopy images were captured to illustrate optical birefringence using the equipment of Olympus BX51. XPS spectra were acquired by an XPS Kratos AXIS Supra instrument.

WAXS/SAXS patterns were acquired using a SWAXS Xenocs Nanoinxider equipped with a Cu-Kα source and detectors for SAXS (200 K) and WAXS (100 K) operating at 30 W, with beam diameters from 200 to 800 mm. In situ XRD variable temperature experiments were performed using the Linkam temperature stage on the same SWAXS Xenocs Nanoinxider instrument. The cross-sectional morphology of fibers was analyzed using the equipment of ZEISS Crossbeam 540 FIB-SEM. Fiber structures were reconstructed using Avizo software from nano-CT imaging. Tensile stress–strain curves of fiber samples were generated using a SUNS EUT4103X Tester, with samples measuring 20 mm in length and 3 mm in width, applying a loading rate of 0.3 mm min⁻¹ with a 10 N sensor, all at room temperature. Fiber area measurements were conducted using SEM, with the average value calculated from at least three samples for each fiber. Wireless signals at various frequencies were generated by connecting an RF/microwave signal generator (ROHDE & SCHWARZ SMB100B) to power amplifiers, including ATA-7030 and LZY-22X+. The luminous brightness of the fiber and textile was measured using an LS-150 Luminance Meter. The electrical performance of the MCP-V fiber was evaluated with a Keithley Electrometer of Model 6517B. Wireless sensor data and

frequency domain signals were analyzed using a Digital Phosphor Oscilloscope of DPO 5104B Tektronix, and an MXA Signal Analyzer of N9020B Keysight. The optical and electrical signals in this study were conducted using MCP-V fibers of fixed length (15 cm) positioned 15 cm away from the transmitter (30 dBm, ~20 kHz).

## Simulation for the wet spinning process

FEA was employed to assess the shear stress during the wet spinning process. Given the rate-dependent viscosity derived from rheological measurements, the MXene spinning solution demonstrated non-Newtonian flow behavior. This behavior was modeled by fitting the data to the power-law viscosity model, as described below.

$$\eta = k\dot{\gamma}^{n-1} \tag{1}$$

where $\eta$ represents the viscosity, $k$ is the flow consistency index, $\dot{\gamma}$ is the shear rate, and $n$ is the flow power-law index. When $n$ is less than 1, the fluid exhibits shear-thinning: The apparent viscosity decreases as the shear rate increases. Conversely, when $n$ is greater than 1, the fluid is shear-thickening. When $n$ equals 1, the fluid behaves as a Newtonian fluid.

The flow rate ($Q$) was maintained at $2.08 \times 10^{-9}\,\text{m}^{-3}\,\text{s}^{-1}$ with the inner radius ($r$) of the nozzle of $1.9 \times 10^{-4}\,\text{m}$ (190 μm). As a result, the actual $\dot{\gamma}$ during wet spinning is ~386 s$^{-1}$ according to Eq. (2) as following[14].

$$\dot{\gamma} = \frac{4Q}{\pi r^3} \tag{2}$$

where Q is the flow rate (m$^3$ s$^{-1}$), and r is the inner radius of the nozzle (m).

In order to explain the rheological behavior during wet spinning, the viscosity of a pure MXene nanosheet spinning dispersion (at a concentration of 40 mg mL$^{-1}$) was measured as a function of shear rate, up to 400 s$^{-1}$.

Furthermore, the dynamics of the velocity field of the incompressible non-Newtonian fluid are described by the Navier-Stokes equation, which incorporates viscosity dependent on the shear rate:

$$\nabla \cdot \mathbf{v} = 0 \tag{3}$$

$$\frac{\partial}{\partial t}(\rho\mathbf{v}) + \nabla \cdot (\rho\mathbf{v}\mathbf{v}) = -\nabla p\mathbf{I} + \nabla \cdot [\eta(\nabla\mathbf{v} + \nabla\mathbf{v}^T)] \tag{4}$$

where ρ represents the constant density for the incompressible non-Newtonian fluid, $\mathbf{I}$ is the identity matrix, $\mathbf{v}$ is the velocity vector, and p is the fluid pressure. The wet spinning process simulation was conducted using ANSYS Fluent software, version 2020R2. The spinning microchannel was designed within Fluent, and finite elements were generated according to the geometry. Boundary conditions were applied to these elements to replicate the experimental conditions. The simulation provided data on shear stress, shear rate, and viscosity distribution.

## Simulation of the fracture mechanism of the MCP fiber

To analyze the fracture mechanism of MCP fibers, a simplified model was designed featuring MXene nanosheets interspersed with CNTs and encapsulated on one side by a PLA layer. The model was structured with a length of 1000 nm and a total width of 180 nm, which includes 150 nm for the MXene nanosheets layer and 30 nm for the PLA layer. This FEA model was developed using the Abaqus/CAE 2019 software, leveraging Abaqus/Explicit, and a Python 3.9 script to delve deeper into the fracture dynamics. Within this model, MXene nanosheets were crafted with a thickness of 1.5 nm, while CNTs featured a thickness of 4.5 nm and a length of 35 nm. Additionally, MXene nanosheets possess

an isotropic bulk Young's modulus of 330 GPa and a Poisson's ratio of 0.30. CNTs exhibit a bulk Young's modulus of 1 TPa and a Poisson's ratio of 0.30, whereas PLA has a bulk Young's modulus of 0.8 GPa and the same Poisson's ratio of 0.30.

Four types of cohesive elements were constructed to represent the different interfacial interactions observed, including MXene-MXene, CNTs-CNTs, CNTs-MXene, and PLA-MXene. All input parameters were calibrated to align with experimental findings, and the cohesive elements conformed to the traction-separation law of elasticity as outlined:

$$\begin{Bmatrix} t_n \\ t_s \\ t_t \end{Bmatrix} = \begin{bmatrix} E_{nn} & & \\ & E_{ss} & \\ & & E_{tt} \end{bmatrix} \begin{Bmatrix} \varepsilon_n \\ \varepsilon_s \\ \varepsilon_t \end{Bmatrix} \tag{5}$$

The terms $t_n$, $t_s$, and $t_t$ denote the nominal tractions in the normal direction and two local shear directions, respectively. Correspondingly, $\varepsilon_n$, $\varepsilon_s$, and $\varepsilon_t$ represent the nominal strains in these directions. $E_{nn}$ refers to the tensile stiffness, while $E_{ss}$ and $E_{tt}$ are the shear moduli.

## Measurement of the electromagnetic performance of the textiles

To investigate the electromagnetic performance of the smart textiles featuring the spiral inductor pattern, MCP fiber antenna, and transmission lines, we employed a vector network analyzer (VNA, ROHDE&SCHWARS, ZNH18) covering the frequency of 30 kHz to 18 GHz was used in an anechoic chamber. Prior to testing, we configured the VNA's frequency range and calibrated it using the open-short-load calibration technique. To assess the $S_{11}$ of the smart textiles, we connected the port to a tuning series capacitor by the channel, targeting the frequency of 5–30 MHz. The impedance parameter $Z_{11}$ of a flower spiral inductor integrated into a smart MCP-based textile (with a diameter of 5 cm, a 1-mm gap, and 8 turns) was conducted in the frequency range employed for analysis was 5–30 MHz. Subsequently, the quality factor (Q) was calculated as follows:

$$Q = \frac{\text{Im}(Z_{11})}{\text{Re}(Z_{11})} \tag{6}$$

The washing procedure was carried out with the ISO 6330 standard, using domestic washing and drying. Smart textiles containing MCP fibers were placed in a home washing machine (SAMSUNG, WA85F5S3), along with the anti-bacterial detergent of Yuri-matic (Yuri Distribution Co. Pte. Ltd), to create a load of 2.3 kg. This was followed by a standard program of washing, rinsing, and spinning for 50 min at 40 °C for each cycle. After every 10 washing cycles, including rinsing and drying, the resistance was recorded to calculate the electrical conductivity, and the $S_{11}$ was tested. Furthermore, the MVTR of the smart textiles was measured in accordance with ASTM E96. Three specimens, each with a diameter of 68 mm, were cut from each fabric sample and placed on top of a standardized cup. The samples were then sealed with a rubber gasket to prevent vapor leakage from the edges. Each cup contained 20 ml of distilled water. The tests were conducted at a temperature of 38 °C and a relative humidity of 50% over 24 h. All three samples were tested for each test.

## Durability of the textiles based on MXene composite fibers

The durability of the smart textile was conducted by the flexible electronics tester (FT2000, Prtronic) from Shanghai Mifang Technology Co., Ltd. The bending and stretching tests were performed when the smart textile was fixed on the parallel-moving fixture. Additionally, the smart textile was mounted on the rotated fixture for the test at the rotating degree from 0° to 360°. During the testing period, the electrical resistance was measured every few cycles to calculate the

corresponding electrical conductivity of the smart textile. Moreover, the wireless communication based on the smart textile for the durability of the temperature was performed to test the sensor performance in the temperature range of −50 °C to 100 °C. To assess the durability of the smart wireless textile made from the body-coupled MCP-V fiber, the textile was placed 15 cm away from the transmitter (30 dBm, ~20 kHz).

## Mechanism of an alternating electromagnetic field generation by body-coupled MCP-V textiles

As the distance between the body skin and the MCP-V decreases, the electric field intensity increases accordingly. Once this field surpasses the air breakdown threshold, a localized plasma discharge may occur, triggering rapid collisions and sequential migration of ions and electrons among air molecules in the gap. This process induces a sharply varying electric displacement field. The equivalent LC circuit model for electromagnetic (EM) wave emission in the MCP-V. The dielectric layer (BaTiO$_3$ mixed PLA) contributes to the radial interface capacitance ($C_{ri}$), while the antenna core (MC) provides axial fiber inductance ($L_{MC}$) and resistance ($R_{MC}$). Additionally, parasitic capacitance ($C_p$) to ground arises from both the human body and the MCP-V fiber. When discharge occurs at the interface capacitance ($C_{ri}$), the dominant EM wave frequency ($f_d$) can be expressed as:

$$f_d = \frac{1}{2\pi\sqrt{L_{MC} \times C_t}} \tag{7}$$

where $C_t$ represents the system capacitance, calculated as the series combination of $C_{ri}$, $C_p$, and $C_{cs}$, where $C_{cs}$ represents the contact interface capacitance during local discharge at the surface. Therefore, the antenna core (MC fibers, shown in blue) facilitates the generation of an alternating electromagnetic field.

## Body-coupled MCP-V textiles remotely interact with a drone

(a) When the voltage at the skin-fiber interface capacitance exceeds the air breakdown threshold, high-frequency LC oscillation is triggered, emitting radio electromagnetic (EM) wave signals (Fig. 5a). The touch-induced electrical signal represents a unique form of wireless communication. The wireless signals generated by the MCP-V textile inherently carry sensing information, eliminating the need for additional encoding steps.

(b) Wireless signal reception and analysis: a coil antenna and a spectrum analyzer are used to capture and read the wireless signal within a specific frequency range from the MCP-V textile. The spectrum analyzer identifies the peak frequency of the wireless signal, which corresponds to the main frequency of the sensing signal. The process of acquiring and analyzing ambient radio frequency signals includes signal acquisition, band-pass filtering, Fourier transform (scanning every 15 ms), extension for signal amplification, and wireless signal conversion. Meanwhile, the buttons in the MCP-V textile were embroidered with MCP-V fibers of varying lengths. As a result, the buttons can emit radio electromagnetic (EM) wave signals with different intensities. Each signal intensity is mapped to a specific control function button.

(c) Signal acquisition for remote interaction with a drone: a modified drone (QQLRC Model M60 Pro+) equipped with a receiver and microprocessor is used to collect wireless signal data at 15 ms intervals. When the finger touches the MCP-V textile, it enables interaction with the drone to control its flight.

## Data availability

The data that support the findings of this study are available from the corresponding author upon request.

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

## Acknowledgements

This work was supported by the Singapore Ministry of Education Academic Research Fund Tier 2 (MOE-T2EP50123-0014 to L.W. and MOE-T2EP50223-0007 to L.W.), the Singapore Ministry of Education Academic Research Fund Tier 1 (RG62/22 to L.W. and RG72/24 to L.W.), A*STAR under MTC IRG (M24N7c0079 to L.W.), and the NTU-PSL Joint Lab collaboration to L.W. This work was partly supported by the National Science Fund for Distinguished Young Scholars (Grant No. 52125302 to Q.F.C.), the National Key Research and Development Program of China (Grant No. 2021YFA0715700 to Q.F.C.), the National Natural Science Foundation of China (Grant No. 52203078 to T.Z.Z., 22075009 to Q.F.C., and 52350012 to Q.F.C.), the Jiangsu Provincial Outstanding Youth Fund Project (SBK20250300200 to T.Z.Z.), the National Postdoctoral Program for Innovative Talents (Grant No. BX2021025 to T.Z.Z.), Postdoctoral Science Foundation (2021M690005 to T.Z.Z.), the New Cornerstone Science Foundation through the XPLORER PRIZE to Q.F.C., CAS Pioneer Hundred Talents Program B to T.Z.Z., and the Suzhou Key Laboratory of Bioinspired Interfacial Science (SZ2024004 to Q.F.C.). The authors thank Professor Lik-ho Tam, Dr. Yangzhe Yu, and Dr. Shixing Yuan for the help of the simulations, and Dr. Qi Zhu for the help of the TEM test. The authors thank the High-Performance Computing Platform at Beihang University, the assistance with nano-CT characterization at the Analysis and Testing Center of Beihang University, and the Physical and Chemical Analysis Center at Suzhou Institute for Advanced Research, University of Science and Technology of China for the support.

## Author contributions

T.Z.Z. and L.W. designed the research. T.Z.Z. conducted the fabrication, characterization, and data analyses with the help of J.Y., C.C., Q.H., W.L.L., and L.C. T.Z.Z. and C.W. co-performed finite element analysis. Y.Q.F. and D.L. assisted in the simulation of the fracture mechanism. T.Z.Z., Q.F.C., and L.W. co-wrote the manuscript. All the authors discussed the results and commented on the manuscript.

## Competing interests

The authors declare no competing interests.
