## [Transparent Peer Review file · Nature Communications]

Ultrastrong MXene composite fibers through static-dynamic densification for wireless electronic textiles

Corresponding Author: Professor Lei Wei

A version of this paper was originally rejected for publication by Nature Communications, however that decision was reconsidered after appeal by the authors.

Version 0:

Reviewer comments:

Reviewer #1

(Remarks to the Author)

This manuscript by Prof. Cheng and Prof. Wei presents a novel approach to filling the voids caused by the inherent transverse wrinkles in MXene nanosheets. Ultrastrong MXene fibers have been continuously fabricated on an industrial scale using a static filling method combined with dynamic thermal drawing to bridge MXene nanosheets. Notably, this breakthrough results in dense MXene fibers with a record tensile strength of ~941.5 MPa and an exceptional electrical conductivity. The enhanced interfacial interactions within the fibers, facilitated by hydrogen bonding, significantly improve stress transfer efficiency. This is attributed to the high alignment and dense structure of the fibers, ultimately boosting their tensile strength. Specifically, a long-range, battery-free wireless human health monitoring and communication system—integrated with these smart textiles—successfully tracks temperature, pulse pressure, and UV intensity while maintaining exceptional durability and stability during daily activities. Furthermore, a smart wireless textile, made from these ultrastrong fibers with body-coupled features, enables remote drone operation and facilitates assisted communication. These innovative applications advance the development of flexible electronics for MXene fibers, bringing them closer to practical implementation.

This manuscript is well-written, and the experimental data is well organized and robustly supports the conclusions. The general readers of Nature Communications will be interested in this new fabrication strategy and the outstanding properties of MXene fibers. Thus, this manuscript is strongly recommended for publication in Nature Communication after the following minor revision:

1. The number of figures in the main text should be adjusted to meet the journal's requirements.
2. The discussion on the applications of FMT fibers in Wi-Fi and Bluetooth is suggested to be toned down due to the length requirement of manuscript.
3. The format of references should be consistent with requirements of Nature Communications, please carefully double check main text.

Reviewer #2

(Remarks to the Author)

This manuscript proposes a static-dynamic densification strategy combining CNT filling and thermal drawing to fabricate MXene fibers with enhanced mechanical and electrical performance. The reported performance metrics are certainly promising. However, the manuscript would benefit from a more rigorous analysis to clearly differentiate this approach from existing literature. While the proposed strategy is technically sound, it appears conceptually similar to previously reported methods that utilize CNTs and polymer encapsulation to improve nanosheet alignment and fiber densification. Without a direct comparison or clearer discussion of how this method advances the state of the art, the novelty of the work remains uncertain. While the manuscript presents interesting results and a practical approach to enhancing MXene fiber properties, it

is not suitable for publication in its current form.

Detailed concerns and suggestions are listed below:

1. The authors do not report the shear rate actually applied during the fiber extrusion process. Given that rheological behavior is highly sensitive to shear rate, this omission weakens the relevance of the discussion on flow characteristics. The shear rate should be calculated based on the specific spinning parameters, such as nozzle diameter and extrusion speed. Furthermore, the power-law parameters flow behavior index and consistency index should be re-evaluated within this relevant shear rate range. Without this, the rheological analysis remains disconnected from the actual fiber formation process.
2. While the authors claim that $\sim 0.46 \mu\text{m}$ CNTs provide optimal alignment and porosity, they do not explore or comment on the performance impact of even shorter CNTs. A brief discussion is needed to justify this specific choice as optimal, rather than arbitrary.
3. The methodology for calculating electrical conductivity is not clearly described, particularly whether the PLA encapsulation layer is included in the cross-sectional area. As PLA is non-conductive, this could significantly affect the reported conductivity values. The authors should clarify the basis of their calculation.
4. The manuscript reports using different coagulation baths for PM and FM fibers without justification. Since coagulation conditions can significantly influence fiber morphology and structure, this inconsistency requires explanation.
5. The manuscript mentions that PLA was selected due to its ability to form hydrogen bonds with MXene nanosheets. While this is a valid starting point, the reasoning would benefit from further elaboration. Various other polymers such as PVA, PEO, and polyamides also form strong hydrogen bonds with 2D materials and have been used in similar contexts.
6. The SS curve in Fig. 3a shows that the FMT fiber can sustain strain levels up to approximately 23%, which is substantially higher than that of the FM fiber. However, it remains unclear whether the MXene component within the PLA matrix remains intact throughout this deformation range, or whether the increased strain is primarily due to the polymeric encapsulation. To clarify this point, it would be helpful to determine whether the MXene structure inside the PLA remains undamaged up to the point of failure. For instance, if the PLA's contribution is excluded, experimental evidence—such as in situ resistance measurements during tensile testing could support whether the MXene remains mechanically stable and conductive throughout the elongation process. Additionally, the fracture image in Figure 3e does not clearly indicate which component (MXene or PLA) fractures first, making it difficult to fully interpret the failure mechanism.
7. The authors should correct typing mistakes throughout the manuscript.
8. Several references do not match the corresponding content in the main text, and some table entries are inconsistent with the cited references.

Reviewer #3

(Remarks to the Author)

This manuscript reports the production of Ti3C2 MXene-based fibers with outstanding electrical conductivity and mechanical properties. This was achieved by a so-called static-dynamic densification process which reduces voids and allows for the production of fibers with low porosity. The manuscript presented detailed mechanistic insights and modelling using DFT and finite element analysis. Finally, the applications of MXene-based fibres for wireless communication, health monitoring, and body-coupled interaction textile have been demonstrated. While the manuscript presents excellent performance for MXene-based fibers with clear applications, due to the lack of in-depth analysis of MXene material and characterizations of the fibers, the reviewer has not been able to fully verify the stated claims. Additionally, the manuscript will benefit from a thorough restructuring as the current presentation consists of many repetitions, ambiguities, references to many figures outside of the main text (10 figures in Extended Data and 57 figures and 20 Tables in Supplementary Information). As such the reviewer does not find this manuscript to be suitable for publication in Nature Communications in its current form. The reviewer's specific comments are as follows:

1. The abstract claims "industrial-scale fabrication of ultrastrong MXene fibers". The reviewer is not convinced this has been demonstrated in the manuscript.
2. Given the use of CNT and PLA, the manuscript cannot call the fibers produced MXene fibers. Appropriate terms, such as composite or coaxial must be used to avoid confusion.
3. Strategies for filling the voids by using small-size fillers have been demonstrated before for graphene. As such the concept presented in this work is not novel as claimed.
4. Page 2, "MXene (Ti3C2Tx) nanosheets emerge as a promising class of 2D carbon-based compounds", the classification of MXene as carbon-based is confusing and inaccurate.
5. Page 3, the manuscript explains the presence of wrinkles and voids. There needs to be some explanation on why these are formed. Specifically, there is no mention of the coagulation process which plays an important role.
6. There are too many acronyms for MXene-based fibers, which can be very confusing. These include: PM, FM, and FMT and labels such as A, B, C, and D as well as I, II, III, IV, and V and the use of these do not appear to be logical, i.e., difficult to identify the samples based on the labels.
7. Some results sections begin with a mini literature review which is not appropriate for results. The introduction section must have already discussed the relevant literature, this suggests the need for a restructuring.
8. Page 4, the first paragraph in "Fabrication of ultrastrong MXene fibers" discusses the fiber production first after which MXene synthesis is presented in the following paragraph. This style of presentation is not only inappropriate but will be confusing to the readership of the paper.
9. Given the PLA is an insulating layer around FM, how is FMT fiber even conducting? How is the conductivity measured? Many applications require surface conductivity and having an insulating sheath will be undesirable, for instance for energy storage application, which is one of the main applications of MXene fibers.
10. Page 5, given MILD synthesis was used for MXene, how was an accordion-like MXene was obtained? The MILD process usually combines etching and delamination leaving behind isolated MXene flakes.

11. Based on the SAED presented in Supplementary Figure 2d inset, it is difficult to confirm the presence of MXene. The authors must present this appropriately with labels for diffractions and estimate d-spacing from the SAED to enable the verification of the results.
12. The XRD for MXene in Supplementary Figure 3 does not show (001) diffractions other than (002), noticeable in high-quality and highly delaminated MXene.
13. Page 5, while the manuscript uses “shear stress (η)” and presents a power-law model, Extended Data Figure 2a shows viscosity vs. shear rate behavior. This is confusing.
14. Page 5, “the formation of transverse wrinkles in MXene nanosheets during the wet spinning requires over 408 kPa of the stress”, how was this value estimated? Shear stress will depend on the shear rate and can be controlled by spinning parameters like flow rate. What flow rate is being used for spinning and what shear rate is achieved? How does this compare to the min shear stress required?
15. Page 6, “short CNT nanofibers featuring carboxylated groups, with a diameter of 5.6 ± 2.3 nm and an optimized length of ~ 0.46 μm (Supplementary Figs. 4 and 5), are considered as ideal filling blocks”, this claim is not supported. Why CNTs with these morphological specifications were ideal? The manuscript presents results for different lengths of CNTs which can be used to justify this but at this point in the manuscript this remains unsupported. This again confirms a restructuring will be required.
16. Page 6, Why is a low n suitable for spinning?
17. The authors must report the precise composition of the FMT fibers, i.e., wt% CNT and wt% PLA.
18. Page 6, FMT fibers had a “low density of ~ 1.97 g/cm³”. This is counter intuitive, if this fibre has a lower porosity and more packed morphology than the so-called FM fiber, then it must have a higher density. This is confusing on so many levels and highlights a more significant issue which invalidates the reported mechanical and electrical properties. The authors must clarify how the density was estimated and what diameter was used for each calculation.
19. Page 6, FMT fibers showed an “exceptional conductivity of $13,567.4$ S cm⁻¹”, how could the conductivity increase? What diameter was used for measuring the conductivity (see question 18)? Given the PLA is insulating, how was the conductivity estimated (see question 9)? For electrical conductivity, the authors must present I-V curves in the Supplementary Information.
20. Page 6, “as shown in Fig. 1f”, this is not shown as Fig. 1f is just a schematic.
21. Page 8, “including various colors”, how were various colors achieved?
22. Page 9, “designated as FM-III”, is this a separate sample with 3% more CNT compared to FM? What is CNT% in FM then? This goes back to the comment 6.
23. When reporting the mechanical properties, it is essential to report all key properties (Young's modulus, tensile strength, elongation at break, and toughness) and not just a selection of the best properties.
24. Page 11, why was a “denser cross-sections for the FM fiber with 3 wt% CNTs” obtained?
25. Page 11, a number of new labels appeared such as FM-180 and FM-380, which further adds to the confusion.
26. Page 11, “tensile force of ~ 1.36 N”, is this ultimate tensile force of the fibre?
27. Page 11-12, “their alignment with operating radio frequencies”, it is unclear what this means, how are they aligned with operating radio frequency?
28. Page 12, “draw-down ratio (τ) was 47”, given the low elongation at break of MXene fibres, how could they be drawn with such a high draw ratio?
29. Page 13, Extended Data Fig. 7e, f, Supplementary Fig. 33, how could MXene fiber in the core sustain such high strains of $>15\%$? Stress-strain shows PLA dominated the mechanical properties. Does MXene break at some low strains? This is confusing and can be misleading. The authors must provide sufficient evidence the MXene core can be stretched to such high strains.
30. Page 15, “This process effectively enhances stress transfer efficiency”, it is unclear what process this sentence is referring to.
31. Page 16, “CNTs were pulled out from the layers of MXene nanosheets”, contrary to the authors' claim, this suggests low interactions between MXene and CNT, so the interactions are not very strong after all. Additionally, the fact that the inner FM core in FMT is protruding shows weak interactions. This needs to be discussed.
32. Page 16, “can be an ideal candidate for smart electronic textiles”, what are the properties of ideal candidates? This is misleading as required properties will be different for different applications.
33. Page 17, “S-parameters” needs a definition.
34. Page 20, “under various walking speeds”, what is moving? Is power unit also moving? In cases it the sentences are ambiguous and require further details.
35. Page 20, “full battery-free wireless system”, the reviewer is not convinced this term is appropriate. This is confusing as the structure presented is just a coil and needs a power source anyway so the term battery-free is not suitable and is misleading.
36. Page 21, “monitor temperature (T), pulse pressure (P), and relative UV intensity (RI), as shown in Fig. 4e”, while Fig. 4e, show some changes, it is difficult to verify the accuracy and suitability of these measurements. For instance, the data presented for P appears to just show noise.
37. Page 21, what are the power, storage, and sensing units composed of? How does the power unit work?
38. Page 21, “overcoming the limitations of NFC, which typically operates within a range of less than 10 cm”, How was this limitation overcome?
39. Page 21, “had slight fluctuations in stress levels during these activities”, this cannot be clearly observed in Fig. 4i.
40. What is the advantage of the health monitoring system presented compared to the existing systems reported?
41. Page 22, “FMTBa fibers”, it is unclear what fiber this label represents.
42. Page 22, “the antenna core (FM fibers, blue) for generating an alternating electromagnetic field”, how does this generate alternating electromagnetic field?
43. The body-coupled sensing has already been reported (DOI: 10.1126/science.adk3755). What is the advantage of this work compared to the existing works?

44. For the drone demonstration, how do different buttons operate wirelessly? What is the innovation here compared to previous works (DOI: 10.1038/s41928-019-0257-7)?
45. Page 31, "Fiber area measurements were conducted using SEM, with the average value calculated from at least three samples for each fiber", what area was used for each fibre?
46. Page 34, the description provided for "Body-coupled FMT-C textiles remotely interact with a drone" is insufficient. For instance, "When the finger touches the FMT-C textile, it enables interaction with the drone to control its flight", how was this interaction facilitated? How could the drone know which button was pressed? Where is the power unit?
47. Supplementary Information Page 4, what is the centrifuge model used and what was the relative centrifugal force in each case?
48. Supplementary Information Page 4, "0.01 g of a dispersing agent was introduced", What was this dispersing agent?
49. Supplementary Information Page 4, provide details of the model, power used, pulse details for the ultrasonic cell.
50. Supplementary Information Page 4, "ratios of MXene nanosheets to short CNTs from 1 wt% to 5 wt%", this is confusing, these are not ratios. How was wt% calculated?
51. Why was the coagulation bath changed from absolute ethyl alcohol to ammonium chloride solution?
52. Supplementary Figure 6b, it appears the MXene/CNT dispersion looks like a paste. This is unusual as MXene at 40 mg/mL tends to flow relatively easily. The authors must report the composition of the sample shown appropriately.
53. Supplementary Figure 7, present viscosity vs. shear rate for pure MXene and MXene/CNT systems to allow for comparison. Also, why is G'/G'' decreasing with concentration? This is counter-intuitive and opposite the trends observed previously for MXene (DOI: 10.1021/acsnano.7b08889).
54. Supplementary Figure 51, for the EDS map of Ba, why is there a strong Ba signal in the core given the core must be FM fiber?
55. This manuscript would also benefit from a close editing. Some examples of language issues that are frequently seen throughout this paper are: "greater porosity with more voids" [page 5], "~3453.2 cm⁻¹" [page 8], "Simultaneously, in FMT fiber, the binding energy of Ti-O shifted up to 459.4 eV and 464.1 eV compared to that in FM fiber" [page 8], "scale-up FM fiber" [page 8], "increase the spinning nozzle" [page 11], "testified by in-situ XRD patterns heated from" [page 12], "This is followed by the further sliding between" [page 15], "FMT textile performed the conductivity" [page 17], "which the design contained a ground layer made of FMT" [page 19], "APP interfaces" [page 21], "due to the thermal generated" [page 21], "The hundreds of meters" [page 22], "rate of 0.3 mm min⁻¹" [page 31], "receiving & analysis" [34].

Version 1:

Reviewer comments:

Reviewer #1

(Remarks to the Author)

The authors have addressed all the related issues. Publication of this manuscript is recommended.

Reviewer #4

(Remarks to the Author)

The manuscript demonstrates a strong composite-fiber strategy (static filling + thermal drawing with PLA) and convincing battery-free NFC textile demos with clear scalability and durability. The authors have addressed most of Reviewer 3's questions satisfactorily.

However, one consistency issue remains: you compute tensile strength from the maximum load using the total composite cross-section (including PLA), whereas electrical conductivity is reported on a core-only basis. This mismatch may unintentionally overestimate performance. Given that the system is a composite fiber and the PLA encapsulation materially affects performance and demonstrations, the authors should report electrical conductivity on a composite-fiber basis—i.e., σ_{comp} using the total cross-section (core + PLA)—and avoid mixing core-only conductivity with composite-area mechanics in headline claims. This will prevent overstatement and remove ambiguity in MX/MC → MCP comparisons.

Point-by-Point Responses to Reviewers' Comments

We thank all the reviewers for their in-depth review of our manuscript and for enriching us with their valuable comments and suggestions, which have helped us further improve the quality of the manuscript. The reviewers' comments are in *blue font*, and the authors' responses are in **black font**. All the changes in the revised Manuscript and Supplementary Information are marked in **red font**.

Reviewer #1:

This manuscript by Prof. Cheng and Prof. Wei presents a novel approach to filling the voids caused by the inherent transverse wrinkles in MXene nanosheets. Ultrastrong MXene fibers have been continuously fabricated on an industrial scale using a static filling method combined with dynamic thermal drawing to bridge MXene nanosheets. Notably, this breakthrough results in dense MXene fibers with a record tensile strength of ~941.5 MPa and an exceptional electrical conductivity. The enhanced interfacial interactions within the fibers, facilitated by hydrogen bonding, significantly improve stress transfer efficiency. This is attributed to the high alignment and dense structure of the fibers, ultimately boosting their tensile strength. Specifically, a long-range, battery-free wireless human health monitoring and communication system—integrated with these smart textiles—successfully tracks temperature, pulse pressure, and UV intensity while maintaining exceptional durability and stability during daily activities. Furthermore, a smart wireless textile, made from these ultrastrong fibers with body-coupled features, enables remote drone operation and facilitates assisted communication. These innovative applications advance the development of flexible electronics for MXene fibers, bringing them closer to practical implementation.

This manuscript is well-written, and the experimental data is well organized and robustly supports the conclusions. The general readers of Nature Communications will be interested in this new fabrication strategy and the outstanding properties of MXene fibers. Thus, this manuscript is strongly recommended for publication in Nature Communication after the following minor revision:

Response:

We really appreciate your in-depth review of our manuscript.

1. The number of figures in the main text should be adjusted to meet the journal's requirements.

Response:

We thank the reviewer for your valuable suggestion. We have revised the number of figures to meet the journal's requirements.

2. The discussion on the applications of FMT fibers in Wi-Fi and Bluetooth is suggested to be toned down due to the length requirement of manuscript.

Response:

We thank the reviewer for your valuable suggestion. We have reduced the discussion on the applications of MCP (The abbreviation FMT has been changed to MCP) fibers in Wi-Fi and Bluetooth by removing Supplementary Figs. 42-44 in the revised manuscript.

3. *The format of references should be consistent with requirements of Nature Communications, please carefully double check main text.*

Response:

We thank the reviewer for your valuable comment. We have revised the format of references to meet the requirements of *Nature Communications*.

Reviewer #2:

This manuscript proposes a static-dynamic densification strategy combining CNT filling and thermal drawing to fabricate MXene fibers with enhanced mechanical and electrical performance. The reported performance metrics are certainly promising. However, the manuscript would benefit from a more rigorous analysis to clearly differentiate this approach from existing literature. While the proposed strategy is technically sound, it appears conceptually similar to previously reported methods that utilize CNTs and polymer encapsulation to improve nanosheet alignment and fiber densification. Without a direct comparison or clearer discussion of how this method advances the state of the art, the novelty of the work remains uncertain. While the manuscript presents interesting results and a practical approach to enhancing MXene fiber properties, it is not suitable for publication in its current form. Detailed concerns and suggestions are listed below:

Response:

We thank the reviewer for your comments. According to your comments and suggestions, we have re-organized the manuscript to emphasize the novelty and breakthroughs of our work. Additionally, we have highlighted the difference and high performance of our study in comparison with existing literature. Furthermore, we have significantly improved the overall quality of the manuscript by addressing all your comments. Here are the details.

1. Scientific and technological breakthroughs achieved in this work:

- **This work reveals voids caused by transverse wrinkles between MXene nanosheets due to the weak interactions and capillary contraction.** In order to solve this problem, this work combines **static filling method** using short carbon nanotubes (CNTs) and **dynamic thermal drawing** to utilize interfacial interactions, reducing porosity and improving nanosheet alignment, thereby increasing stress transfer efficiency and enhancing tensile strength and conductivity of the resulting fibers.
- The thermal drawing process not only enhances the mechanical properties of MXene composite fibers but also generates **an encapsulation layer** on their exterior, which **improves the durability of the fibers**, making them more suitable for practical applications, especially in electronic textiles.
- **A long-range, battery-free wireless human health monitoring and communication system** utilizing high mechanical strength and electrical conductivity has been successfully developed. This system effectively tracks temperature, pulse pressure, and UV intensity while demonstrating exceptional durability and stability during daily activities over a 12-hour period.
- We have further **developed a body-coupled MXene composite fiber by coating the fibers with functional layers.** The resulting smart wireless textile, composed of body-coupled ultrastrong MXene composite fibers, enables long-range, battery-free remote control and assisted communication through reliable electrical signals and high-luminance light.

2. Differences and high performance in comparison with the existing literature:

Various methods have been developed to fabricate MXene fibers using CNTs and polymer encapsulation. One strategy involves utilizing CNTs as structural building blocks and additives through scrolling and wet spinning methods (*Small* 2018, 4, 1802225; *Small* 2018, 14, 1801203; *Mater. Lett.*

2023, 336, 133891; *Carbon* 2022, 200, 38). However, due to **weak interfacial interactions and voids caused by inherent transverse wrinkles between the MXene nanosheets**, the resulting fibers exhibit low tensile strength, toughness, and electrical conductivity, which hinders their practical applications. Another strategy employs polymers for encapsulation, such as pure aramid nanofibers (ANFs) (*ACS Appl. Mater. Interfaces* 2021, 13, 41933; *Nano-Micro Lett.* 2022, 14, 111) and thermoplastic polyurethane (TPU) mixed with multi-walled CNTs (*Adv. Fiber Mater.* 2024, 6, 1657), typically via wet spinning. Although these studies have demonstrated improved nanosheet alignment and facilitated hydrogen bond between MXene and polymer, the resulting fibers still show relatively low tensile strength (~ 502.9 MPa) and electrical conductivity (~ 3000 S cm⁻¹). **These limitations are primarily attributed to voids caused by the intrinsic transverse wrinkles between MXene nanosheets.** For example, as shown in **Table R1**, the previously reported core-sheath CNTs and Ti₃C₂T_x (MXene) (CNT@MXene) fibers with coaxial structure were fabricated by utilizing the mixture of short multi-walled CNTs and TPU as the encapsulation layer and the mixture of MXene and TPU as the core through coaxial wet spinning. Although the CNT@MXene fiber with a diameter of ~ 760 μ m has been fabricated, the MXene fiber showed a low tensile strength of ~ 10.07 MPa and a low electrical conductivity of ~ 2.22 S cm⁻¹ for the inner MXene fiber. As a result, the textile based on the obtained CNT@MXene fiber performed an electromagnetic interference shielding effectiveness of ~ 23.4 dB.

In this work, we demonstrate a controllable method for the continuous kilometer-scale fabrication of **ultrastrong MXene composite fibers** by utilizing **static filling wet spinning and dynamic thermal drawing**. This process employs short carboxylated CNTs to fill the voids caused by inherent transverse wrinkles between MXene nanosheets and to bridge MXene nanosheets through hydrogen bond, followed by dynamic thermal drawing with a polylactic acid (PLA) encapsulation layer that further bridges the inner MXene nanosheets through hydrogen bond. As a result, the dense MXene composite fibers achieve a record tensile strength of ~ 941.5 MPa and an exceptional electrical conductivity of $\sim 12,836.4$ S cm⁻¹ for the inner MXene fiber. This static-dynamic densification strategy significantly reduces voids, achieving a low porosity of $\sim 4.2\%$ and enhancing the orientation factor of MXene nanosheets in the fibers to ~ 0.945 . Enhanced interfacial interactions within the fibers, facilitated by hydrogen bonds between CNTs and MXene nanosheets, as well as between PLA and MXene nanosheets, improve stress transfer efficiency due to the high alignment and dense structure, thus promoting tensile strength. The textiles made from the developed ultrastrong MXene composite fibers not only demonstrate high electromagnetic performance (S_{11} of -25.0 dB at the frequency band of 13.56 MHz) but also sustain mechanical durability. Moreover, a long-range, battery-free wireless human health monitoring and communication system, using smart textiles, is effectively capable of tracking temperature, pulse pressure, and UV intensity while maintaining superior durability and stability during daily activities over a 12-hour period. Additionally, a smart wireless textile, made by ultrastrong MXene composite fibers with body-coupled features, enables remote drone operation and supports assisted communication.

Table R1. Comparison between the existing literature and this work.

	Adv. Fiber Mater. 2024, 6, 1657	This work
Fiber	Core-sheath CNT@MXene fiber	MCP fiber
Spinning solution	CNT-TPU, MXene-TPU	MXene aqueous solution with short carboxylated CNTs
Additive	TPU	Short carboxylated CNTs
Prepared method	Coaxial wet spinning	Static filling wet spinning + dynamic thermal drawing
Interfacial interactions of inner MXene fibers	-	Hydrogen bond
Interfacial interactions between the encapsulation polymer layer and inner MXene fibers	-	Hydrogen bond
Encapsulation layer	CNT and TPU	PLA
Diameter	~760 μm	~158.6 μm
Orientation factor (f)	-	~0.945
Porosity (%)	-	~4.2
Tensile strength	~10.07 MPa	~941.5 MPa
Toughness	-	~147.9 MJ m ⁻³
Conductivity	~2.22 S cm ⁻¹ (inner MXene fiber)	~12,836.4 S cm ⁻¹ (inner MXene fiber)
Function	EMI SE: ~ 23.4 dB	Wireless electronic textiles: (1) High S ₁₁ of -25.0 dB at the frequency band of 13.56 MHz with high durability; (2) Enable remote drone operation and support assisted communication

1. The authors do not report the shear rate actually applied during the fiber extrusion process. Given that rheological behavior is highly sensitive to shear rate, this omission weakens the relevance of the discussion on flow characteristics. The shear rate should be calculated based on the specific spinning parameters, such as nozzle diameter and extrusion speed. Furthermore, the power-law parameters flow behavior index and consistency index should be re-evaluated within this relevant shear rate range. Without this, the rheological analysis remains disconnected from the actual fiber formation process?

Response:

We thank the reviewer for the comment. In this work, the flow rate (Q) was maintained at $2.08 \times 10^{-9} \text{ m}^3 \text{ s}^{-1}$ with the inner radius (r) of the nozzle of $1.9 \times 10^{-4} \text{ m}$ (190 μm). As a result, the actual shear rate ($\dot{\gamma}$) during wet spinning is $\sim 386 \text{ s}^{-1}$ according to the equation (R1) (ACS Cent. Sci. 2020, 6, 254).

$$\dot{\gamma} = \frac{4Q}{\pi r^3} \quad (\text{R1})$$

where Q is the flow rate ($\text{m}^3 \text{s}^{-1}$), and r is the inner radius of the nozzle (m).

In order to explain the rheological behavior during wet spinning, the viscosity of a pure MXene nanosheet spinning dispersion (at a concentration of 40 mg mL^{-1}) was re-measured as a function of shear rate, up to 400 s^{-1} . As shown in **Fig. R1a**, the viscosity-shear rate curve demonstrated the shear-thinning behavior, indicating that the solution was non-Newtonian. As the $\dot{\gamma}$ increased, the randomly distributed MXene nanosheets in spinning dispersion progressively aligned more closely with the axial direction of the shear stress. Moreover, this non-Newtonian solution was fitted by the power law model $\eta = k\dot{\gamma}^{n-1}$, where η is the viscosity, k is the flow consistency index, and n is the flow behavior index. For finite element analysis (FEA) simulations of the wet spinning process, the n was re-evaluated to be ~ 0.01 based on the updated experimental results. According to the shear stress calculated from the FEA simulation (**Fig. R1b**), the shear stress in the axial center of the spinning tube (Y-axis) reached up to $\sim 16.7 \text{ kPa}$ (**Fig. R1c**), while the shear stress increased sharply to $\sim 80 \text{ kPa}$ along the Z-axis from the center to the edges (**Fig. R2b**). However, the formation of transverse wrinkles in MXene nanosheets during the wet spinning requires over 408 kPa of the stress, as indicated by dynamic FEA calculations, to compact the nanosheets sufficiently and reduce voids (**Fig. R2c**). Therefore, the shear stress generated during the wet spinning assembly was insufficient to significantly reduce the voids caused by transverse wrinkles in the pure MXene fiber (The abbreviation PM has been changed to MX), resulting in high porosity. We have updated Fig. 1b and Supplementary Figs. 7, 8 and added more discussion in the section of “Methods” in the revised Manuscript and Supplementary Information.

Fig. R1. Finite element analysis (FEA) of the wet spinning process. **a**, Viscosity as a function of shear rate of pure MXene nanosheets spinning dispersion at a concentration of 40 mg mL^{-1} . According to the fitting curve of power law, the flow power-law index of n is ~ 0.01 less than 1, exhibiting shear-thinning for spinning dispersion. **b**, FEA model of shear stress distribution for pure MXene spinning dispersion during the wet spinning process. **c**, The shear stress along the Z-axis distance according to

the FEA simulation. The origin of the Z-axis ($Z=-0.19$) corresponds to the lower part of the spinning tube outlet, and the endpoint corresponds to the upper part ($Z=+0.19$) of the outlet.

Fig. R2. Fabrication flow to achieve ultrastrong MXene composite fibers. **a**, Illustration of the fabrication of MCP fiber by the combination of static filling method and dynamic thermal drawing with an in-situ generated encapsulation layer through the hydrogen bonds between CNTs and MXene nanosheets, and PLA and MXene nanosheets, while the MX fiber with lots of voids generated from transverse wrinkles during the wet spinning. **b**, The shear stress distribution in the axial center of the spinning tube along the Y-axis, according to FEA simulation. **c**, The required shear stress was to compact the transverse wrinkles nanosheets sufficiently to reduce voids, obtained from the dynamic FEA simulation. **d**, Photograph of the kilometer-scale MCP fiber, extending to a length of thousands of meters. **e**, Star plot of tensile strength, toughness, specific strength, Young's modulus, and electrical conductivity of MXene composite fibers, including MX, MC, MCP, and MCP-V. **f**, Illustration of the smart wireless textiles incorporating various spiral inductors textile based on MCP fiber, featuring sensing, and storage units, along with MCP-V textiles for remote control of the drone and assisted communication.

Revision in Manuscript:

Page 30, line 14 (“Simulation for the wet spinning process in Methods”):

The flow rate (Q) was maintained at $2.08 \times 10^{-9} \text{ m}^3 \text{ s}^{-1}$ with the inner radius (r) of the nozzle of $1.9 \times 10^{-4} \text{ m}$ ($190 \text{ }\mu\text{m}$). As a result, the actual $\dot{\gamma}$ during wet spinning is $\sim 386 \text{ s}^{-1}$ according to the equation (2) as following¹⁴.

$$\dot{\gamma} = \frac{4Q}{\pi r^3} \quad (2)$$

where Q is the flow rate ($\text{m}^3 \text{ s}^{-1}$), and r is the inner radius of the nozzle (m).

In order to explain the rheological behavior during wet spinning, the viscosity of a pure MXene nanosheet spinning dispersion (at a concentration of 40 mg mL^{-1}) was measured as a function of shear rate, up to 400 s^{-1} .

2. While the authors claim that $\sim 0.46 \text{ }\mu\text{m}$ CNTs provide optimal alignment and porosity, they do not explore or comment on the performance impact of even shorter CNTs. A brief discussion is needed to justify this specific choice as optimal, rather than arbitrary.

Response:

We thank the reviewer for your valuable comment. As shown in **Fig. R3**, we prepared the shorter CNTs with a length of $0.19 \pm 0.04 \text{ }\mu\text{m}$. These CNTs were used to fabricate the MXene-CNT fiber (re-designated as MC-0.19) at a fixed weight percentage of $\sim 1\%$. As the CNTs length decreased from $\sim 0.46 \text{ }\mu\text{m}$ to $\sim 0.19 \text{ }\mu\text{m}$, the porosity of the MC fibers increased from $\sim 15.7\%$ to $\sim 18.8\%$ (**Fig. R4**). Meanwhile, the orientation factor (f) of MC fibers decreased from ~ 0.848 to ~ 0.842 (**Fig. R5**). Although shorter CNTs filled voids and promoted the alignment of MXene nanosheets through hydrogen bond (compared to pure MXene fibers, MX), the interfacial interactions were weaker when filling the voids between the nanosheets due to their shorter length ($\sim 0.19 \text{ }\mu\text{m}$), as compared to longer CNTs ($\sim 0.46 \text{ }\mu\text{m}$) (**Fig. R6**). In contrast, longer CNTs with lengths from ~ 2.55 to $\sim 13.91 \text{ }\mu\text{m}$ tended to increase the porosity, as their lengths exceeded those of the voids. As a result, the MC fibers, using CNTs with an optimized length of $\sim 0.46 \text{ }\mu\text{m}$, exhibited a high tensile strength up to $284.3 \pm 4.7 \text{ MPa}$ and electrical conductivity of $9,597.3 \pm 40.7 \text{ S cm}^{-1}$, compared to $109.3 \pm 2.2 \text{ MPa}$ and $8,944.1 \pm 85.8 \text{ S cm}^{-1}$ for the MX fiber (**Figs. R7 and R8**). We have provided a discussion and updated Supplementary Figs. 14-19 in the revised Manuscript and Supplementary Information.

Fig. R3. Distribution of length for CNTs with different lengths according to the AFM images. **a**, The length of $0.19 \pm 0.04 \text{ }\mu\text{m}$. **b**, The length of $0.46 \pm 0.08 \text{ }\mu\text{m}$. **c**, The length of $2.55 \pm 0.48 \text{ }\mu\text{m}$. **d**, The

length of $7.98 \pm 0.79 \mu\text{m}$. **e**, The length of $13.91 \pm 1.62 \mu\text{m}$. **f-j**, Length distribution curve of CNTs ranging from $\sim 0.19 \mu\text{m}$ to $\sim 13.91 \mu\text{m}$.

Fig. R4. **a**, SAXS patterns of MC fibers fabricated with different lengths of CNTs from $\sim 0.19 \mu\text{m}$ to $\sim 13.91 \mu\text{m}$ at the weight percentage of 1%. **b**, XRD patterns of the fabricated fibers. **c**, The porosity and density of the fabricated MC fibers.

Fig. R5. **a**, WAXS patterns of MC fibers fabricated with different lengths of CNTs from $\sim 0.19 \mu\text{m}$ to $\sim 13.91 \mu\text{m}$ at the weight percentage of 1%. MC-13.91 fiber was fabricated from a length of $\sim 13.91 \mu\text{m}$. MC-7.98 fiber was fabricated from a length of $\sim 7.98 \mu\text{m}$. MC-2.55 fiber was fabricated from a length of $\sim 2.55 \mu\text{m}$. MC-0.46 fiber was fabricated from a length of $\sim 0.46 \mu\text{m}$. MC-0.19 fiber was fabricated from a length of $\sim 0.19 \mu\text{m}$. **b**, Plots of the azimuthal angle of the fabricated fibers. **c**, The f of the fabricated MC fibers with different lengths of CNTs.

Fig. R6. SEM images of the cross-sections of MC fibers fabricated with different lengths of CNTs from $\sim 0.46 \mu\text{m}$ to $\sim 13.91 \mu\text{m}$ at the weight percentage of 1%. **a**, MC-13.91 fiber fabricated from the length of $\sim 13.91 \mu\text{m}$. **b**, MC-7.98 fiber fabricated from the length of $\sim 7.98 \mu\text{m}$. **c**, MX fiber. **d**, MC-2.55 fiber fabricated from the length of $\sim 2.55 \mu\text{m}$. **e**, MC-0.46 fiber fabricated from the length of $\sim 0.46 \mu\text{m}$. **f**, MC-0.19 fiber fabricated from the length of $\sim 0.19 \mu\text{m}$.

Fig. R7. Stress-strain curves for MC fibers fabricated with different lengths of CNTs from $\sim 0.46 \mu\text{m}$ to $\sim 13.91 \mu\text{m}$ at the weight percentage of 1%. **a**, MC-13.91 fiber fabricated from the length of $\sim 13.91 \mu\text{m}$. **b**, MC-7.98 fiber fabricated from the length of $\sim 7.98 \mu\text{m}$. **c**, MX fiber. **d**, MC-2.55 fiber fabricated from the length of $\sim 2.55 \mu\text{m}$. **e**, MC-0.46 fiber fabricated from the length of $\sim 0.46 \mu\text{m}$. **f**, MC-0.19 fiber fabricated from the length of $\sim 0.19 \mu\text{m}$.

Fig. R8. **a**, Stress-strain curves of MC fibers fabricated with different lengths of CNTs from ~ 0.16 μm to ~ 13.91 μm at the weight percentage of 1%, in which MC-13.91 fiber was fabricated from CNTs with the length of ~ 13.91 μm , MC-7.98 fiber was fabricated from CNTs with the length of ~ 7.98 μm , MC-2.55 fiber was fabricated from CNTs with the length of ~ 2.55 μm , MC-0.46 fiber was fabricated from CNTs with the length of ~ 0.46 μm , and MC-0.19 fiber was fabricated from CNTs with the length of ~ 0.19 μm . **b**, Tensile strength and toughness of the fabricated MC fibers. **c**, Conductivities of the fabricated MC fibers.

Revision in Manuscript:

Page 11, line 1 (“Interfacial interactions and morphological characterization of MXene composite fibers”):

The length of CNTs is a key factor in this static filling method. CNTs of varying lengths, from ~ 0.19 μm to ~ 13.91 μm , were used as the filler to fabricate MC fibers with a fixed weight percentage of $\sim 1\%$ (Supplementary Fig. 14). As the CNTs length increased to ~ 0.46 μm , the porosity of the MC fibers decreased from $\sim 18.8\%$ to $\sim 15.8\%$, compared to the $\sim 22.7\%$ porosity of the MX fiber. Meanwhile, the orientation factor (f) of MC fibers improved from ~ 0.842 to ~ 0.848 , compared to that of MX with ~ 0.831 (Supplementary Figs. 15-17, and Supplementary Table 1). This improvement was attributed to the fact that shorter CNTs filled the voids more effectively and enhanced the alignment of MXene nanosheets through hydrogen bonds. However, due to their shorter length (~ 0.19 μm), the interfacial interactions were weaker when filling the voids between the nanosheets, as compared to longer CNTs (~ 0.46 μm). In contrast, longer CNTs tended to increase the porosity, as their lengths exceeded those of the voids. As a result, the MC fibers, using CNTs with an optimized length of ~ 0.46 μm , exhibited a high tensile strength up to 284.3 ± 4.7 MPa and electrical conductivity of $9,597.3 \pm 40.7$ S cm⁻¹, compared to 109.3 ± 2.2 MPa and $8,944.1 \pm 85.8$ S cm⁻¹ for MX fiber (Supplementary Figs. 18, 19, and Supplementary Tables 2, 3), thanks to the effective filling by the shorter CNTs³⁰.

3. The methodology for calculating electrical conductivity is not clearly described, particularly whether the PLA encapsulation layer is included in the cross-sectional area. As PLA is non-conductive, this could significantly affect the reported conductivity values. The authors should clarify the basis of their calculation.

Response:

We thank the reviewer for your comment. Since our electronic textile relies solely on the conductivity of the inner MXene fibers, the calculation of the MCP and MCP-V fibers' conductivity excludes the PLA area and outer polymer layers from the total cross-sectional area. The corresponding calculation equation is provided in equation (6) of the Supplementary Information. Additionally, we employed SEM-FIB to obtain the cross-sections of the MCP fibers and used software (Image-Pro Plus) to analyze and determine the cross-sectional area of inner MXene fibers.

4. The manuscript reports using different coagulation baths for PM and FM fibers without justification. Since coagulation conditions can significantly influence fiber morphology and structure, this inconsistency requires explanation.

Response:

We thank the reviewer for your comment. During the wet assembly, for example, wet spinning, the coagulation conditions can influence fiber morphology and structure. For example, ethyl alcohol coagulation can facilitate the fast coagulation, while the chitosan aqueous solution exhibits the slow coagulation (*ACS Cent. Sci.* 2020, 6, 254). However, two-dimensional (2D) nanosheets assembled via wet chemical methods inevitably undergo capillary contraction during the drying process (*Nat. Commun.* 2020, 11, 2645; *Science* 2024, 383, 771; *Carbon* 2014, 66, 84). This capillary-induced contraction leads to significant structural shrinkage, which in turn results in the formation of transverse wrinkles in the nanosheets when MXene fibers are fabricated in combination with a pre-stretching step during wet spinning (*ACS Cent. Sci.* 2020, 6, 254; *ACS Nano* 2015, 15, 3320). Therefore, lots of voids were generated due to transverse wrinkles between MXene nanosheets due to the capillary contraction during wet spinning regardless which coagulant was used, such as ethyl alcohol or ammonium chloride aqueous solution.

Moreover, we have fabricated the pure MXene (PM has been revised as MX) fiber in coagulant bath containing absolute ethyl alcohol. Although continuous extrusion was possible using coagulant bath containing absolute ethyl alcohol, the resulting fibers were too weak to handle after spinning with more voids and hard to obtain the mechanical properties, as in the reported work (*ACS Cent. Sci.* 2020, 6, 254). Meanwhile, a larger number of transverse wrinkles were also observed in the MX fibers with lots of voids, as shown in **Fig. R9**. Based on the experimental result, the inherent transverse wrinkles in MXene fibers were generated when pure MXene fibers were fabricated using coagulant bath, either ammonium chloride solution or absolute ethanol. In contrast, we can easily fabricate MXene fibers (FM has been revised as MC) through one-step static filling method through hydrogen bonds using short CNTs in a coagulant bath containing absolute ethyl alcohol. The obtained MC fibers exhibited high tensile strength. Therefore, we used different coagulation baths to fabricate MX and MC fibers.

Fig. R9. SEM image of MX fibers fabricated in coagulant bath containing ammonium chloride aqueous solution bath (a) and absolute ethyl alcohol bath (b). TEM image of MX fibers fabricated in coagulant bath containing ammonium chloride aqueous solution bath (c) and absolute ethyl alcohol bath (d).

5. The manuscript mentions that PLA was selected due to its ability to form hydrogen bonds with MXene nanosheets. While this is a valid starting point, the reasoning would benefit from further elaboration. Various other polymers such as PVA, PEO, and polyamides also form strong hydrogen bonds with 2D materials and have been used in similar contexts.

Response:

We thank the reviewer for your comment. Although other polymers such as polyvinyl alcohol (PVA), polyethylene oxide (PEO), and polyamides can also form hydrogen bonds with 2D materials such as MXene nanosheets (*J. Appl. Polym. Sci.* 2017, 134, 45295; *Adv. Funct. Mater.* 2021, 31, 2010944; *J. Power Sources* 2018, 396, 683), the obtained fibers exhibited the low tensile strength, limited toughness, and electrical conductivity using these polymers as the additives. These shortcomings are primarily attributed to voids generated by inherent transverse wrinkles between the MXene nanosheets during wet assembly. To address this issue, this work aims to fabricate MXene composite fibers via a static filling method incorporating short CNTs, followed by dynamic thermal drawing. This approach leverages interfacial interactions, such as hydrogen bonds between MXene and CNTs, and between PLA and MXene nanosheets, to minimize void formation and improve the overall mechanical and electrical performance of the fibers. Meanwhile, during the dynamic thermal drawing process, the drawing stresses generated from the PLA preform further reduce porosity and improve nanosheets alignment, thereby enhancing the tensile strength and electrical conductivity of the resulting MXene composite fibers. In addition to enhancing the mechanical and electrical properties, the thermal drawing process

forms an encapsulating layer (PLA) on the fiber surface, which enhances durability and makes the fibers more suitable for practical applications, particularly in electronic textiles. Therefore, PLA was selected as the optimal choice for this work.

6. The SS curve in Fig. 3a shows that the FMT fiber can sustain strain levels up to approximately 23%, which is substantially higher than that of the FM fiber. However, it remains unclear whether the MXene component within the PLA matrix remains intact throughout this deformation range, or whether the increased strain is primarily due to the polymeric encapsulation. To clarify this point, it would be helpful to determine whether the MXene structure inside the PLA remains undamaged up to the point of failure. For instance, if the PLA's contribution is excluded, experimental evidence—such as in situ resistance measurements during tensile testing could support whether the MXene remains mechanically stable and conductive throughout the elongation process. Additionally, the fracture image in Figure 3e does not clearly indicate which component (MXene or PLA) fractures first, making it difficult to fully interpret the failure mechanism.

Response:

We thank the reviewer for your comment. We have provided the real-time resistance-strain curves of the MX, MC, and MCP fibers, as shown in **Fig. R10**. The results demonstrate that the MCP fiber exhibits a ~2% change in resistance prior to the fracture during a single stretch intact throughout this deformation range, compared to ~69% for the MX fiber and ~26% for the MC fiber. The results indicate that the MXene component within the PLA matrix remains intact throughout this deformation range, and the MXene structure inside the PLA remains completely undamaged up to the point of failure. This enhanced performance is attributed to the dense structure of the MCP fiber with high tensile strength and toughness, achieved through a combination of static filling method and dynamic thermal drawing process. Moreover, the results indicate that MXene and PLA undergo co-fracture. We have added the real-time resistance-strain curves in Fig. 3c in the revised Manuscript.

Fig. R10. Real-time resistance-strain curves of MXene composite fibers.

Revision in Manuscript:

Page 13, line 26 (“Mechanical properties of MXene composite fibers”):

Furthermore, the real-time resistance-strain curves of the MX, MC, and MCP fibers, as shown in Fig. 3c. The MCP fiber exhibits a ~2% change in resistance prior to the fracture during a single stretch intact

throughout this deformation range, compared to ~69% for the MX fiber and ~26% for the MC fiber. The results indicate that the MXene component within the PLA matrix remains intact throughout this deformation range, and the MXene structure inside the PLA remains completely undamaged up to the point of failure, suggesting that MXene and PLA undergo co-fracture. Meanwhile, the cyclic loading behavior of the...

7. The authors should correct typing mistakes throughout the manuscript.

Response:

We thank the reviewer for your comment. We have carefully corrected typing mistakes throughout the manuscript.

8. Several references do not match the corresponding content in the main text, and some table entries are inconsistent with the cited references.

Response:

We thank the reviewer for your comment. We have revised the references to match the corresponding content in the main text and updated the table entries to ensure consistency with the cited references.

Reviewer #3:

This manuscript reports the production of Ti_3C_2 MXene-based fibers with outstanding electrical conductivity and mechanical properties. This was achieved by a so-called static-dynamic densification process which reduces voids and allows for the production of fibers with low porosity. The manuscript presented detailed mechanistic insights and modelling using DFT and finite element analysis. Finally, the applications of MXene-based fibres for wireless communication, health monitoring, and body-coupled interaction textile have been demonstrated. While the manuscript presents excellent performance for MXene-based fibers with clear applications, due to the lack of in-depth analysis of MXene material and characterizations of the fibers, the reviewer has not been able to fully verify the stated claims. Additionally, the manuscript will benefit from a thorough restructuring as the current presentation consists of many repetitions, ambiguities, references to many figures outside of the main text (10 figures in Extended Data and 57 figures and 20 Tables in Supplementary Information). As such the reviewer does not find this manuscript to be suitable for publication in Nature Communications in its current form. The reviewer's specific comments are as follows:

Response:

We thank the reviewer for your comments. According to your comments and suggestions, we have re-organized the manuscript to emphasize the novelty and breakthroughs of our work and reduced Supplementary Figs. 14, 42-44, including moving Extended Data figures to the Supplementary Information. Moreover, we have provided additional analyses of the MXene material (such as component content analysis, I-V curves, cross-sectional area of the fiber, etc) and characterizations of the fibers to strengthen the validity of our work. Additionally, we have improved the overall quality of the manuscript by addressing all your comments. Here are the details about **scientific and technological breakthroughs achieved in this work:**

- **This work reveals voids caused by transverse wrinkles between MXene nanosheets due to the weak interactions and capillary contraction.** In order to solve this problem, this work combines static filling method using short carbon nanotubes (CNTs) and dynamic thermal drawing to utilize interfacial interactions, reducing porosity and improving nanosheet alignment, thereby increasing stress transfer efficiency and enhancing tensile strength and conductivity of the resulting fibers.
- The thermal drawing process not only enhances the mechanical properties of MXene composite fibers but also generates **an encapsulation layer** on their exterior, which **improves the durability of the fibers**, making them more suitable for practical applications, especially in electronic textiles.
- **A long-range, battery-free wireless human health monitoring and communication system** utilizing high mechanical strength and electrical conductivity has been successfully developed. This system effectively tracks temperature, pulse pressure, and UV intensity while demonstrating exceptional durability and stability during daily activities over a 12-hour period.
- We have further **developed a body-coupled MXene composite fiber by coating the fibers with functional layers.** The resulting smart wireless textile, composed of body-coupled ultrastrong MXene composite fibers, enables long-range, battery-free remote control and assisted communication through reliable electrical signals and high-luminance light.

1. The abstract claims “industrial-scale fabrication of ultrastrong MXene fibers”. The reviewer is not convinced this has been demonstrated in the manuscript.

Response:

We thank the reviewer for your comment. We have revised “industrial-scale fabrication of ultrastrong MXene fibers” as “kilometer-scale fabrication of ultrastrong MXene fibers” in the revised Manuscript.

Revision in Manuscript:

Page 1, line 32 (“Abstract”):

The inherent transverse wrinkles and resultant voids between MXene ($\text{Ti}_3\text{C}_2\text{T}_x$) nanosheets, caused by weak interactions during the assembly process, present significant challenges in preserving their intrinsic high mechanical and electrical properties of macroscopic fibers. In this work, we demonstrate a controllable method for the continuous **kilometer-scale** fabrication of ultrastrong MXene **composite** fibers by utilizing a static filling technique with...

2. Given the use of CNT and PLA, the manuscript cannot call the fibers produced MXene fibers. Appropriate terms, such as composite or coaxial must be used to avoid confusion.

Response:

We thank the reviewer for your comment. We have revised the description from “MXene fibers” to “MXene composite fibers” in the revised Manuscript and Supplementary Information.

3. Strategies for filling the voids by using small-size fillers have been demonstrated before for graphene. As such the concept presented in this work is not novel as claimed.

Response:

We thank the reviewer for your comment. In previously reported works, small-size two-dimensional (2D) nanosheets were introduced into the interlayer to enhance the interfacial interaction and reduce porosity of graphene materials (*Science* 2015, 349, 1083; *Nat. Commun.* 2020, 11, 2077; *Proc. Natl. Acad. Sci. U.S.A.* 2020, 117, 8727). However, in our work, **lots of voids in the MXene fibers are caused by transverse wrinkles between MXene nanosheets during the wet spinning process, and once these wrinkles are formed, they cannot be eliminated later through interfacial interactions.** Meanwhile, small-sized 2D fillers are ineffective in eliminating the voids present in MXene fibers, compared with one-dimensional (1D) fillers.

In our work, in order to solve this problem, **we employ 1D short carboxylated CNTs to static fill the voids caused by inherent transverse wrinkles between MXene nanosheets and to bridge MXene nanosheets through hydrogen bond, followed by dynamic thermal drawing with polylactic acid (PLA) encapsulation layer that further bridges the inner MXene nanosheets through hydrogen bond.** Meanwhile, during the dynamic thermal drawing process, the drawing stresses generated from the PLA preform further reduce porosity and improve nanosheets alignment, thereby enhancing the tensile strength and electrical conductivity of the resulting MXene composite fibers. This static-dynamic densification strategy significantly reduces voids, achieving a low porosity of ~4.2% and

enhancing the orientation factor of MXene nanosheets to ~ 0.945 . Enhanced interfacial interactions within the fibers, facilitated by hydrogen bonds between CNTs and MXene nanosheets, as well as between PLA and MXene nanosheets, improve stress transfer efficiency due to the high alignment and dense structure, thus promoting tensile strength. As a result, the dense MXene-CNTs-PLA (designated as MCP) composite fiber achieves a record tensile strength of ~ 941.5 MPa and an exceptional electrical conductivity of $\sim 12,836.4$ S cm^{-1} for inner MXene-CNTs (designated as MC) fiber. The smart textiles made from the developed ultrastrong MCP composite fibers not only demonstrate high electromagnetic performance but also sustain mechanical durability. Moreover, a long-range, battery-free wireless human health monitoring and communication system, using smart textiles, is effectively capable of tracking temperature, pulse pressure, and UV intensity while maintaining superior durability and stability during daily activities over a 12-hour period. Additionally, a smart wireless textile, made by ultrastrong MXene composite fibers with body-coupled features, enables remote drone operation and supports assisted communication.

4. Page 2, “MXene ($\text{Ti}_3\text{C}_2\text{T}_x$) nanosheets emerge as a promising class of 2D carbon-based compounds”, the classification of MXene as carbon-based is confusing and inaccurate.

Response:

We thank the reviewer for your comment. We have revised “MXene ($\text{Ti}_3\text{C}_2\text{T}_x$) nanosheets emerge as a promising class of 2D carbon-based compounds” to be “MXene ($\text{Ti}_3\text{C}_2\text{T}_x$) nanosheets emerge as a promising class of 2D materials” in the revised Manuscript.

Revision in Manuscript:

Page 2, line 26 (“Introduction”):

Two-dimensional (2D) MXene ($\text{Ti}_3\text{C}_2\text{T}_x$) nanosheets emerge as a promising class of 2D materials for developing high-performance fibers, owing to their high tensile strength of ~ 17.3 GPa and electrical conductivity ($\sim 2.4 \times 10^4$ S cm^{-1})¹¹⁻¹³.

5. Page 3, the manuscript explains the presence of wrinkles and voids. There needs to be some explanation on why these are formed. Specifically, there is no mention of the coagulation process which plays an important role.

Response:

We thank the reviewer for your comment. Two-dimensional (2D) nanosheets assembled via wet chemical methods undergo capillary contraction during the drying process (*Nat. Commun.* 2020, 11, 2645; *Science* 2024, 383, 771; *Carbon* 2014, 66, 84). This capillary-induced contraction leads to significant structural shrinkage, which in turn results in the formation of transverse wrinkles in the nanosheets when MXene fibers are fabricated in combination with a pre-stretching step during wet spinning (*ACS Cent. Sci.* 2020, 6, 254; *ACS Nano* 2015, 15, 3320). As a result, lots of voids were generated in the obtained MXene fibers. Meanwhile, although MXene fibers have been fabricated using various coagulation processes, such as coagulation baths containing NH_4^+ , Mg^{2+} , or Ca^{2+} ions, these strategies primarily aim to promote the axial alignment of MXene nanosheets during wet spinning.

However, they do not address the formation of transverse wrinkles due to the capillary contraction, which occurs during the wet spinning process. We have provided more discussion about the presence of wrinkles and voids, as well as the coagulation process, in the revised Manuscript.

Revision in Manuscript:

Page 3, line 6 and 13 (“Introduction”):

Increasing efforts have been devoted to enhancing the performance of MXene fibers by promoting the axial alignment of MXene nanosheets, primarily using wet spinning¹⁸. Most studies frequently utilize cross-linkers, combined with the pre-stretching step, to effectively align the MXene nanosheets and eliminate wrinkles, thus improving the overall properties of fibers¹⁹. **Meanwhile, although MXene fibers have been fabricated using various coagulation processes with different coagulation baths^{14,18}, these strategies also primarily aim to promote the axial alignment of MXene nanosheets during wet spinning.** However, there remains a noticeable gap between the intrinsic mechanical and electrical properties of MXene nanosheets and the realized properties of macroscopic fibers, an area ripe for further exploration²⁰. **This discrepancy often stems from the loose assembly of MXene nanosheets in the transverse direction, leading to the formation of transverse wrinkles due to the weak interfacial interactions and the effect of capillary contraction^{21,22},** similar to those observed with graphene nanosheets²³.

6. There are too many acronyms for MXene-based fibers, which can be very confusing. These include: PM, FM, and FMT and labels such as A, B, C, and D as well as I, II, III, IV, and V and the use of these do not appear to be logical, i.e., difficult to identify the samples based on the labels.

Response:

We thank the reviewer for your comment. As suggested, we have re-defined the relevant abbreviations and provided additional details for labels in the revised Manuscript. Pure MXene fiber is re-defined as “MX”, MXene fibers filled with short CNTs (MXene-CNTs) are re-defined as “MC”, and MC fibers embedded in a PLA preform via thermal drawing (MXene-CNTs-PLA) are re-defined as “MCP”. Meanwhile, MC fibers were fabricated with different lengths of CNTs from ~0.19 μm to ~13.91 μm at a weight percentage of 1%. MC-13.91 fiber was fabricated from a length of ~13.91 μm , MC-7.98 fiber was fabricated from a length of ~7.98 μm , MC-2.55 fiber was fabricated from a length of ~2.55 μm , MC-0.46 fiber was fabricated from a length of ~0.46 μm , and MC-0.19 fiber was fabricated from a length of ~0.19 μm . Additionally, MC fibers were fabricated with different weight percentages of CNTs from 1% to 5% with a length of ~0.46 μm . MC-1% fiber was fabricated from a 1% weight percentage of CNTs, MC-2% fiber was fabricated from a 2% weight percentage of CNTs, MC-3% fiber was fabricated from a 3% weight percentage of CNTs, MC-4% fiber was fabricated from a 4% weight percentage of CNTs, and MC-5% fiber was fabricated from a 5% weight percentage of CNTs.

7. Some results sections begin with a mini literature review which is not appropriate for results. The introduction section must have already discussed the relevant literature, this suggests the need for a restructuring.

Response:

We thank the reviewer for your comment. We have revised and restructured the results section in the revised Manuscript.

8. Page 4, the first paragraph in “Fabrication of ultrastrong MXene fibers” discusses the fiber production first after which MXene synthesis is presented in the following paragraph. This style of presentation is not only inappropriate but will be confusing to the readership of the paper.

Response:

We thank the reviewer for your comment. We have adjusted the discussion on MXene synthesis to be presented before fiber production in the first paragraph of the “Fabrication of ultrastrong MXene fibers” section in the revised Manuscript.

Revision in Manuscript:

Page 4, line 18 (“Fabrication of ultrastrong MXene composite fibers”):

Etching transformed the initial Ti_3AlC_2 into accordion-like $\text{Ti}_3\text{C}_2\text{T}_x$, resulting in $\sim 11.1 \mu\text{m}$ lateral size and $\sim 1.5 \text{ nm}$ thick MXene nanosheets after vibration exfoliation, confirmed by the scanning electron microscope (SEM) and atomic force microscope (AFM) images (Supplementary Figs. 1 and 2). These nanosheets, showing defect-free hexagonal crystallinity, were further verified by high-resolution transmission electron microscopy (HR-TEM) images and selected area electron diffraction patterns (Supplementary Fig. 3), along with the X-ray diffraction (XRD) patterns with the (001) diffractions (Supplementary Fig. 4)^{14,27}. Due to weak interfacial interactions between MXene nanosheets in the transverse direction, significant transverse wrinkles occur, leading to increased voids in the resulting macroscopic pure MXene fibers (designated as MX) with high porosity, as illustrated in Supplementary Fig. 2. Unlike 2D nanosheet structures, the 1D nanofiber structure can effectively prevent the formation of transverse wrinkles under shear stress during wet spinning^{23,28}.

9. Given the PLA is an insulating layer around FM, how is FMT fiber even conducting? How is the conductivity measured? Many applications require surface conductivity and having an insulating sheath will be undesirable, for instance for energy storage application, which is one of the main applications of MXene fibers.

Response:

We thank the reviewer for your comment. Since our electronic textile relies solely on the conductivity of the inner MXene fibers, the calculation of the MCP and MCP-V fibers' conductivity excludes the PLA area and outer polymer layers from the total cross-sectional area. The corresponding calculation equation is provided in equation (6) of the Supplementary Information. Additionally, we employed SEM-FIB to obtain the cross-sections of the MCP fibers and used software (Image-Pro Plus) to analyze and determine the cross-sectional area of inner MXene fibers. Moreover, the combined static filling and dynamic thermal drawing process not only enhances the mechanical and electrical properties of the MXene composite fibers, but also forms a PLA encapsulation layer on their surface, which significantly improves their durability. These advantages fulfill the requirements for highly durable and functional

composite fibers suitable for practical applications in smart electronic textiles, as demonstrated in this work. Although energy storage applications require MXene fibers with surface electrical conductivity and no insulating sheath, the MXene fibers with an encapsulation layer fabricated in this work are intended for use in smart wireless electronic textiles. In the field of wearable electronic textiles, composite conductive fibers with encapsulation layers not only enable wireless transmission functionality but also ensure the durability of the electronic textiles for practical applications.

10. Page 5, given MILD synthesis was used for MXene, how was an accordion-like MXene was obtained? The MILD process usually combines etching and delamination leaving behind isolated MXene flakes.

Response:

We thank the reviewer for your comment. We have provided the fabricated details: The MXene nanosheet solutions were synthesized using the following procedure: Initially, 2.7 g of Ti_3AlC_2 powders were combined with a solution consisting of 5.7 g of LiF dissolved in 60 mL of 9 M HCl at room temperature. Subsequently, the mixture was stirred at 50 °C for 30 hours to ensure a complete reaction. The resulting accordion-like MXene product was obtained by thorough washing: three cycles with 9 M HCl solution and then ten cycles with deionized water, each cycle involving 5 minutes of centrifugation at 3500 rpm. The sediments obtained were dispersed into 150 mL of deionized water with continuous vibration for 15 minutes until the supernatant solution reached a pH of ~7. The solution was then centrifuged at 1500 rpm for 30 minutes to obtain the supernatant solution, followed by centrifugation at 3500 rpm for another 30 minutes to obtain the sediments. Finally, these sediments were dispersed in deionized water to prepare the MXene nanosheet solutions with different concentrations.

11. Based on the SAED presented in Supplementary Figure 2d inset, it is difficult to confirm the presence of MXene. The authors must present this appropriately with labels for diffractions and estimate d-spacing from the SAED to enable the verification of the results.

Response:

We thank the reviewer for your comment. We have provided labels for diffractions in the selected area electron diffraction pattern (SAED) in **Fig. R11**. The MXene monolayer was further characterized by transmission electron microscopy (TEM), revealing well-defined lattice fringes with a spacing of ~0.26 nm, corresponding to the (100) plane of $\text{Ti}_3\text{C}_2\text{T}_x$ (*Nat. Commun.* 2020, 11, 2825; *Nano-Micro Lett.* 2025, 17, 235), as shown in the HR-TEM images. The selected area electron diffraction (SAED) pattern further verified the hexagonal symmetry of the MXene structure. We have updated Supplementary Fig. 3d in the revised Supplementary Information.

Fig. R11. **a**, and **b**, SEM image and the size distribution for exfoliated MXene nanosheets with a lateral size of $\sim 11.1 \mu\text{m}$. **c**, TEM image and **d**, Corresponding HR-TEM image of exfoliated MXene nanosheets. The selected area electron diffraction pattern (inset in **d**) confirms the hexagonal single crystal structure without obvious defects.

12. The XRD for MXene in Supplementary Figure 3 does not show (001) diffractions other than (002), noticeable in high-quality and highly delaminated MXene.

Response:

We thank the reviewer for your comment. Due to the high intensity of (002) diffractions, the (001) diffractions have been weakened. As shown in **Fig. R12**, the resulting XRD pattern of MXene in this work also displays the (001) diffraction peaks, consistent with those reported in previous studies (*ACS Cent. Sci.* 2020, 6, 254). We have updated Supplementary Fig. 4 in the revised Supplementary Information.

Fig. R12. **a**, XRD patterns of Ti_3AlC_2 , $\text{Ti}_3\text{C}_2\text{T}_x$ (MXene), and CNTs. According to the XRD patterns, the absence of (104) and (105) peaks demonstrates a complete removal of the Al layer from the Ti_3AlC_2 , indicating the successful preparation of MXene nanosheets. **b**, XRD patterns of MXene show the (001) diffractions.

13. Page 5, while the manuscript uses “shear stress (η)” and presents a power-law model, Extended Data Figure 2a shows viscosity vs. shear rate behavior. This is confusing.

Response:

We thank the reviewer for your comment. Due to the weak interfacial interactions between MXene nanosheets in the transverse direction, significant wrinkles occur, resulting in increased voids and high porosity in the macroscopic MX fibers. To evaluate the shear stress during wet spinning, finite element analysis (FEA) was conducted based on the rheological behavior of the MXene spinning solution, determined by the viscosity-shear rate curve. Meanwhile, an appropriate rheological model of the spinning solution must be established for the FEA simulation. This requires viscosity vs. shear rate data, which can then be fitted to a corresponding model, such as the power-law model. We have revised “shear stress (η)” as “shear stress” in the revised Manuscript.

Revision in Manuscript:

Page 5, line 31 (“Fabrication of ultrastrong MXene composite fibers”):

the axial direction of the shear stress. Then, this non-Newtonian solution was fitted by the power law model $\eta=k\dot{\gamma}^{n-1}$ for the FEA simulation during the wet spinning²⁹, where the flow power-law index (n) is 0.01. According to the **shear stress** calculated from the FEA simulation, the shear stress in the axial center of the spinning tube (Y-axis) reached up to **~16.7 kPa**, while the shear stress increased sharply to ~80 kPa along the Z-axis from the center to the...

14. Page 5, “the formation of transverse wrinkles in MXene nanosheets during the wet spinning requires over 408 kPa of the stress”, how was this value estimated? Shear stress will depend on the shear rate and can be controlled by spinning parameters like flow rate. What flow rate is being used for spinning and what shear rate is achieved? How does this compare to the min shear stress required?

Response:

We thank the reviewer for the comment. In this work, the flow rate (Q) was maintained at $2.08 \times 10^{-9} \text{ m}^3 \text{ s}^{-1}$ with the inner radius (r) of the nozzle of $1.9 \times 10^{-4} \text{ m}$ (190 μm). As a result, the actual shear rate ($\dot{\gamma}$) during wet spinning is $\sim 386 \text{ s}^{-1}$ according to the equation (R1) (ACS Cent. Sci. 2020, 6, 254).

$$\dot{\gamma} = \frac{4Q}{\pi r^3} \quad (\text{R1})$$

where Q is the flow rate ($\text{m}^3 \text{ s}^{-1}$), and r is the inner radius of the nozzle (m).

In order to explain the rheological behavior during wet spinning, the viscosity of a pure MXene nanosheet spinning dispersion (at a concentration of 40 mg mL^{-1}) was re-measured as a function of shear rate, up to 400 s^{-1} . As shown in **Fig. R1a**, the viscosity-shear rate curve demonstrated the shear-thinning behavior, indicating that the solution was non-Newtonian. As the $\dot{\gamma}$ increased, the randomly distributed MXene nanosheets in spinning dispersion progressively aligned more closely with the axial direction of the shear stress. Moreover, this non-Newtonian solution was fitted by the power law model $\eta=k\dot{\gamma}^{n-1}$, where η is the viscosity, k is the flow consistency index, and n is the flow behavior index. For finite element analysis (FEA) simulations of the wet spinning process, the n was re-evaluated to be

~0.01 based on the updated experimental results. According to the shear stress calculated from the FEA simulation (Fig. R1b), the shear stress in the axial center of the spinning tube (Y-axis) reached up to ~16.7 kPa (Fig. R1c), while the shear stress increased sharply to ~80 kPa along the Z-axis from the center to the edges (Fig. R2b). However, the formation of transverse wrinkles in MXene nanosheets during the wet spinning requires over 408 kPa of the stress, as indicated by dynamic FEA calculations, to compact the nanosheets sufficiently and reduce voids (Fig. R2c). Therefore, the shear stress generated during the wet spinning assembly was insufficient to significantly reduce the voids caused by transverse wrinkles in the pure MXene fiber (The abbreviation PM has been changed to MX), resulting in high porosity. We have updated Fig. 1b and Supplementary Figs. 7, 8 and added more discussion in the section of “Methods” in the revised Manuscript and Supplementary Information.

Fig. R1. Finite element analysis (FEA) of the wet spinning process. **a**, Viscosity as a function of shear rate of pure MXene nanosheets spinning dispersion at a concentration of 40 mg mL^{-1} . According to the fitting curve of power law, the flow power-law index of n is ~ 0.01 less than 1, exhibiting shear-thinning for spinning dispersion. **b**, FEA model of shear stress distribution for pure MXene spinning dispersion during the wet spinning process. **c**, The shear stress along the Z-axis distance according to the FEA simulation. The origin of the Z-axis ($Z=0$) corresponds to the lower part of the spinning tube outlet, and the endpoint corresponds to the upper part ($Z=+0.19$) of the outlet.

Fig. R2. Fabrication flow to achieve ultrastrong MXene composite fibers. **a**, Illustration of the fabrication of MCP fiber by the combination of static filling method and dynamic thermal drawing with an in-situ generated encapsulation layer through the hydrogen bonds between CNTs and MXene nanosheets, and PLA and MXene nanosheets, while the MX fiber with lots of voids generated from transverse wrinkles during the wet spinning. **b**, The shear stress distribution in the axial center of the spinning tube along the Y-axis, according to FEA simulation. **c**, The required shear stress was to compact the transverse wrinkles nanosheets sufficiently to reduce voids, obtained from the dynamic FEA simulation. **d**, Photograph of the kilometer-scale MCP fiber, extending to a length of thousands of meters. **e**, Star plot of tensile strength, toughness, specific strength, Young's modulus, and electrical conductivity of MXene composite fibers, including MX, MC, MCP, and MCP-V. **f**, Illustration of the smart wireless textiles incorporating various spiral inductors textile based on MCP fiber, featuring sensing, and storage units, along with MCP-V textiles for remote control of the drone and assisted communication.

Revision in Manuscript:

Page 30, line 14 (“Simulation for the wet spinning process in Methods”):

The flow rate (Q) was maintained at $2.08 \times 10^{-9} \text{ m}^3 \text{ s}^{-1}$ with the inner radius (r) of the nozzle of $1.9 \times 10^{-4} \text{ m}$ ($190 \text{ }\mu\text{m}$). As a result, the actual $\dot{\gamma}$ during wet spinning is $\sim 386 \text{ s}^{-1}$ according to the equation (2) as following¹⁴.

$$\dot{\gamma} = \frac{4Q}{\pi r^3} \quad (2)$$

where Q is the flow rate ($\text{m}^3 \text{ s}^{-1}$), and r is the inner radius of the nozzle (m).

In order to explain the rheological behavior during wet spinning, the viscosity of a pure MXene nanosheet spinning dispersion (at a concentration of 40 mg mL^{-1}) was measured as a function of shear rate, up to 400 s^{-1} .

15. Page 6, “short CNT nanofibers featuring carboxylated groups, with a diameter of $5.6 \pm 2.3 \text{ nm}$ and an optimized length of $\sim 0.46 \text{ }\mu\text{m}$ (Supplementary Figs. 4 and 5), are considered as ideal filling blocks”, this claim is not supported. Why CNTs with these morphological specifications were ideal? The manuscript presents results for different lengths of CNTs which can be used to justify this but at this point in the manuscript this remains unsupported. This again confirms a restructuring will be required.

Response:

We thank the reviewer for your valuable comment. As shown in **Fig. R3**, we prepared the shorter CNTs with a length of $0.19 \pm 0.04 \text{ }\mu\text{m}$. These CNTs were used to fabricate the MXene-CNT fiber (re-designated as MC-0.19) at a fixed weight percentage of $\sim 1\%$. As the CNTs length decreased from $\sim 0.46 \text{ }\mu\text{m}$ to $\sim 0.19 \text{ }\mu\text{m}$, the porosity of the MC fibers increased from $\sim 15.7\%$ to $\sim 18.8\%$ (**Fig. R4**). Meanwhile, the orientation factor (f) of MC fibers decreased from ~ 0.848 to ~ 0.842 (**Fig. R5**). Although shorter CNTs filled voids and promoted the alignment of MXene nanosheets through hydrogen bond (compared to pure MXene fibers, MX), the interfacial interactions were weaker when filling the voids between the nanosheets due to their shorter length ($\sim 0.19 \text{ }\mu\text{m}$), as compared to longer CNTs ($\sim 0.46 \text{ }\mu\text{m}$) (**Fig. R6**). In contrast, longer CNTs with lengths from ~ 2.55 to $\sim 13.91 \text{ }\mu\text{m}$ tended to increase the porosity, as their lengths exceeded those of the voids. As a result, the MC fibers, using CNTs with an optimized length of $\sim 0.46 \text{ }\mu\text{m}$, exhibited a high tensile strength up to $284.3 \pm 4.7 \text{ MPa}$ and electrical conductivity of $9,597.3 \pm 40.7 \text{ S cm}^{-1}$, compared to $109.3 \pm 2.2 \text{ MPa}$ and $8,944.1 \pm 85.8 \text{ S cm}^{-1}$ for the MX fiber (**Fig. R7 and R8**). We have provided a discussion and updated Supplementary Figs. 14-19 in the revised Manuscript and Supplementary Information.

Fig. R3. Distribution of length for CNTs with different lengths according to the AFM images. **a**, The length of $0.19 \pm 0.04 \text{ }\mu\text{m}$. **b**, The length of $0.46 \pm 0.08 \text{ }\mu\text{m}$. **c**, The length of $2.55 \pm 0.48 \text{ }\mu\text{m}$. **d**, The

length of $7.98 \pm 0.79 \mu\text{m}$. **e**, The length of $13.91 \pm 1.62 \mu\text{m}$. **f-j**, Length distribution curve of CNTs ranging from $\sim 0.19 \mu\text{m}$ to $\sim 13.91 \mu\text{m}$.

Fig. R4. **a**, SAXS patterns of MC fibers fabricated with different lengths of CNTs from $\sim 0.19 \mu\text{m}$ to $\sim 13.91 \mu\text{m}$ at the weight percentage of 1%. **b**, XRD patterns of the fabricated fibers. **c**, The porosity and density of the fabricated MC fibers.

Fig. R5. **a**, WAXS patterns of MC fibers fabricated with different lengths of CNTs from $\sim 0.19 \mu\text{m}$ to $\sim 13.91 \mu\text{m}$ at the weight percentage of 1%. MC-13.91 fiber was fabricated from a length of $\sim 13.91 \mu\text{m}$. MC-7.98 fiber was fabricated from a length of $\sim 7.98 \mu\text{m}$. MC-2.55 fiber was fabricated from a length of $\sim 2.55 \mu\text{m}$. MC-0.46 fiber was fabricated from a length of $\sim 0.46 \mu\text{m}$. MC-0.19 fiber was fabricated from a length of $\sim 0.19 \mu\text{m}$. **b**, Plots of the azimuthal angle of the fabricated fibers. **c**, The f of the fabricated MC fibers with different lengths of CNTs.

Fig. R6. SEM images of the cross-sections of MC fibers fabricated with different lengths of CNTs from $\sim 0.46 \mu\text{m}$ to $\sim 13.91 \mu\text{m}$ at the weight percentage of 1%. **a**, MC-13.91 fiber fabricated from the length of $\sim 13.91 \mu\text{m}$. **b**, MC-7.98 fiber fabricated from the length of $\sim 7.98 \mu\text{m}$. **c**, MX fiber. **d**, MC-2.55 fiber fabricated from the length of $\sim 2.55 \mu\text{m}$. **e**, MC-0.46 fiber fabricated from the length of $\sim 0.46 \mu\text{m}$. **f**, MC-0.19 fiber fabricated from the length of $\sim 0.19 \mu\text{m}$.

Fig. R7. Stress-strain curves for MC fibers fabricated with different lengths of CNTs from $\sim 0.46 \mu\text{m}$ to $\sim 13.91 \mu\text{m}$ at the weight percentage of 1%. **a**, MC-13.91 fiber fabricated from the length of $\sim 13.91 \mu\text{m}$. **b**, MC-7.98 fiber fabricated from the length of $\sim 7.98 \mu\text{m}$. **c**, MX fiber. **d**, MC-2.55 fiber fabricated from the length of $\sim 2.55 \mu\text{m}$. **e**, MC-0.46 fiber fabricated from the length of $\sim 0.46 \mu\text{m}$. **f**, MC-0.19 fiber fabricated from the length of $\sim 0.19 \mu\text{m}$.

Fig. R8. **a**, Stress-strain curves of MC fibers fabricated with different lengths of CNTs from ~ 0.16 μm to ~ 13.91 μm at the weight percentage of 1%, in which MC-13.91 fiber was fabricated from CNTs with the length of ~ 13.91 μm , MC-7.98 fiber was fabricated from CNTs with the length of ~ 7.98 μm , MC-2.55 fiber was fabricated from CNTs with the length of ~ 2.55 μm , MC-0.46 fiber was fabricated from CNTs with the length of ~ 0.46 μm , and MC-0.19 fiber was fabricated from CNTs with the length of ~ 0.19 μm . **b**, Tensile strength and toughness of the fabricated MC fibers. **c**, Conductivities of the fabricated MC fibers.

Revision in Manuscript:

Page 11, line 1 (“Interfacial interactions and morphological characterization of MXene composite fibers”):

The length of CNTs is a key factor in this static filling method. CNTs of varying lengths, from ~ 0.19 μm to ~ 13.91 μm , were used as the filler to fabricate MC fibers with a fixed weight percentage of $\sim 1\%$ (Supplementary Fig. 14). As the CNTs length increased to ~ 0.46 μm , the porosity of the MC fibers decreased from $\sim 18.8\%$ to $\sim 15.8\%$, compared to the $\sim 22.7\%$ porosity of the MX fiber. Meanwhile, the orientation factor (f) of MC fibers improved from ~ 0.842 to ~ 0.848 , compared to that of MX with ~ 0.831 (Supplementary Figs. 15-17, and Supplementary Table 1). This improvement was attributed to the fact that shorter CNTs filled the voids more effectively and enhanced the alignment of MXene nanosheets through hydrogen bonds. However, due to their shorter length (~ 0.19 μm), the interfacial interactions were weaker when filling the voids between the nanosheets, as compared to longer CNTs (~ 0.46 μm). In contrast, longer CNTs tended to increase the porosity, as their lengths exceeded those of the voids. As a result, the MC fibers, using CNTs with an optimized length of ~ 0.46 μm , exhibited a high tensile strength up to 284.3 ± 4.7 MPa and electrical conductivity of $9,597.3 \pm 40.7$ S cm⁻¹, compared to 109.3 ± 2.2 MPa and $8,944.1 \pm 85.8$ S cm⁻¹ for MX fiber (Supplementary Figs. 18, 19, and Supplementary Tables 2, 3), thanks to the effective filling by the shorter CNTs³⁰.

16. Page 6, Why is a low n suitable for spinning?

Response:

We thank the reviewer for your comment. The non-Newtonian MXene-CNTs spinning solution exhibits a shear-thinning behaviour during fluidic flow, which can be characterized by a power law viscosity model: $\eta=k\dot{\gamma}^{n-1}$. As shear rate ($\dot{\gamma}$) increases, the randomly orientated nematic MXene phases become more aligned in the direction of the shear stress. Less physical interaction occurs, leading to a decrease of viscosity (η) and power law index (n , where $n<1$). The degree of shear thinning can be quantified by the power-law exponent ($n-1$), where a smaller power-law exponent reflects a stronger shear-thinning effect, which is favorable for fiber spinning.

17. The authors must report the precise composition of the FMT fibers, i.e., wt% CNT and wt.% PLA.

Response:

We thank the reviewer for your comment. To determine the PLA content, we first dissolved the outer layer of PLA of the MCP fiber (FMT fiber in the initial submission) using chloroform. The contents of PLA and MC were then quantified by gravimetric analysis using an analytical balance. Additionally, to evaluate the respective contents of CNTs and MXene in the inner MC of MCP fibers after thermal drawing, thermogravimetric analysis (TGA) was employed. The MCP-71 (MCP) fiber consisted of 59.6 wt% MXene, 2.2 wt% CNTs, and 38.2 wt% PLA (**Table R2**). We have provided the precise composition of the MCP fibers in Supplementary Table 10 in the revised Supplementary Information.

Table R2. The precise composition of the MCP fibers.

Samples	MXene (wt%)	CNTs (wt%)	PLA (wt%)
MCP-55	43.5	1.6	54.9
MCP-63	49.2	1.8	49.0
MCP-67	55.6	2.0	42.4
MCP-71	59.6	2.2	38.2

18. Page 6, FMT fibers had a “low density of ~1.97 g/cm3”. This is counter intuitive, if this fibre has a lower porosity and more packed morphology than the so-called FM fiber, then it must have a higher density. This is confusing on so many levels and highlights a more significant issue which invalidates the reported mechanical and electrical properties. The authors must clarify how the density was estimated and what diameter was used for each calculation.

Response:

We thank the reviewer for your comment. In this work, we have provided the density of inner MXene fiber and MCP fibers in **Table R3**. Due to the low density of the encapsulation PLA layer polymer (1.26 g cm^{-3}), the MCP (MCP-71) fiber has a density of $\sim 1.97 \text{ g cm}^{-3}$, lower than that of the inner MC fiber

(4.40 g cm⁻³). There is no contradiction between the compact inner MC fiber and the low density of MCP fibers. Since our electronic textile relies solely on the conductivity of the inner MXene fibers, the calculation of the MCP and MCP-V fibers' conductivity excludes the PLA area and outer polymer layers from the total cross-sectional area. The corresponding calculation equation is provided in equation (6) of the Supplementary Information. For the calculation of mechanical properties, the PLA of MCP and MCP-V fibers contains the PLA area and outer polymer layers from the total cross-sectional area. Additionally, we employed SEM-FIB to obtain the cross-sections of the MCP fibers and used software (Image-Pro Plus) to analyze and determine the cross-sectional area of inner MXene fibers. We also provide the diameters of fiber in **Table R4**. Meanwhile, all tables have been provided in the revised Supplementary Information.

Table R3. The density and porosity of the fabricated MCP fibers with increasing draw-down ratios.

Samples	d -spacing (Å)	Density (ρ) (Inner MC fiber) (g cm ⁻³)	Density (ρ) (MCP) (g cm ⁻³)	Porosity (Inner MC fiber) (%)
MC-3%	12.76	3.90 ± 0.05	-	6.2 ± 1.2
MCP-55	11.98	3.81 ± 0.1	1.43 ± 0.16	13.9 ± 2.3
MCP-63	11.80	4.10 ± 0.04	1.68 ± 0.10	8.8 ± 0.9
MCP-67	11.70	4.30 ± 0.05	1.88 ± 0.08	5.2 ± 1.1
MCP-71	11.55	4.40 ± 0.03	1.97 ± 0.11	4.2 ± 0.7

Table R4. The diameters of the obtained MCP and inner MC fibers fabricated with various draw-down ratios.

Samples	Outer diameter of MCP (μm)	Inner diameter of MC (μm)
MCP-55	188.1	101.9
MCP-63	176.6	101.6
MCP-67	165.5	100.7
MCP-71	158.6	100.1

19. Page 6, FMT fibers showed an “exceptional conductivity of 13,567.4 S cm⁻¹”, how could the conductivity increase? What diameter was used for measuring the conductivity (see question 18)? Given the PLA is insulating, how was the conductivity estimated (see question 9)? For electrical conductivity, the authors must present I-V curves in the Supplementary Information.

Response:

We thank the reviewer for your comment. Due to the static filling using CNTs and thermal drawing stresses, the MCP-V fibers had a high orientation factor of 0.951 and low porosity (~4.1%) after coating, resulting in a high electrical conductivity of 13,567.4 ± 210.5 S cm⁻¹. Moreover, since our electronic

textile relies solely on the conductivity of the inner MXene fibers, the calculation of the MCP and MCP-V fibers' conductivity excludes the PLA area and outer polymer layers from the total cross-sectional area. The corresponding calculation equation is provided in equation (6) of the Supplementary Information. Additionally, we employed SEM-FIB to obtain the cross-sections of the MCP fibers and used software (Image-Pro Plus) to analyze and determine the cross-sectional area of inner MXene fibers. The area of the inner MC of MCP and MCP-V fibers was used to calculate the conductivity as shown in **Table R5**. Moreover, as suggested, we have provided the current-voltage (I-V) curves of fibers, which were measured using a digital source meter (Keithley 2400) with a two-wire configuration (**Fig. R13**). The results indicated that the I and V exhibited a linear relationship.

Table R5. The cross-sectional area of the inner MC fibers and the electrical conductivity of the fabricated MCP and MCP-V fibers.

Samples	S (μm^2) Inner MC fiber	Conductivity (Inner MXene fiber) (S cm^{-1})
MX	5754.9 ± 73.9	$8,944.1 \pm 85.8$
MC-3%	5139.2 ± 104.6	$11,364.8 \pm 65.8$
MCP-55	5096.2 ± 69.9	$9,245.8 \pm 54.85$
MCP-63	4976.1 ± 86.5	$10,986.8 \pm 67.45$
MCP-67	4607.4 ± 124.6	$11,498.5 \pm 125.7$
MCP-71	4581.2 ± 62.5	$12,836.4 \pm 108.69$
MCP-V	4487.9 ± 143.5	$13,567.4 \pm 210.5$

Fig. R13. The current-voltage (I-V) curves of fibers measured using a digital source meter (Keithley 2400) with a two-wire configuration.

20. Page 6, “as shown in Fig. 1f”, this is not shown as Fig. 1f is just a schematic.

Response:

We thank the reviewer for your comment. We have revised the corresponding discussion from “Moreover, as shown in Fig. 1f, this advanced smart wireless textile based on the MCP-V fiber achieved remote drone control and supported assisted communication.” to “Moreover, the schematic showed that this advanced smart wireless textile based on the MCP-V fiber achieved remote drone control and supported assisted communication (Fig. 1f).”

Revision in Manuscript:

Page 6, line 25 (“Fabrication of ultrastrong MXene composite fibers”):

demonstrated the highest tensile strength of ~1,088.5 MPa, toughness of ~293.5 MJ m⁻³, and exceptional conductivity of 13,567.4 S cm⁻¹ (Fig. 1e). Moreover, the schematic showed that this advanced smart wireless textile based on the MCP-V fiber achieved remote drone control and supported assisted communication (Fig. 1f).

21. Page 8, “including various colors”, how were various colors achieved?

Response:

We thank the reviewer for your comment. The obtained MCP fibers were dyed in color baths of different hues at 70 °C, with a dyeing speed of 10 meters per minute. Afterward, the fibers were dried at room temperature for 24 hours to obtain MCP fibers in various colors. We have added the details in Supplementary Fig. 12 in the revised Supplementary Information.

22. Page 9, “designated as FM-III”, is this a separate sample with 3% more CNT compared to FM? What is CNT% in FM then? This goes back to the comment 6

Response:

We thank the reviewer for your comment. MC-3% is MC fiber with 3 wt% CNTs. The component content of MXene and CNTs in MC fibers was calculated according to the TGA curves (**Fig. R14**). M_{MXene} , M_{CNTs} , and M_{MC} are the fractions of weight loss for MXene, CNTs, and MC fibers fabricated with different weight percentages of CNTs. The weight content (W_{CNTs}) of CNTs in the MC fiber was calculated using equation (R2). The results are shown in **Table R6**, indicating that MC-3% contains an actual 3.2 wt% of CNTs. We have added Supplementary Fig. 21 and Supplementary Table 4 in the revised Supplementary Information.

$$W_{CNTs} = \frac{M_{MC} - M_{MXene}}{M_{CNTs} - M_{MXene}} \quad (R2)$$

Fig. R14. TGA curves of pure MXene, CNTs, MC-1%, MC-2%, MC-3%, MC-4%, and MC-5%. These results were obtained in a nitrogen atmosphere using a heating rate of 10 K min⁻¹.

Table R6. The weight percentages of CNTs and MXene in MC fibers calculated according to the TGA curves.

Samples	MXene (wt%)	CNTs (wt%) added in the experiment	CNTs (wt%) calculated according to the TGA
MX	100	0	0
MC-1%	98.9	1	1.1
MC-2%	97.7	2	2.3
MC-3%	96.8	3	3.2
MC-4%	95.6	4	4.4
MC-5%	94.3	5	5.6

23. *When reporting the mechanical properties, it is essential to report all key properties (Young's modulus, tensile strength, elongation at break, and toughness) and not just a selection of the best properties.*

Response:

We thank the reviewer for your comment. In addition to tensile strength, strain (elongation at break), and toughness, we have provided the Young's modulus of the obtained fibers, as shown in **Tables R7-R9**. The MCP (MCP-71) fiber exhibited a significantly higher Young's modulus (~61.4 GPa) compared to that of the MX fiber (~21.5 GPa). We have updated Supplementary Tables 2, 14, and 16 in the revised Supplementary Information.

Table R7. The tensile strength, Young's modulus, strain, and toughness of the fabricated MX fibers with different lengths of CNTs.

Samples	S (μm^2) Whole fiber	Tensile strength (MPa)	Young's modulus (GPa)	Strain (%)	Toughness (MJ m ⁻³)
MC-13.91	5749.7 \pm 112.5	50.4 \pm 1.0	14.9 \pm 0.3	0.50 \pm 0.01	0.14 \pm 0.01
MC-7.98	5377.3 \pm 99.5	93.2 \pm 2.0	18.5 \pm 0.1	0.56 \pm 0.01	0.29 \pm 0.01
MX	5754.9 \pm 73.9	109.3 \pm 2.2	21.5 \pm 0.1	0.58 \pm 0.01	0.36 \pm 0.02
MC-2.55	5181.8 \pm 88.5	182.5 \pm 1.2	24.1 \pm 0.2	1.87 \pm 0.04	1.96 \pm 0.10
MC-0.46	5116.8 \pm 68.8	284.3 \pm 4.7	32.8 \pm 0.3	2.56 \pm 0.03	4.42 \pm 0.20
MC-0.19	5017.0 \pm 96.8	225.0 \pm 0.4	26.2 \pm 0.2	2.26 \pm 0.03	2.96 \pm 0.05

Table R8. The tensile strength, Young's modulus, and toughness of the fabricated MC fibers with different weight percentages of CNTs.

Samples	S (μm^2) Whole fiber	Tensile strength (MPa)	Young's modulus (GPa)	Strain (%)	Toughness (MJ m ⁻³)
MX	5754.9 \pm 73.9	109.3 \pm 2.2	21.5 \pm 0.1	0.58 \pm 0.01	0.36 \pm 0.02
MC-1%	5116.8 \pm 68.8	284.3 \pm 4.7	32.8 \pm 0.3	2.56 \pm 0.03	4.42 \pm 0.20
MC-2%	4968.3 \pm 97.7	569.6 \pm 3.9	37.2 \pm 1.0	5.54 \pm 0.09	17.38 \pm 1.29
MC-3%	5139.2 \pm 104.6	631.8 \pm 4.5	42.3 \pm 1.1	6.14 \pm 0.17	23.19 \pm 1.19
MC-4%	5160.8 \pm 84.6	502.8 \pm 3.5	34.9 \pm 1.4	4.45 \pm 0.03	11.54 \pm 0.43
MC-5%	5530.6 \pm 96.8	346.4 \pm 6.6	29.1 \pm 0.7	3.44 \pm 0.07	6.73 \pm 0.42

Table R9. The tensile strength, Young's modulus, and toughness of the fabricated MCP fibers with increasing draw-down ratios.

Samples	S (μm^2) Whole fiber	Tensile strength (MPa)	Young's modulus (GPa)	Strain (%)	Toughness (MJ m ⁻³)
PLA	28165.5 \pm 43.5	93.6 \pm 0.3	1.0 \pm 0.1	342.92 \pm 2.11	256.6 \pm 7.5
MCP-55	28015.8 \pm 98.5	418.3 \pm 5.9	21.9 \pm 1.9	49.03 \pm 0.86	180.4 \pm 6.9
MCP-63	21787.3 \pm 104.6	549.2 \pm 13.2	31.4 \pm 3.0	39.82 \pm 0.61	164.8 \pm 2.1
MCP-67	16481.2 \pm 76.6	683.4 \pm 9.0	48.6 \pm 3.0	28.75 \pm 0.73	153.7 \pm 4.9
MCP-71	14654.2 \pm 48.7	941.5 \pm 5.9	61.4 \pm 2.1	22.50 \pm 0.07	147.9 \pm 4.2

24. Page 11, why was a “denser cross-sections for the FM fiber with 3 wt% CNTs” obtained?

Response:

We thank the reviewer for your comment. Various weight percentages of short CNTs with an optimized length of $\sim 0.46 \mu\text{m}$, ranging from 1% to 5%, were used as the filler to fabricate the MC fibers (denoted as MC-1% to MC-5%). The porosity of MC fiber (MC-3%) reduced to $\sim 6.2\%$ when using 3 wt% CNTs, which was lower than that of MX fibers. Meanwhile, the MC fiber has a high density of 3.90 g cm^{-3} compared to 3.15 g cm^{-3} for the MX fiber. Moreover, the orientation factor (f) of the obtained MC fibers increased to ~ 0.915 because of the hydrogen bonds between CNTs and MXene nanosheets, surpassing that of the MX fiber. However, using more than 3 wt% CNTs as the fillers led to an increase in the porosity and a decrease in the alignment of the MXene nanosheets. The obtained SEM images further showed the denser cross-sections for the MC fiber with 3 wt% CNTs.

25. Page 11, a number of new labels appeared such as FM-180 and FM-380, which further adds to the confusion

Response:

We thank the reviewer for your comment. The new labels from MC-180 to MC-380 represent that MXene fibers were fabricated using various nozzles from $180 \mu\text{m}$ to $380 \mu\text{m}$ during wet spinning. MC fibers fabricated through different spinning nozzles with diameters of $180 \mu\text{m}$, $250 \mu\text{m}$, $380 \mu\text{m}$, and $500 \mu\text{m}$ are labeled as MC-180, MC-250, MC-380, and MC-500.

26. Page 11, “tensile force of $\sim 1.36 \text{ N}$ ”, is this ultimate tensile force of the fibre?

Response:

We thank the reviewer for your comment. The “tensile force of $\sim 1.36 \text{ N}$ ” is the ultimate tensile force of the fiber. We have revised the description in the revised Manuscript.

Revision in Manuscript:

Page 11, line 29, 31, and 34, and Page12, line 2 (“Interfacial interactions and morphological characterization of MXene composite fibers”):

further evidenced by the SEM images **with the denser cross-sections for the MC fiber with 3 wt% CNTs**, as illustrated in Supplementary Figs. 23 and 24. The dynamic thermal drawing process requires enough **ultimate tensile force** for MC fibers to fabricate the encapsulation layer. In this work, the MC-180 fiber fabricated with a $180 \mu\text{m}$ nozzle (Supplementary Figs. 25-29 and Supplementary Tables 6-8) shows only an **ultimate tensile force** of $\sim 1.36 \text{ N}$, which cannot meet the dynamic thermal drawing process. Thus, we further **increase the spinning nozzle diameter** from $180 \mu\text{m}$ to $380 \mu\text{m}$, and then the achieved MC-380 fiber with a diameter of $\sim 102.0 \mu\text{m}$ reaches the **ultimate tensile force** of $\sim 4.87 \text{ N}$, which is successfully fabricated into MCP fibers with a diameter of $\sim 158.6 \mu\text{m}$. These MCP fibers have an **ultimate tensile force** of $\sim 18.59 \text{ N}$, which effectively meet the load-bearing requirements for practical...

27. Page 11-12, “their alignment with operating radio frequencies”, it is unclear what this means, how are they aligned with operating radio frequency?

Response:

We thank the reviewer for your comment. The MC-180 fiber fabricated with a 180 μm nozzle shows an ultimate tensile force of ~ 1.36 N, which cannot meet the dynamic thermal drawing process. Thus, we further increase the spinning nozzle from 180 μm to 380 μm , and then the achieved MC-380 fiber with a diameter of ~ 102.0 μm reaches the ultimate tensile force of 4.87 ± 0.03 N, which is successfully fabricated into MCP fibers with a diameter of ~ 158.6 μm . These MCP fibers have an ultimate tensile force of ~ 18.59 N, which effectively meets the load-bearing requirements for practical applications, such as embroidery techniques. The MC-180 fabricated using a 180 μm nozzle and MC-250 fibers fabricated using a 250 μm nozzle have diameters smaller than ~ 102.0 μm , making it difficult to assemble them into spiral inductors for potential use in wireless systems operating at the 13.56 MHz frequency band. In contrast, the MC-380 fiber, which exhibits both high ultimate tensile force and tensile strength, also possesses a larger diameter that is more suitable for assembling spiral inductors for such wireless applications. Therefore, the MC-380 fiber’s compatibility with operating radio frequencies (e.g., 13.56 MHz) makes it well-suited for use in wearable wireless textiles. We have clarified such information in the revised Manuscript.

Revision in Manuscript:

Page 12, line 3 (“Interfacial interactions and morphological characterization of MXene composite fibers”):

fabricated into MCP fibers with a diameter of ~ 158.6 μm . These MCP fibers have an **ultimate tensile force** of ~ 18.59 N, which effectively meet the load-bearing requirements for practical applications, such as embroidery techniques. **Additionally, their alignment with operating radio frequencies (e.g., 13.56 MHz) due their suitable diameter makes them well-suited for wearable wireless textiles.**

28. Page 12, “draw-down ratio (τ) was 47”, given the low elongation at break of MXene fibres, how could they be drawn with such a high draw ratio?

Response:

We thank the reviewer for your comment. The draw-down ratio (τ) in this context can be defined as:

$$\tau = \sqrt{\frac{v_D}{v_F}} \quad (\text{R3})$$

where v_D is the drawing speed, and v_F is the feeding speed.

The draw-down ratio (τ) is not the same as elongation. During the thermal drawing process, it was the PLA preform, not the MXene fibers, that was drawn under different τ values. Notably, the PLA and the inner MC fiber did not come into contact until the draw-down ratio (τ) reached 47. Therefore, using a high τ ranging from 47 to 71 did not break the MXene composite fibers.

29. Page 13, Extended Data Fig. 7e, f, Supplementary Fig. 33, how could MXene fiber in the core sustain such high strains of >15%? Stress-strain shows PLA dominated the mechanical properties. Does MXene break at some low strains? This is confusing and can be misleading. The authors must provide sufficient evidence the MXene core can be stretched to such high strains

Response:

We thank the reviewer for your comment. We have provided the real-time resistance-strain curves of the MX, MC, and MCP fibers, as shown in **Fig. R10**. The results demonstrate that the MCP fiber exhibits a ~2% change in resistance prior to the fracture during a single stretch intact throughout this deformation range, compared to ~69% for the MX fiber and ~26% for the MC fiber. The results indicate that the MXene component within the PLA matrix remains intact throughout this deformation range, and the MXene structure inside the PLA remains undamaged up to the point of failure. This enhanced performance is attributed to the dense structure of the MCP fiber with high tensile strength and toughness. Moreover, the results indicate that MXene and PLA undergo co-fracture. We have added the real-time resistance-strain curves in Fig. 3c in the revised Manuscript.

Fig. R10. Real-time resistance-strain curves of MXene composite fibers.

Revision in Manuscript:

Page 13, line 26 (“Mechanical properties of MXene composite fibers”):

Furthermore, the real-time resistance-strain curves of the MX, MC, and MCP fibers, as shown in Fig. 3c. The MCP fiber exhibits a ~2% change in resistance prior to the fracture during a single stretch intact throughout this deformation range, compared to ~69% for the MX fiber and ~26% for the MC fiber. The results indicate that the MXene component within the PLA matrix remains intact throughout this deformation range and the MXene structure inside the PLA remains undamaged up to the point of failure, suggesting that MXene and PLA undergo co-fracture. Meanwhile, the cyclic loading behavior of the...

30. Page 15, “This process effectively enhances stress transfer efficiency”, it is unclear what process this sentence is referring to.

Response:

We thank the reviewer for your comment. Initially, during stretching, neighboring MXene nanosheets slide relative to each other because of the lowest E_a and ET values for MXene-MXene, alongside the straightening of molecular chains in the encapsulation PLA layer. As the tensile load increases, the hydrogen bonds between short CNTs and MXene nanosheets, exhibiting higher E_a and ET values for CNTs-MXene, gradually begin to break, leading to increased energy dissipation. This is followed by further sliding between MXene nanosheets and among CNTs. Furthermore, the fracture of the hydrogen bonds between the encapsulation PLA layer and MXene nanosheets with the highest E_a and ET for PLA-MXene initiates and progressively straightens along the loading direction. Upon continued stretching, these hydrogen bonds are completely disrupted. The pulled-out CNTs indicate that the interfacial interactions between the short CNTs and MXene have been broken, which enhances energy dissipation and contributes to improved tensile strength. This whole process of the break of interfacial bonds effectively enhances stress transfer efficiency. We have revised this session in the revised Manuscript.

Revision in Manuscript:**Page 16, line 1 (“Fracture mechanism of MXene composite fibers”):**

hydrogen bonds are completely disrupted. **The pulled-out CNTs indicate that the interfacial interactions between the short CNTs and MXene have been completely broken, which enhances energy dissipation and contributes to improved tensile strength. This whole process of the break of interfacial bonds effectively enhances stress transfer efficiency and substantially increases energy dissipation.** Moreover, as demonstrated in Supplementary Fig. 39, an FEA model for a segment of the MCP fiber was developed to further reveal the...

31. Page 16, “CNTs were pulled out from the layers of MXene nanosheets”, contrary to the authors’ claim, this suggests low interactions between MXene and CNT, so the interactions are not very strong after all. Additionally, the fact that the inner FM core in FMT is protruding shows weak interactions. This needs to be discussed.

Response:

We thank the reviewer for your comment. In this work, density functional theory (DFT) calculation was conducted to analyze the interfacial interactions within MCP fibers across four different interfaces, including MXene-MXene, CNTs-CNTs, CNTs-MXene, and PLA-MXene. The absorbed energy (E_a) and electron transfer number (ET) were evaluated to assess the bond strength at these interfaces. **Fig. R15d** illustrates a trend that bond strength increased with the rising E_a and ET between the surfaces. The PLA-MXene and CNTs-MXene interfaces exhibit relatively high E_a of -7.15 eV and -6.82 eV, respectively. These values are significantly higher than those for CNTs-CNTs (-1.46 eV) and MXene-MXene (-1.35 eV) interactions. Additionally, the ET indicated that the interaction strengths follow a similar sequence of MXene-MXene with a value of 0.026 e < CNTs-CNTs with that of 0.032 e < CNTs-MXene with that of 0.855 e for hydrogen bond < PLA-MXene with that of 1.037 e for hydrogen bond. The results indicate that the hydrogen bond interactions at the PLA-MXene and CNT-MXene interfaces are stronger than those at the MXene-MXene and CNTs-CNTs interfaces in our MCP fiber system.

Therefore, the possible fracture mechanism of the MCP fiber is depicted in **Fig. R15e**. Initially, during stretching, neighboring MXene nanosheets slide relative to each other because of the lowest E_a and ET values for MXene-MXene, alongside the straightening of molecular chains in the encapsulation PLA layer. As the tensile load increases, the hydrogen bonds between short CNTs and MXene nanosheets, exhibiting higher E_a and ET values for CNTs-MXene, gradually begin to break, leading to increased energy dissipation. This is followed by further sliding between MXene nanosheets and among CNTs. Furthermore, the fracture of the hydrogen bonds between the encapsulation PLA layer and MXene nanosheets with the highest E_a and ET for PLA-MXene initiates and progressively straightens along the loading direction. Upon continued stretching, these hydrogen bonds are completely disrupted. The pulled-out CNTs indicate that the interfacial interactions between the short CNTs and MXene have been broken, which enhances energy dissipation and contributes to improved tensile strength. We have revised the relevant discussion to make this session clearer in the revised Manuscript.

Fig. R15. Mechanical properties and fracture mechanism of MXene fiber. **a**, Stress-strain curves of the obtained MXene composite fibers. **b**, Comparison of tensile strength and conductivity between MCP fiber and the previously reported MXene-based fibers. **c**, Real-time resistance-strain curves of MXene composite fibers: MX fiber, MC fiber, and MCP fiber. **d**, DFT calculation of E_a and ET for four different interfaces, including MXene-MXene, CNTs-CNTs, CNTs-MXene. **e**, Fracture schematic diagram of MCP fiber, as well as sliding between MXene nanosheets and failure of the strong hydrogen bonds of CNTs-MXene and PLA-MXene interfaces. **f**, FEA of the local fracture process of the MCP fiber. **g**, Fracture morphology of the MCP fiber with the pull out of short CNTs and the crack of the encapsulated PLA layer.

Revision in Manuscript:

Page 15, line 17 (“Fracture mechanism of MXene composite fibers”):

significantly higher than those for CNTs-CNTs (-1.46 eV) and MXene-MXene (-1.35 eV) interactions. Additionally, the *ET* indicated the interaction strengths follow a similar sequence of MXene-MXene with a value of 0.026 e < CNTs-CNTs with that of 0.032 e < CNTs-MXene with that of 0.855 e for hydrogen bond < PLA-MXene with that of 1.037 e for hydrogen bond. The results indicate that the hydrogen bond interactions at the PLA-MXene and CNT-MXene interfaces are stronger than those at the MXene-MXene and CNTs-CNTs interfaces.

Page 16, line 1 (“Fracture mechanism of MXene composite fibers”):

hydrogen bonds are completely disrupted. The pulled-out CNTs indicate that the interfacial interactions between the short CNTs and MXene have been completely broken, which enhances energy dissipation and contributes to improved tensile strength. This whole process of the break of interfacial bonds effectively enhances stress transfer efficiency and substantially increases energy dissipation. Moreover, as demonstrated in Supplementary Fig. 43, an FEA model for a segment of the MCP fiber was developed to further reveal the...

32. Page 16, “can be an ideal candidate for smart electronic textiles”, what are the properties of ideal candidates? This is misleading as required properties will be different for different applications.

Response:

We thank the reviewer for your comment. Smart textiles made from fibers should offer robust mechanical and electrical performance to be effectively integrated to cover our bodies. These textiles are repeatedly subjected to physical stresses, such as moving and deformation, particularly under harsh conditions or when in contact with human skin. They are also susceptible to damage from regular cleaning and maintenance, such as washing and drying processes, which can lead to performance degradation and limit their long-term usability. Therefore, an effective solution could be the development of fibers with an encapsulation layer, which would further enhance the mechanical properties of the fibers, thereby enhancing the durability of smart textiles. Meanwhile, fibers with high mechanical properties and electrical conductivity are essential for smart electronic textiles. Therefore, our MCP fiber, featuring ultrahigh tensile strength and excellent electrical conductivity with in-situ generation of encapsulation PLA layer, is promising for use in smart electronic textiles.

Revision in Manuscript:

Page 17, line 5 (“Electromagnetic performance and mechanical durability of smart textiles”):

stresses of daily wear and laundering²⁵. Our MCP fiber, featuring ultrahigh tensile strength and excellent electrical conductivity with in-situ generation of encapsulation PLA layer, **is promising for use in smart electronic textiles**. Utilizing digitally controlled embroidery techniques, we precisely integrated computationally optimized electromagnetic patterns into...

33. Page 17, “S-parameters” needs a definition.

Response:

We thank the reviewer for your comment. The “S-parameters” refers to “scattering parameters”. We have provided the definition of “S-parameters” in the revised Manuscript.

Revision in Manuscript:

Page 17, line 19 (“Electromagnetic performance and mechanical durability of smart textiles”):

textile and E-Cu textile in Supplementary Fig. 48, respectively. Fig. 4b suggests that the MCP textile with the flower spiral inductor pattern offered the high electromagnetic performance of **the scattering parameters (S-parameters, S_{11})** values of -25.0 dB at the frequency band of 13.56 MHz, compared to -24.2 dB for Cu textile and -24.6 dB for E-Cu textile. Additionally, the MCP textiles achieved a quality factor (Q) higher than 10 across all textiles at the frequency band of 13.56 MHz (**Supplementary Fig. 49**), indicating their potential for wireless systems.

34. Page 20, “under various walking speeds”, what is moving? Is power unit also moving? In cases it the sentences are ambiguous and require further details.

Response:

We thank the reviewer for your comment. The test was conducted while a human equipped with the wireless power unit and receivers walked at various speeds. We have provided more details in the revised Manuscript.

Revision in Manuscript:

Page 20, line 7 (“Battery-free wireless human health monitoring system”):

A wireless power unit was first developed, as demonstrated in **Supplementary Fig. 53** and Supplementary Video 8, which effectively powered light-emitting devices by near field communication (NFC). We assessed the consistency of the wireless power transfer by measuring the rectified voltage at two types of receivers (R2.0 and R3.2), each operating at ~2.0 V and ~3.2 V, **when a human equipped with the wireless power unit and receivers at various walking speeds (v)**. The received voltage remained stable at around 2.0 V and 3.2 V for both types of receivers, with v ranging from 0 km h⁻¹ to 10.5 km h⁻¹. Furthermore, the unit...

35. Page 20, “full battery-free wireless system”, the reviewer is not convinced this term is appropriate. This is confusing as the structure presented is just a coil and needs a power source anyway so the term battery-free is not suitable and is misleading.

Response:

We thank the reviewer for your comment. We have revised the sentence “sufficient to support a full battery-free wireless system for daily activities” to be “sufficient to support a wireless system for daily activities” in the revised Manuscript.

Revision in Manuscript:

Page 20, line 12 (“Battery-free wireless human health monitoring system”):

maintained a consistent voltage output for all receivers during various motions over half-hour intervals, such as static, stooping, squatting, walking, and even running. This demonstrated the high stability of the wireless power unit, **sufficient to support a wireless system for daily activities**. Additionally, our wireless power unit consistently powers the LED light even when...

36. Page 21, “monitor temperature (T), pulse pressure (P), and relative UV intensity (RI), as shown in Fig. 4e”, while Fig. 4e, show some changes, it is difficult to verify the accuracy and suitability of these measurements. For instance, the data presented for P appears to just show noise.

Response:

We thank the reviewer for your comment. For the measurements of temperature (T), pulse pressure (P), and relative UV intensity (RI), the data were obtained in a stationary state through a spiral inductor pattern using the MCP fiber. Therefore, the data are relatively stable over a 10 or 30-second period. In particular, the sensors used here are all commercial products. The spiral inductor pattern using the MCP fiber functions as a signal transmission component rather than a sensor. In order to verify the accuracy and suitability of these measurements, we have tested the data of T, P, and RI using the commercial equipment. The results indicated that the data we obtained through testing are generally consistent with those measured by commercial equipment with the rigid printed circuit board (PCB) and sensors. Both systems demonstrate good stability and accuracy of temperature, beats per minute (BPM), and relative UV intensity (Fig. R16). Moreover, we have adjusted the type of commercial sensor (purchased from Nengsida Electronics Technology Co., Ltd.) and re-measured the pulse signal (Fig. R17f). These results demonstrate that the spiral inductor pattern constructed with MCP fiber enables stable and reliable signal transmission. We have updated Fig. 4f, g in the revised Manuscript.

Fig. R16 Comparison of data obtained from the sensing unit based on MCP textile and commercial equipment with the rigid printed circuit board (PCB) and sensors. a, Temperature (T). b, Pulse pressure (P). c, Relative UV intensity (RI). d-f, Calibration of the measured T, P, and RI with a reference of

commercial equipments. The commercial equipment was purchased from TEXAS INSTRUMENTS (TRF7970A EVM) for testing T and RI, while the commercial equipment for testing P was purchased from Nengsida Electronics Technology Co., Ltd (MPX01).

Fig. R17. Electromagnetic performance and mechanical durability of smart textiles based on MCP fibers. **a**, Smart textile embroidered with a flower spiral inductor pattern using the MCP fiber through embroidery machine method, featuring a gap space of 1 mm and 8 turns. **b**, The S-parameters (S_{11}) of the smart textiles, embroidered with flower patterns using various fibers, including Cu, E-Cu, and MCP fibers. **c**, The S_{11} and conductivity of the MCP fiber in smart textile with a flower pattern at the bending angles from 0° to 150° . **d**, The electrical conductivity retention of the smart textile during 90 washing cycles. **e**, Photography of the back side of the wireless sensing unit and its block diagram. **f**, The stability of the sensing unit, including monitoring human being's T, P, and RI over a 30-second period. **g**, Comparison of the P measured by using the sensing unit and commercial equipment. **h**, A long-range, battery-free wireless human health monitoring system embedded in a hoodie, which consisted of a wireless power unit, wireless sensing unit, and storage unit. **i**, Actual measurement data of the human health monitoring system to monitor the T, P, and RI over a 12-hour period between 8:00 AM and 8:00 PM with different activities.

Revision in Manuscript:

Page 20, line 21 (“Battery-free wireless human health monitoring system”):

These performance enables our wireless power unit to be a promising candidate for practical applications. Next, a wireless sensing unit based on a flexible PCB (F-PCB) with a diameter of 1.7 cm, was fixed in the center of the flower spiral inductor pattern, to monitor temperature (T), pulse pressure (P), and relative UV intensity (RI), as shown in Fig. 4e. Data from the sensing unit was transferred to a database in the data storage unit for storage and further analysis. **Figure 4f and Supplementary Video 10 demonstrate that this wireless sensing unit consistently and stably monitored the T, P, and RT over 30 seconds, indicating its practical application compared with commercial equipment (Fig. 4g).** Finally, users can access and evaluate these data through the wireless storage unit by NFC touching through...

37. Page 21, what are the power, storage, and sensing units composed of? How does the power unit work?

Response:

We thank the reviewer for your comment. The power unit consists of two spiral inductors assembled using MCP fibers. One functions as the receiver, integrated with a custom-designed flexible printed circuit board (F-PCB) that manages power and data, while the other serves as the transmitter, also equipped with a custom-designed F-PCB. Two electronic conductors made from MCP fiber connect the receiver and the transmitter for power and data transfer. The wireless sensing unit comprises a flower-shaped spiral inductor based on MCP fiber and a custom-designed F-PCB with a diameter of 1.7 cm. The F-PCB, which integrates temperature, pressure, and UV sensors, is positioned at the center of the spiral inductor to monitor temperature (T), pulse pressure (P), and relative UV intensity (RI). Moreover, the storage unit comprises a flower-shaped spiral inductor made from MCP fiber and a custom-designed F-PCB as a database, which stores the data collected by the sensing unit.

Additionally, as shown in **Fig. R17h** and **Fig. R18g, h**, when a smartphone equipped with near-field communication (NFC) capability is brought close to the smart textile (the power unit), power from the smartphone is transferred to the sensing unit via a near-field relay based on the smart textile designed for long-range NFC over a distance of more than 50 cm. At the terminal end of near-field relay, the relay will wireless powers and activates the sensing unit. Simultaneously, the collected data is transmitted back to the smartphone via the same near-field relay and subsequently forwarded to a data storage unit for further processing. Consequently, by keeping the smartphone near the smart textile throughout the day, the human health monitoring system can continuously track T, P, and RI under various activities, enabling all-day health monitoring. Therefore, Our system effectively operated at distances exceeding 50 cm, overcoming the limitations of NFC, which typically operates within a range of less than 10 cm. This smart textile operates without an external portable energy source, as the smartphone serves as both the power supply and data receiver through its NFC function (**Fig. R19**).

Fig. R17. Electromagnetic performance and mechanical durability of smart textiles based on MCP fibers. **a**, Smart textile embroidered with a flower spiral inductor pattern using the MCP fiber through embroidery machine method, featuring a gap space of 1 mm and 8 turns. **b**, The S-parameters (S_{11}) of the smart textiles, embroidered with flower patterns using various fibers, including Cu, E-Cu, and MCP fibers. **c**, The S_{11} and conductivity of the MCP fiber in smart textile with a flower pattern at the bending angles from 0° to 150° . **d**, The electrical conductivity retention of the smart textile during 90 washing cycles. **e**, Photography of the back side of the wireless sensing unit and its block diagram. **f**, The stability of the sensing unit, including monitoring human being's T, P, and RI over a 30-second period. **g**, Comparison of the P measured by using the sensing unit and commercial equipment. **h**, A long-range, battery-free wireless human health monitoring system embedded in a hoodie, which consisted of a wireless power unit, wireless sensing unit, and storage unit. **i**, Actual measurement data of the human health monitoring system to monitor the T, P, and RI over a 12-hour period between 8:00 AM and 8:00 PM with different activities.

Fig. R18. Long-range, battery-free wireless human health monitoring system. (a) The block diagram of the wireless power unit and photography of powering light-emitting devices. The received voltage at different v (b) from 0 km h⁻¹ to 10.5 km h⁻¹ and human being's condition (c) when output from the smart textile in wireless power unit. (d) Photograph of reading the database when transferred from the sensing unit with the block diagram of the storage unit. Wireless sensing unit assembled from MCP textile. (e) The retention of the storage unit performance when being immersed in water for 30 days. (f) The retention of the storage unit performance under a pressing load ranging from 0 kg to 120 kg. (g) The block diagram for long-range wireless communication. (h) Photography of a long-range, battery-free wireless human health monitoring system assembled from MCP textile through a relay for long-range (~50 cm) wireless communication.

Fig. R19. Schematic block diagram of the designed chip within the sensing unit.

38. Page 21, “overcoming the limitations of NFC, which typically operates within a range of less than 10 cm”, How was this limitation overcome?

Response:

We thank the reviewer for your comment. As shown in **Fig. R17h** and **Fig. R18g, h**, when a smartphone equipped with near-field communication (NFC) capability is brought close to the smart textile (the power unit), power from the smartphone is transferred to the sensing unit via a near-field relay based on the smart textile designed for long-range NFC over a distance of more than 50 cm. At the terminal end of near-field relay, the relay will wireless powers and activates the sensing unit. Simultaneously, the collected data is transmitted back to the smartphone via the same near-field relay and subsequently forwarded to a data storage unit for further processing. Consequently, by keeping the smartphone near the smart textile throughout the day, the human health monitoring system can continuously track T, P, and RI under various activities, enabling all-day health monitoring. Therefore, Our system effectively

operated at distances exceeding 50 cm, overcoming the limitations of NFC, which typically operates within a range of less than 10 cm.

Fig. R17. Electromagnetic performance and mechanical durability of smart textiles based on MCP fibers. **a**, Smart textile embroidered with a flower spiral inductor pattern using the MCP fiber through embroidery machine method, featuring a gap space of 1 mm and 8 turns. **b**, The S-parameters (S_{11}) of the smart textiles, embroidered with flower patterns using various fibers, including Cu, E-Cu, and MCP fibers. **c**, The S_{11} and conductivity of the MCP fiber in smart textile with a flower pattern at the bending angles from 0° to 150° . **d**, The electrical conductivity retention of the smart textile during 90 washing cycles. **e**, Photography of the back side of the wireless sensing unit and its block diagram. **f**, The stability of the sensing unit, including monitoring human being's T, P, and RI over a 30-second period. **g**, Comparison of the P measured by using the sensing unit and commercial equipment. **h**, A long-range, battery-free wireless human health monitoring system embedded in a hoodie, which consisted of a wireless power unit, wireless sensing unit, and storage unit. **i**, Actual measurement data of the human health monitoring system to monitor the T, P, and RI over a 12-hour period between 8:00 AM and 8:00 PM with different activities.

Fig. R18. Long-range, battery-free wireless human health monitoring system. (a) The block diagram of the wireless power unit and photography of powering light-emitting devices. The received voltage at different v (b) from 0 km h^{-1} to 10.5 km h^{-1} and human being's condition (c) when output from the smart textile in wireless power unit. (d) Photograph of reading the database when transferred from the sensing unit with the block diagram of the storage unit. Wireless sensing unit assembled from MCP textile. (e) The retention of the storage unit performance when being immersed in water for 30 days. (f) The retention of the storage unit performance under a pressing load ranging from 0 kg to 120 kg . (g) The block diagram for long-range wireless communication. (h) Photography of a long-range, battery-free wireless human health monitoring system assembled from MCP textile through a relay for long-range ($\sim 50 \text{ cm}$) wireless communication.

39. Page 21, “had slight fluctuations in stress levels during these activities”, this cannot be clearly observed in Fig. 4i.

Response:

We thank the reviewer for your comment. The spiral inductor pattern using the MCP fiber functions as a signal transmission component rather than a sensor. Moreover, we have adjusted the type of commercial sensor (purchased from Nengsida Electronics Technology Co., Ltd.) and re-measured the pulse signal (Fig. R17f). Moreover, we have updated Fig. R17i (inset for P data). Compared to the stationary state, dynamic movement, such as going to the office, to lunch, to exercise, and to dinner, causes slight fluctuations in the pulse signal. We have updated Fig. 4i in the revised Manuscript.

Fig. R17. Electromagnetic performance and mechanical durability of smart textiles based on MCP fibers. **a**, Smart textile embroidered with a flower spiral inductor pattern using the MCP fiber through embroidery machine method, featuring a gap space of 1 mm and 8 turns. **b**, The S-parameters (S_{11}) of the smart textiles, embroidered with flower patterns using various fibers, including Cu, E-Cu, and MCP fibers. **c**, The S_{11} and conductivity of the MCP fiber in smart textile with a flower pattern at the bending angles from 0° to 150° . **d**, The electrical conductivity retention of the smart textile during 90 washing cycles. **e**, Photography of the back side of the wireless sensing unit and its block diagram. **f**, The stability of the sensing unit, including monitoring human being's T, P, and RI over a 30-second period. **g**, Comparison of the P measured by using the sensing unit and commercial equipment. **h**, A long-range, battery-free wireless human health monitoring system embedded in a hoodie, which consisted of a wireless power unit, wireless sensing unit, and storage unit. **i**, Actual measurement data of the human health monitoring system to monitor the T, P, and RI over a 12-hour period between 8:00 AM and 8:00 PM with different activities.

Revision in Manuscript:

Page 21, line 12 (“Battery-free wireless human health monitoring system”):

various daily activities such as walking, working, dining out, and exercising between 8:00 AM and 8:00 PM over the 12-hour period, as shown in Fig. 4i. Notably, T increased during outdoor activities such as commuting to the office, going to lunch, exercising, and dining out, **due to the body heat generated by these activities. Meanwhile, the P had slight fluctuations in the pulse signal during these activities (inset for P data).** Additionally, RI rose during outdoor...

40. What is the advantage of the health monitoring system presented compared to the existing systems reported?

Response:

We thank the reviewer for your comment. The demonstrated approach in this work offers the following advantages for practical daily applications, particularly for wireless transmission within smart textiles.

- The MCP fibers are continuously fabricated on a large scale with high tensile strength and conductivity.
- The tensile strength of MCP fibers surpasses that of other materials used in NFC and wireless charging applications (**Table R10**), highlighting their advantages for wearable electronic textiles in these areas.
- MCP fibers with an encapsulation layer of PLA are better suited for wearable smart textiles, particularly when subjected to physical stresses such as movement and deformation, as well as under harsh conditions or when in direct contact with human skin.
- Thanks to their dense structure and high tensile strength, smart textiles based on MCP fibers demonstrate high durability through cycles of bending, twisting, stretching, and even poking, as well as washing.

The advantages of MCP fibers offered in electronic textiles are discussed and presented in **Table R10**. The MCP and MCP-V fibers exhibit high tensile strength compared to other materials, such as copper wire, encapsulation copper wire (E-Cu), and liquid metal fibers (*Nat Commun* 2022, 13, 2190), and conductive thread, making them stable and durable for wireless applications. Moreover, the MCP fiber in the smart textile exhibits a minimal conductivity relative change of ~0.3% and ~0.2% after 9×10^4 cycles of 180° bending, highlighting their durability in NFC and wireless charging applications compared to other materials.

Moreover, **Fig. R17C** demonstrates that the MCP textile maintained a stable S_{11} of -25 dB and high electrical conductivity across bending angles ranging from 0° to 150° , confirming its ability to function stably at various bending angles. In contrast, the Cu textile fractured after only 300 cycles of bending, while the E-Cu textile lasted 1,450 cycles. Moreover, after 3×10^4 twisting cycles of 360° , the MCP fiber in textile retained a high conductivity of $12,772.2 \text{ S cm}^{-1}$ (99.5% retention), while the Cu and E-Cu textiles broke after just 150 and 650 cycles (**Fig. R20G**), respectively. After 5×10^4 cycles of stretching under 20% strain illustrated in **Fig. R20H**, the MCP fiber in textile also maintained a conductivity of $\sim 12,797.9 \text{ S cm}^{-1}$ (99.7% retention), while the Cu textile fractured after 6×10^3 cycles and the E-Cu textile fractured after 1.5×10^4 cycles. Furthermore, as shown in **Fig. R17D**, after 90 cycles of washing, the MCP fiber in textile performed a conductivity of $\sim 12,810.7 \text{ S cm}^{-1}$ (99.8% retention). Additionally, our wireless power unit consistently powered the LED light even when subjected to bending angles from 0° to 150° , stretching strains ranging from 0% to 20%, and poking distances from 0 mm to 20 mm (**Fig. R21** and **Supplementary Video 9** in the revised Supplementary Materials). These results demonstrate the MCP fiber's high mechanical properties, attributed to the stability of hydrogen bonds, making it a promising candidate for practical applications. Additionally, the entire wireless system operated effectively across temperatures ranging from -50°C to 100°C , further testifying the suitability of this system for daily applications even under extreme environments. These additional results and discussions have been added in the revised Manuscript.

Fig. R17. Electromagnetic performance and mechanical durability of smart textiles based on MCP fibers. **a**, Smart textile embroidered with a flower spiral inductor pattern using the MCP fiber through embroidery machine method, featuring a gap space of 1 mm and 8 turns. **b**, The S-parameters (S_{11}) of the smart textiles, embroidered with flower patterns using various fibers, including Cu, E-Cu, and MCP fibers. **c**, The S_{11} and conductivity of the MCP fiber in smart textile with a flower pattern at the bending angles from 0° to 150° . **d**, The electrical conductivity retention of the smart textile during 90 washing cycles. **e**, Photography of the back side of the wireless sensing unit and its block diagram. **f**, The stability of the sensing unit, including monitoring human being's T, P, and RI over a 30-second period. **g**, Comparison of the P measured by using the sensing unit and commercial equipment. **h**, A long-range, battery-free wireless human health monitoring system embedded in a hoodie, which consisted of a wireless power unit, wireless sensing unit, and storage unit. **i**, Actual measurement data of the human health monitoring system to monitor the T, P, and RI over a 12-hour period between 8:00 AM and 8:00 PM with different activities.

Table R10. Comparison of the tensile strength and electrical resistance change under mechanical bending of MCP and MCP-V fibers with those of reported conductive materials for wireless applications.

Sample	Type	Dimension (μm)	Tensile strength (MPa)	Tested cycles	Resistance relative change (%)	Ref.
MCP-V fiber	Textile	~430.6 (diameter)	~1,088.5	~90,000	~0.2	This work
MCP fiber	Textile	~158.6 (diameter)	~941.5	~90,000	~0.3	This work
Copper wire	Textile	~100 (diameter)	~260.3	~300	break	This work
E-Cu	Textile	~130 (diameter)	~177.0	~1,450	break	This work
Liquid metal fiber	Textile	305 (diameter)	~28	24,000	<1	55
Liquid metal	PDMS	289 (thickness)	-	5,000	<1	R22
Liquid metal (LM)	PVA	150 (thickness)	-	10,000	<1	R23
Conductive thread	Textile	370 (diameter)	~60	10,000	<1	55
Elektrisola E-threads	Textile	120 (diameter)	-	300	-	R24
MXene	PET	5.5 (thickness)	-	5,000	<1	R25
MXene	PDMS/PI	~2 (thickness)	-	1,000	5	R26
Graphene	PEN	~50 (thickness)	-	11,000	<3	R27
Graphene	A4 Paper	~7.8 (thickness)	-	2,000	5	R28
Silver conductive ink	Textile	~75 (thickness)	-	1,000	5-10	R29
Copper paste	PET	~110 (thickness)	-	500	~760%	R30

Fig. R20. Mechanical durability of the smart textiles. (a) The embroidery machine method to fabricate the smart textiles using the MCP fiber. (b) MCP textiles with rectangle, circle, and flower patterns. S_{11} (c) and Q (d) of MCP textiles with rectangle, circle, and flower patterns. (e) The mechanical durability of Cu, E-Cu, and MCP textiles subjected to 180° bending cycles. (f) S_{11} curves during 9×10^4 cycles of bending for MCP textiles with flower patterns. The mechanical durability of Cu, E-Cu, and MCP textiles subjected to 360° twisting cycles (g) and stretching cycles at 20% strain (h). Stress-strain curves of Cu wire (i) and E-Cu wire (j). (k) Tensile strength and toughness of Cu wire and E-Cu wire.

Fig. R21. Durability of the wireless power unit in the smart textile utilizing MCP fibers: (a) Performance under bending angles ranging from 0° to 150°. (b) Stretching under strains varying from 0% to 20%. (c) Resistance to poking with distances from 0 mm to 20 mm.

41. Page 22, “FMT_{Ba} fibers”, it is unclear what fiber this label represents.

Response:

We thank the reviewer for your comment. We first fabricated MC fibers via static wet spinning using short CNTs. Subsequently, the MC fibers were fed into a hollow PLA-BaTiO₃ preform to form the MC-

PLA-BaTiO₃ (denoted as MCP_{Ba}), followed by the dynamic thermal drawing using optimized parameters. We have provided the details in the revised Manuscript.

Revision in Manuscript:

Page 21, line 20 (“Battery-free, body-coupled interaction textile”):

We applied the static filling method and dynamic thermal drawing to fabricate robust MCP-V fibers by coating MC-PLA-BaTiO₃ (denoted as MCP_{Ba}) fibers, fabricated from MC fibers and PLA-BaTiO₃ via thermal drawing (as illustrated in Supplementary Figs. 58 and 59).

42. Page 22, “the antenna core (FM fibers, blue) for generating an alternating electromagnetic field”, how does this generate alternating electromagnetic field?

Response:

We thank the reviewer for your comment. As the distance between the body skin and the MCP-V decreases, the electric field intensity increases accordingly. Once this field surpasses the air breakdown threshold, a localized plasma discharge may occur, triggering rapid collisions and sequential migration of ions and electrons among air molecules in the gap. This process induces a sharply varying electric displacement field. **Fig. R22a** illustrates the equivalent LC circuit model for electromagnetic (EM) wave emission in the MCP-V. The dielectric layer (BaTiO₃ mixed PLA) contributes to the radial interface capacitance (C_{ri}), while the antenna core (MC) provides axial fiber inductance (L_{MC}) and resistance (R_{MC}). Additionally, parasitic capacitance (C_p) to ground arises from both the human body and the MCP-V fiber. When discharge occurs at the interface capacitance (C_{ri}), the dominant EM wave frequency (f_d) can be expressed as:

$$f_d = \frac{1}{2\pi\sqrt{L_{MC} \times C_t}} \quad (R4)$$

where C_t represents the system capacitance, calculated as the series combination of C_{ri} , C_p , and C_{cs} , where C_{cs} represents the contact interface capacitance during local discharge at the surface. Therefore, the antenna core (MC fibers, shown in blue) facilitates the generation of an alternating electromagnetic field.

Additionally, plasma discharge across the air gap forms a temporary conductive path, lowering the interface impedance between air and the MCP-V. This reduction helps maintain a strong electric field across the luminance layer. When the interfacial electric field between the skin and the MCP-V surpasses the critical threshold of the luminance material, doped carriers are excited into the conduction band and subsequently recombine with holes, emitting visible light. Therefore, when the MCP-V comes into contact with human skin (e.g., a finger), an interfacial contact capacitance is established, which activates the optically responsive layer to emit visible light. If the electric field across this interface exceeds the air breakdown threshold, a localized plasma discharge occurs. This discharge generates an additional electric displacement field, disrupting the equilibrium of bound charge pairs and enabling the wireless transmission of electrical signals for tactile sensing. We have provided the details in the section of “Mechanism of an alternating electromagnetic field generation by body-coupled MCP-V textiles in Methods” in the revised Manuscript.

Fig. R22. Applications of the MCP-V textile. **a**, Schematic diagram illustrating the mechanism for wireless optical and electrical signal generation and transmission using a single MCP-V fiber with various colors under body-coupled EM fields. Where C_{cs} represents the contact interface capacitance during local discharge at the surface, C_{fi} is the radial interface capacitance, R_{MC} is the resistance, and L_{MC} is the axial fiber inductance provided by the MC fiber. Additionally, C_p is the parasitic capacitance with the ground formed by the human body and the MCP-V fiber. **b**, The dyed MCP-V fibers emit a red glow when exposed to body-coupled EM fields for the assisted communication. **c**, Embroider patterns on textiles using MCP-V fiber. **d**, Durability of the textile incorporating MCP-V fiber under bending, stretching, and poking conditions. **e**, A conceptual illustration shows how MCP-V textiles can be used to control a drone remotely. **f**, Photograph showcasing MCP-V textiles being used to control a drone remotely. **g**, Block diagram illustrating the interaction between MCP-V textiles and a drone, depicting signal input from the textile interface, processing of commands, wireless transmission to the drone, and execution of drone actions in response to the received signals. **h**, The flight path of the drone controlled by the MCP-V textile, accompanied by real-life images.

43. *The body-coupled sensing has already been reported (DOI: 10.1126/science.adk3755). What is the advantage of this work compared to the existing works?*

Response:

We thank the reviewer for your comment. To better reflect differences and high performance of MXene composite fibers achieved in this study, we have provided a detailed comparison of the performance with the existing literature (DOI: 10.1126/science.adk3755, Science 2024, 384, 74), as shown in **Table R11**. In the existing literature, a layer-by-layer coating process was employed to fabricate i-fibers by

incorporating BaTiO₃ and ZnS:Cu²⁺ phosphors into a double-network resin composed of vinyl silicone resin and acetoxy silicone resin. This mixture was then coated onto the surface of commercially available silver-plated conductive fibers. **The resulting fibers exhibited a low tensile strength of ~56.4 MPa.** The fabricated i-fibers are suitable for applications in wireless textile electronic systems and assisted optical communication.

In our work, we applied the static filling method and dynamic thermal drawing to fabricate robust MCP-V fibers by coating MCP_{Ba} fibers, fabricated from MC fibers and PLA-BaTiO₃ via thermal drawing. The resulting MCP-V fibers exhibited a three-layer structure, consisting of the MC fiber core, a PLA-BaTiO₃ layer, and an outer vinyl silicone-acetoxy silicone resin (VSASR)-ZnS-Cu²⁺ coating. **Consequently, the MCP-V fibers had a high orientation factor of 0.951 and low porosity (~4.1%) after coating, resulting in a tensile strength of 1,088.5 ± 10.1 MPa, which was much higher than i-fiber.** We also demonstrated toughness of 293.5 ± 10.5 MJ m⁻³, and excellent electrical conductivity of 13,567.4 ± 210.5 S cm⁻¹ for inner MC fiber. As a result, hundreds of meters of MCP-V fibers were successfully fabricated. Moreover, thanks to their high tensile strength, toughness, and functionality for body-coupled wireless optical and electrical signal transmission, our MCP-V fibers can be embroidered onto textiles to create various patterns and letters that emit light with different colors for the assisted communication. The textile based on fibers maintains a stable red light emission even under stretching, bending, and poking. After 9×10⁴ bending cycles at 180°, 5×10⁴ stretching cycles at 18% strain, and 2×10⁴ poking cycles at a distance of 30 mm, the MCP-V fiber textile retains its high-luminance light output and stable electrical signals (close to 100% retention rate). **However, the optical power density of the i-fiber reported exhibited a 3.8% decrease under a bending angle of 130°.** The high durability of textile based on MCP-fiber is attributed to the dense structure of fibers, fabricated by combining static filling and dynamic thermal drawing. Additionally, a smart wireless textile, made by MCP-V fibers with body-coupled features, enables remote drone operation and supports assisted communication.

Table R11. Comparison between Science 2024, 384, 74 and this work.

	Science 2024, 384, 74	This work
Fiber	i-fiber	MCP-V fiber
Inner core	Commercial silver plated conductive fiber	MC fiber
Additive	BaTiO ₃ and ZnS:Cu ²⁺ phosphors	Short CNTs, BaTiO ₃ , and ZnS-Cu ²⁺ phosphors
Prepared method	Layer-by-layer coating	Static filling wet spinning + dynamic thermal drawing, coating
Interfacial interactions of fibers	-	Hydrogen bond
Interfacial interactions between the encapsulation polymer layer and inner fibers	-	Hydrogen bond
Encapsulation layer	Vinyl silicon resin-acetoxy silicone resin	PLA, vinyl silicone-acetoxy silicone resin (VSASR)
Diameter	300-750 μm	~430.6 μm
Orientation factor (f)	-	~0.951
Porosity (%)	-	~4.1
Tensile strength	~56.4 MPa	~1088.5 MPa
Toughness	-	~293.5 MJ m ⁻³
Electrical conductivity	-	~13,567.4 S cm ⁻¹ (inner MXene fiber)
Durability	The optical power density decreased by 3.8% under a bending angle of 130°	Performance of wireless optical and electrical signals with 100% retention rate: (1) 9×10 ⁴ bending cycles at 180°; (2) 5×10 ⁴ stretching cycles at 18% strain; (3) 2×10 ⁴ poking cycles at a distance of 30 mm
Function	Wireless textile electronics system and assisted optical communication	Wireless electronic textiles: Enable remote drone operation and support assisted communication

44. *For the drone demonstration, how do different buttons operate wirelessly? What is the innovation here compared to previous works (DOI: 10.1038/s41928-019-0257-7)?*

Response:

We thank the reviewer for your comment. The buttons in the MCP-V textile were embroidered with MCP-V fibers of varying lengths. As a result, the buttons can emit radio electromagnetic (EM) wave signals with different intensities. Each signal intensity is mapped to a specific control function button.

(1) The body-coupled MCP-V textiles remotely interact with the drone as follows.

- a) When the voltage at the skin-fiber interface capacitance exceeds the air breakdown threshold, high-frequency LC oscillation is triggered, emitting radio electromagnetic (EM) wave signals

(**Fig. R23**). The touch-induced electrical signal represents a novel form of wireless communication. The wireless signals generated by the MCP-V textile inherently carry sensing information, eliminating the need for additional encoding steps.

- b) Wireless signal reception and analysis: A coil antenna and a spectrum analyzer are used to capture and read the wireless signal within a specific frequency range from the MCP-V textile. The spectrum analyzer identifies the peak frequency of the wireless signal, which corresponds to the main frequency of the sensing signal. The process of acquiring and analyzing ambient radio frequency signals includes signal acquisition, band-pass filtering, Fourier transform (scanning every 15 ms), extension for signal amplification, and wireless signal conversion (**Fig. R22a, g**).
- c) Signal acquisition for remote interaction with a drone: A modified drone (QQLRC Model M60 Pro+) equipped with a receiver and microprocessor is used to collect wireless signal data at 15 ms intervals. When the finger touches the MCP-V textile, it enables interaction with the drone to control its flight.

Fig. R23 a, Illustration of the working principle of body-coupled MCP-V fiber. Where E_{MC} represents the induced electric field within the MC fiber antenna, and E_s denotes the EM field acting on the MCP-V fiber's surface. **b**, The photograph shows the MCP-V fiber being wirelessly powered by the finger, while also capturing the surrounding electromagnetic energy. **c**, Directional diagrams illustrating wireless optical luminance. **d**, Directional diagrams depicting wireless electrical signals. Data are presented based on three repeated measurements (mean \pm SD).

Fig. R22. Applications of the MCP-V textile. **a**, Schematic diagram illustrating the mechanism for wireless optical and electrical signal generation and transmission using a single MCP-V fiber with various colors under body-coupled EM fields. Where C_{cs} represents the contact interface capacitance during local discharge at the surface, C_{ri} is the radial interface capacitance, R_{MC} is the resistance, and L_{MC} is the axial fiber inductance provided by the MC fiber. Additionally, C_p is the parasitic capacitance with the ground formed by the human body and the MCP-V fiber. **b**, The dyed MCP-V fibers emit a red glow when exposed to body-coupled EM fields for the assisted communication. **c**, Embroider patterns on textiles using MCP-V fiber. **d**, Durability of the textile incorporating MCP-V fiber under bending, stretching, and poking conditions. **e**, A conceptual illustration shows how MCP-V textiles can be used to control a drone remotely. **f**, Photograph showcasing MCP-V textiles being used to control a drone remotely. **g**, Block diagram illustrating the interaction between MCP-V textiles and a drone, depicting signal input from the textile interface, processing of commands, wireless transmission to the drone, and execution of drone actions in response to the received signals. **h**, The flight path of the drone controlled by the MCP-V textile, accompanied by real-life images.

(2) The innovation of the body-coupled MCP-V textiles.

To better reflect differences and high performance of MXene composite fibers achieved in this study, we have provided a detailed comparison of the performance with the existing literature (DOI: 10.1038/s41928-019-0257-7, *Nat. Electron.* 2019, 2, 243), as shown in **Table R12**. In the existing literature, the textile platform features a metamaterial structure composed of a planar comb-shaped pattern on the top layer, an intermediate fabric layer, and a bottom layer of unpatterned metallic

conductor. **While the textile enables body sensing and wireless touch detection, its durability has been minimally explored, aside from demonstrating stability during standing, walking, and running.** Additionally, when the index finger approaches the metamaterial textile, an immediate change occurred in both the relative signal strength indicator (RSSI) and the image displayed on the smartphone application **via the Bluetooth protocol with wireless power from an external power source.**

In our work, we applied the static filling method and dynamic thermal drawing to fabricate robust MCP-V fibers by coating MCP_{Ba} fibers, fabricated from MC fibers and PLA-BaTiO₃ via thermal drawing. The resulting MCP-V fibers exhibited a three-layer structure, consisting of the MC fiber core, a PLA-BaTiO₃ layer, and an outer vinyl silicone-acetoxy silicone resin (VSASR)-ZnS-Cu²⁺ coating. **Consequently, the MCP-V fibers had a high orientation factor of 0.951 and low porosity (~4.1%) after coating, resulting in a tensile strength of 1,088.5 ± 10.1 MPa, which was much higher than i-fiber.** We also demonstrated toughness of 293.5 ± 10.5 MJ m⁻³, and excellent electrical conductivity of 13,567.4 ± 210.5 S cm⁻¹ for inner MC fiber. As a result, hundreds of meters of MCP-V fibers were successfully fabricated. Moreover, thanks to their high tensile strength, toughness, and functionality for body-coupled wireless optical and electrical signal transmission, our MCP-V fibers can be embroidered onto textiles to create various patterns and letters that emit light with different colors for the assisted communication. The textile based on fibers maintains a stable red light emission even under stretching, bending, and poking. After 9×10⁴ bending cycles at 180°, 5×10⁴ stretching cycles at 18% strain, and 2×10⁴ poking cycles at a distance of 30 mm, the MCP-V fiber textile retains its high-luminance light output and stable electrical signals (close to 100% retention rate). **However, the optical power density of the i-fiber reported exhibited a 3.8% decrease under a bending angle of 130°.** The high durability of textile based on MCP-fiber is attributed to the dense structure of fibers, fabricated by combining static filling and dynamic thermal drawing. Additionally, a smart wireless textile, made by MCP-V fibers with body-coupled features, enables remote drone operation and supports assisted communication.

Table R12. Comparison between Nat. Electron. 2019, 2, 243 and this work.

	Nat. Electron. 2019, 2, 243	This work
Material	Metamaterial textiles and metallic conductor	MCP-V fiber
Inner core	-	MC fiber
Additive	-	Short CNTs, BaTiO ₃ , and ZnS-Cu ²⁺ phosphors
Prepared method	No mentioned	Static filling wet spinning + dynamic thermal drawing, coating
Interfacial interactions of fibers	-	Hydrogen bond
Interfacial interactions between the encapsulation polymer layer and inner fibers	-	Hydrogen bond
Encapsulation layer	-	PLA, vinyl silicone-acetoxo silicone resin (VSASR)
Diameter		~430.6 μm
Orientation factor (f)	-	~0.951
Porosity (%)	-	~4.1
Tensile strength	-	~1088.5 MPa
Toughness	-	~293.5 MJ m ⁻³
Electrical conductivity	2000-5000 S cm ⁻¹ (for conductive textiles)	~13,567.4 S cm ⁻¹ (inner MXene fiber)
Durability	Stand, walk, run, and keep stability	Performance of wireless optical and electrical signals with 100% retention rate: (1) 9×10 ⁴ bending cycles at 180°; (2) 5×10 ⁴ stretching cycles at 18% strain; (3) 2×10 ⁴ poking cycles at a distance of 30 mm
Power	Wireless power by the Bluetooth protocol from an external power source	Power free
Function	Instantaneous change in RSSI and image displayed on the smartphone application when the index finger is placed near the metamaterial textile	Wireless electronic textiles: Enable remote drone operation and support assisted communication

Revision in Manuscript:

Page 33, line 23 (“*Body-coupled MCP-V textiles remotely interact with a drone in Methods*”):

(b) Wireless signal **reception and analysis**: A coil antenna and a spectrum analyzer are used to capture and read the wireless signal within a specific frequency range from the **MCP-V** textile. The spectrum analyzer identifies the peak frequency of the wireless signal, which corresponds to the main frequency of the sensing signal. The process of acquiring and analyzing ambient radio frequency signals includes signal acquisition, band-pass filtering, Fourier transform (scanning every 15 ms), extension for signal amplification, and wireless signal conversion. **Meanwhile, the buttons in the MCP-V textile were embroidered with MCP-V fibers of varying lengths. As a result, the buttons can emit radio electromagnetic (EM) wave signals with different intensities. Each signal intensity is mapped to a specific control function button.**

45. Page 31, “*Fiber area measurements were conducted using SEM, with the average value calculated from at least three samples for each fiber*”, what area was used for each fibre?

Response:

We thank the reviewer for your comment. We employed SEM-FIB to obtain the cross-sections of the MCP fibers and used software (Image-Pro Plus) to analyze and determine the cross-sectional area of MXene composite fibers, as shown in **Tables R7-9, 13, and 14**. The tensile strength and electrical conductivity of fibers were calculated according to the S. We have updated Supplementary Tables 2, 14, and 16 in the revised Supplementary Information.

Table R7. The tensile strength, Young’s modulus, strain, and toughness of the fabricated MC fibers with different lengths of CNTs.

Samples	S (μm^2) Whole fiber	Tensile strength (MPa)	Young’s modulus (GPa)	Strain (%)	Toughness (MJ m⁻³)
MC-13.91	5749.7 \pm 112.5	50.4 \pm 1.0	14.9 \pm 0.3	0.50 \pm 0.01	0.14 \pm 0.01
MC-7.98	5377.3 \pm 99.5	93.2 \pm 2.0	18.5 \pm 0.1	0.56 \pm 0.01	0.29 \pm 0.01
MX	5754.9 \pm 73.9	109.3 \pm 2.2	21.5 \pm 0.1	0.58 \pm 0.01	0.36 \pm 0.02
MC-2.55	5181.8 \pm 88.5	182.5 \pm 1.2	24.1 \pm 0.2	1.87 \pm 0.04	1.96 \pm 0.10
MC-0.46	5116.8 \pm 68.8	284.3 \pm 4.7	32.8 \pm 0.3	2.56 \pm 0.03	4.42 \pm 0.20
MC-0.19	5017.0 \pm 96.8	225.0 \pm 0.4	26.2 \pm 0.2	2.26 \pm 0.03	2.96 \pm 0.05

Table R8. The tensile strength, Young's modulus, strain, and toughness of the fabricated MC fibers with different weight percentages of CNTs.

Samples	S (μm^2) Whole fiber	Tensile strength (MPa)	Young's modulus (GPa)	Strain (%)	Toughness (MJ m^{-3})
MX	5754.9 \pm 73.9	109.3 \pm 2.2	21.5 \pm 0.1	0.58 \pm 0.01	0.36 \pm 0.02
MC-1%	5116.8 \pm 68.8	284.3 \pm 4.7	32.8 \pm 0.3	2.56 \pm 0.03	4.42 \pm 0.20
MC-2%	4968.3 \pm 97.7	569.6 \pm 3.9	37.2 \pm 1.0	5.54 \pm 0.09	17.38 \pm 1.29
MC-3%	5139.2 \pm 104.6	631.8 \pm 4.5	42.3 \pm 1.1	6.14 \pm 0.17	23.19 \pm 1.19
MC-4%	5160.8 \pm 84.6	502.8 \pm 3.5	34.9 \pm 1.4	4.45 \pm 0.03	11.54 \pm 0.43
MC-5%	5530.6 \pm 96.8	346.4 \pm 6.6	29.1 \pm 0.7	3.44 \pm 0.07	6.73 \pm 0.42

Table R9. The tensile strength, Young's modulus, strain, and toughness of the fabricated MCP fibers with increasing draw-down ratios.

Samples	S (μm^2) Whole fiber	Tensile strength (MPa)	Young's modulus (GPa)	Strain (%)	Toughness (MJ m^{-3})
PLA	28165.5 \pm 43.5	93.6 \pm 0.3	1.0 \pm 0.1	342.92 \pm 2.11	256.6 \pm 7.5
MCP-55	28015.8 \pm 98.5	418.3 \pm 5.9	21.9 \pm 1.9	49.03 \pm 0.86	180.4 \pm 6.9
MCP-63	21787.3 \pm 104.6	549.2 \pm 13.2	31.4 \pm 3.0	39.82 \pm 0.61	164.8 \pm 2.1
MCP-67	16481.2 \pm 76.6	683.4 \pm 9.0	48.6 \pm 3.0	28.75 \pm 0.73	153.7 \pm 4.9
MCP-71	14654.2 \pm 48.7	941.5 \pm 5.9	61.43 \pm 2.1	22.50 \pm 0.07	147.9 \pm 4.2

Table R13. The tensile strength, strain, and toughness of the MC fibers fabricated through different spinning nozzles with diameters from 180 μm to 500 μm .

Samples	S (μm^2) Whole fiber	Ultimate tensile force (N)	Tensile strength (MPa)	Strain (%)	Toughness (MJ m^{-3})
MC-180	1152.6 \pm 64.3	1.36 \pm 0.01	1,056.0 \pm 5.8	8.60 \pm 0.12	45.9 \pm 2.7
MC-250	2284.7 \pm 86.5	1.98 \pm 0.02	822.0 \pm 6.5	7.54 \pm 0.29	32.7 \pm 1.0
MC-380	5139.2 \pm 104.6	4.87 \pm 0.03	631.8 \pm 4.5	6.14 \pm 0.17	23.2 \pm 1.2
MC-500	13780.7 \pm 96.7	10.44 \pm 0.08	507.0 \pm 4.1	5.00 \pm 0.03	11.7 \pm 0.4

Table R14. The electrical conductivity of the fabricated MCP fibers with increasing draw-down ratios.

Samples	S (μm^2) Inner MC fiber	Conductivity (Inner MXene fiber) (S cm^{-1})
MC-3%	5139.2 \pm 104.6	11,856.7 \pm 39.55
MCP-55	5096.2 \pm 69.9	9,245.8 \pm 54.85
MCP-63	4976.1 \pm 86.5	10,986.8 \pm 67.45
MCP-67	4607.4 \pm 124.6	11,498.5 \pm 125.7
MCP-71	4581.2 \pm 62.5	12,836.4 \pm 108.69

46. Page 34, the description provided for “Body-coupled FMT-C textiles remotely interact with a drone” is insufficient. For instance, “When the finger touches the FMT-C textile, it enables interaction with the drone to control its flight”, how was this interaction facilitated? How could the drone know which button was pressed? Where is the power unit?

Response:

We thank the reviewer for your comment. Body-coupled MCP-V textiles remotely interact with the drone as follows.

(a) When the voltage at the skin-fiber interface capacitance exceeds the air breakdown threshold, high-frequency LC oscillation is triggered, emitting radio electromagnetic (EM) wave signals **without power unit (Fig. R23)**. The touch-induced electrical signal represents a novel form of wireless communication. The wireless signals generated by the MCP-V textile inherently carry sensing information, eliminating the need for additional encoding steps.

(b) Wireless signal reception and analysis: A coil antenna and a spectrum analyzer are used to capture and read the wireless signal within a specific frequency range from the MCP-V textile. The spectrum analyzer identifies the peak frequency of the wireless signal, which corresponds to the main frequency of the sensing signal. The process of acquiring and analyzing ambient radio frequency signals includes signal acquisition, band-pass filtering, Fourier transform (scanning every 15 ms), extension for signal amplification, and wireless signal conversion (Fig. R22a, g). **Meanwhile, the buttons in the MCP-V textile were embroidered with MCP-V fibers of varying lengths. As a result, the buttons can emit radio electromagnetic (EM) wave signals with different intensities. Each signal intensity is mapped to a specific control function button.**

(c) Signal acquisition for remote interaction with a drone: A modified drone (QQLRC Model M60 Pro+) equipped with a receiver and microprocessor is used to collect wireless signal data at 15 ms intervals. When the finger touches the MCP-V textile, it enables interaction with the drone to control its flight.

Fig. R23. a, Illustration of the working principle of body-coupled MCP-V fiber. Where E_{MC} represents the induced electric field within the MC fiber antenna, and E_s denotes the EM field acting on the MCP-V fiber's surface. **b**, The photograph shows the MCP-V fiber being wirelessly powered by the finger, while also capturing the surrounding electromagnetic energy. **c**, Directional diagrams illustrating wireless optical luminance. **d**, Directional diagrams depicting wireless electrical signals. Data are presented based on three repeated measurements (mean \pm SD).

Fig. R22. Applications of the MCP-V textile. **a**, Schematic diagram illustrating the mechanism for wireless optical and electrical signal generation and transmission using a single MCP-V fiber with various colors under body-coupled EM fields. Where C_{cs} represents the contact interface capacitance during local discharge at the surface, C_{ni} is the radial interface capacitance, R_{MC} is the resistance, and L_{MC} is the axial fiber inductance provided by the MC fiber. Additionally, C_p is the parasitic capacitance with the ground formed by the human body and the MCP-V fiber. **b**, The dyed MCP-V fibers emit a red glow when exposed to body-coupled EM fields for the assisted communication. **c**, Embroider patterns on textiles using MCP-V fiber. **d**, Durability of the textile incorporating MCP-V fiber under bending, stretching, and poking conditions. **e**, A conceptual illustration shows how MCP-V textiles can be used to control a drone remotely. **f**, Photograph showcasing MCP-V textiles being used to control a drone remotely. **g**, Block diagram illustrating the interaction between MCP-V textiles and a drone, depicting signal input from the textile interface, processing of commands, wireless transmission to the drone, and execution of drone actions in response to the received signals. **h**, The flight path of the drone controlled by the MCP-V textile, accompanied by real-life images.

Revision in Manuscript:

Page 33, line 23 (“Body-coupled MCP-V textiles remotely interact with a drone in Methods”):

(b) Wireless signal **reception and analysis**: A coil antenna and a spectrum analyzer are used to capture and read the wireless signal within a specific frequency range from the **MCP-V** textile. The spectrum analyzer identifies the peak frequency of the wireless signal, which corresponds to the main frequency of the sensing signal. The process of acquiring and analyzing ambient radio frequency signals includes signal acquisition, band-pass filtering, Fourier transform (scanning every 15 ms), extension for signal

amplification, and wireless signal conversion. Meanwhile, the buttons in the MCP-V textile were embroidered with MCP-V fibers of varying lengths. As a result, the buttons can emit radio electromagnetic (EM) wave signals with different intensities. Each signal intensity is mapped to a specific control function button.

47. Supplementary Information Page 4, what is the centrifuge model used and what was the relative centrifugal force in each case?

Response:

We thank the reviewer for your comment. We prepared MXene nanosheet dispersion by using the high-speed refrigerated centrifuge model HC-3016R from Anhui USTC Zonkia Scientific Instruments Co., Ltd. A speed of 3500 rpm corresponds to a relative centrifugal force of 1329×g, while 1500 rpm corresponds to 244×g. We have provided the details in the revised Supplementary Information.

Revision in Supplementary Information:

Page 3, line 8 and 13 (“Fabrication of MXene ($Ti_3C_2T_x$) nanosheet dispersion in Supplementary Methods”):

The MXene nanosheet solutions were synthesized using the following procedure: Initially, 2.7 g of Ti_3AlC_2 powders were combined with a solution consisting of 5.7 g of LiF dissolved in 60 mL of 9 M HCl at room temperature. Subsequently, the mixture was stirred at 50 °C for 30 hours to ensure a complete reaction. The resulting accordion-like MXene product underwent thorough washing: three cycles with 9 M HCl solution and then approximately ten cycles with deionized water, each cycle involving 5 minutes of centrifugation at 3500 rpm (a relative centrifugal force of 1329×g) using the high-speed refrigerated centrifuge model HC-3016R from Anhui USTC Zonkia Scientific Instruments Co., Ltd. The sediments obtained were dispersed into 150 mL of deionized water with continuous vibration for 15 minutes until the supernatant solution reached a pH of ~7. The solution was then centrifuged at 1500 rpm (a relative centrifugal force of 244×g) for 30 minutes to obtain the supernatant solution, followed by centrifugation at 3500 rpm for another 30 minutes to obtain sediments. Finally, these sediments were dispersed in deionized water to prepare the MXene nanosheet solutions with different concentrations.

48. Supplementary Information Page 4, “0.01 g of a dispersing agent was introduced”, What was this dispersing agent?

Response:

We thank the reviewer for your comment. The dispersant for carbon nanotubes aqueous solutions (XFZ20) was a surfactant, purchased from Jiangsu Xianfeng Nanomaterials Technology Co., Ltd. We have provided the details in the revised Supplementary Information.

Revision in Supplementary Information:

Page 3, line 18 (“Preparation of carboxylated multiwalled carbon nanotube solution in Supplementary Methods”):

0.01 g of a dispersing agent (XFZ20, purchased from Jiangsu Xianfeng Nanomaterials Technology Co., Ltd) was introduced into 50 mL of deionized water and stirred continuously for 1 hour. Following this, 1 g of carboxylated multiwalled carbon nanotubes (CNTs)...

49. Supplementary Information Page 4, provide details of the model, power used, pulse details for the ultrasonic cell.

Response:

We thank the reviewer for your comment. The carboxylated multiwalled carbon nanotube solution was prepared using an ultrasonic cell disruptor (model SCIENTZ-IID, SCIENTZ Co., Ltd.) operating at 700 W with a pulse mode of 2 seconds on and 2 seconds off. We have provided the details in the revised Supplementary Information.

Revision in Supplementary Information:

Page 3, line 24 (“Preparation of carboxylated multiwalled carbon nanotube solution in Supplementary Methods”):

resulting solution containing the CNTs underwent ultrasonic dispersion using an ultrasonic cell disruptor (model SCIENTZ-IID, SCIENTZ Co., Ltd.) operating at 700 W with a pulse mode of 2 seconds on and 2 seconds off for 2 hours in an ice bath. Upon centrifugation of the CNTs dispersion solution at 3500 rpm for 30 minutes, a supernatant solution containing dispersed nanotubes was obtained.

50. Supplementary Information Page 4, “ratios of MXene nanosheets to short CNTs from 1 wt% to 5 wt%”, this is confusing, these are not ratios. How was wt% calculated?

Response:

We thank the reviewer for your comment. In order to make more clear, we have revised the sentence “Additionally, different ratios of MXene nanosheets to short CNTs from 1 wt% to 5 wt% were adjusted to prepare various MC fibers, denoted as MC-1%, MC-2%, MC-3%, MC-4%, and MC-5%” to be “Additionally, a series of MC fibers containing both MXene nanosheets and short CNTs were fabricated, with the CNTs accounting for 1 wt%, 2 wt%, 3 wt%, 4 wt%, and 5 wt% of the total fiber weight. These fibers were designated as MC-1%, MC-2%, MC-3%, MC-4%, and MC-5%, respectively”.

The component content of MXene and CNTs in MC fibers was calculated according to the TGA curves (**Fig. R14**). M_{MXene} , M_{CNTs} , and M_{MC} are the fractions of weight loss for MXene, CNTs, and MC fibers fabricated with different weight percentages of CNTs. The weight content (W_{CNTs}) of CNTs in the MC fiber was calculated using equation (R2). The results are shown in **Table R6**, indicating that MC-3% contains an actual 3.2 wt% of CNTs. We have added Supplementary Fig. 21 and Supplementary Table 4 in the revised Supplementary Information.

$$W_{CNTs} = \frac{M_{MC} - M_{MXene}}{M_{CNTs} - M_{MXene}} \quad (R2)$$

Fig. R14. TGA curves of pure MXene, CNTs, MC-1%, MC-2%, MC-3%, MC-4%, and MC-5%. These results were obtained in a nitrogen atmosphere using a heating rate of 10 K min⁻¹.

Table R6. The weight percentages of CNTs and MXene in MC fibers calculated according to the TGA curves.

Samples	MXene (wt%)	CNTs (wt%) added in the experiment	CNTs (wt%) calculated according to the TGA
MX	100	0	0
MC-1%	98.9	1	1.1
MC-2%	97.7	2	2.3
MC-3%	96.8	3	3.2
MC-4%	95.6	4	4.4
MC-5%	94.3	5	5.6

51. Why was the coagulation bath changed from absolute ethyl alcohol to ammonium chloride solution?

Response:

We thank the reviewer for your comment. During the wet assembly, for example, wet spinning, the coagulation conditions can influence fiber morphology and structure. For example, ethyl alcohol coagulation can facilitate the fast coagulation, while the chitosan aqueous solution exhibits the slow coagulation (*ACS Cent. Sci.* 2020, 6, 254). However, two-dimensional (2D) nanosheets assembled via wet chemical methods inevitably undergo capillary contraction during the drying process (*Nat. Commun.* 2020, 11, 2645; *Science* 2024, 383, 771; *Carbon* 2014, 66, 84). This capillary-induced contraction leads to significant structural shrinkage, which in turn results in the formation of transverse wrinkles in the nanosheets when MXene fibers are fabricated in combination with a pre-stretching step during wet spinning (*ACS Cent. Sci.* 2020, 6, 254; *ACS Nano* 2015, 15, 3320). Therefore, lots of voids were generated due to transverse wrinkles between MXene nanosheets due to the capillary contraction during

wet spinning regardless which coagulant was used, such as ethyl alcohol or ammonium chloride aqueous solution.

Moreover, we have fabricated the pure MXene (PM has been revised as MX) fiber in coagulant bath containing absolute ethyl alcohol. Although continuous extrusion was possible using coagulant bath containing absolute ethyl alcohol, the resulting fibers were too weak to handle after spinning with more voids and hard to obtain the mechanical properties, as in the reported work (*ACS Cent. Sci.* 2020, 6, 254). Meanwhile, a larger number of transverse wrinkles were also observed in the MX fibers with lots of voids, as shown in **Fig. R9**. Based on the experimental result, the inherent transverse wrinkles in MXene fibers were generated when pure MXene fibers were fabricated using coagulant bath, either ammonium chloride solution or absolute ethanol. In contrast, we can easily fabricate MXene fibers (FM has been revised as MC) through one-step static filling method through hydrogen bonds using short CNTs in a coagulant bath containing absolute ethyl alcohol. The obtained MC fibers exhibited high tensile strength. Therefore, we used different coagulation baths to fabricate MX and MC fibers.

Fig. R9. SEM image of MX fibers fabricated in coagulant bath containing ammonium chloride aqueous solution bath (a) and absolute ethyl alcohol bath (b). TEM image of MX fibers fabricated in coagulant bath containing ammonium chloride aqueous solution bath (c) and absolute ethyl alcohol bath (d).

52. Supplementary Figure 6b, it appears the MXene/CNT dispersion looks like a paste. This is unusual as MXene at 40 mg/mL tends to flow relatively easily. The authors must report the composition of the sample shown appropriately.

Response:

We thank the reviewer for your comment. We confirm that our MXene/CNTs dispersion concentration is 40 mg/mL (**Fig. R24b**). We have updated Supplementary Fig. 9 in the revised Supplementary Information.

Fig. R24 **a**, Photo of MXene and CNTs dispersion. **b**, Photo of MXene-CNTs dispersion with the concentration of 40 mg mL⁻¹ for wet spinning. **c**, POM image of MXene-CNTs spinning dispersion with the concentrations of 40 mg mL⁻¹, exhibiting optical birefringence.

53. Supplementary Figure 7, present viscosity vs. shear rate for pure MXene and MXene/CNT systems to allow for comparison. Also, why is G'/G'' decreasing with concentration? This is counter-intuitive and opposite the trends observed previously for MXene (DOI: 10.1021/acsnano.7b08889).

Response:

We thank the reviewer for your comment. We have provided the viscosity vs. shear rate for pure MXene and MXene/CNTs systems (**Fig. R25**) to allow for comparison in Supplementary Fig. 7. For the spinning solution, the viscous modulus (G'') reflects the ease of flowability, while the elastic modulus (G') indicates the system's ability to maintain structural integrity during application. In our work, as the concentration increases, both moduli exhibit a pronounced increase with rising concentration, while G' dominates across the entire frequency range, suitable for the wet spinning of MXene/CNTs spinning solution, while maintaining their shape. However, the increasing degree of G' is smaller than that of G'' because of the high concentration. Therefore, the G'/G'' decreased as concentration increased in our work. In the reference (DOI: 10.1021/acsnano.7b08889, ACS Nano 2018, 12, 2685), the concentration range is 0.18-3.60 mg mL⁻¹, and no clear correlation between the G'/G'' ratio and concentration was observed over the full frequency spectrum. In contrast, our spinning solution operates at a much higher concentration range of 5-40 mg mL⁻¹, so it might not be appropriate to make a direct comparison between these two works.

Fig. R25 Viscosity as a function of shear rate (s⁻¹) with different concentrations of MXene-CNTs spinning dispersion from 5 mg mL⁻¹ to 40 mg mL⁻¹, and pure MXene spinning dispersion with the concentration of 40 mg mL⁻¹.

54. *Supplementary Figure 51, for the EDS map of Ba, why is there a strong Ba signal in the core given the core must be FM fiber?*

Response:

We thank the reviewer for your comment. After the MCP-V fiber is encapsulated and cured with resin, we cut through the entire cross-section with a blade, and a small amount of BaTiO₃ particles (~100 nm) from the PLA-BaTiO₃ layer may be transferred onto the inner MC fiber. This results in a minor Ba signal, which is unavoidable during the cutting process.

55. *This manuscript would also benefit from a close editing. Some examples of language issues that are frequently seen throughout this paper are: “greater porosity with more voids” [page 5], “~3453.2 cm⁻¹” [page 8], “Simultaneously, in FMT fiber, the binding energy of Ti-O shifted up to 459.4 eV and 464.1 eV compared to that in FM fiber” [page 8], “scale-up FM fiber” [page 8], “increase the spinning nozzle” [page 11], “testified by in-situ XRD patterns heated from” [page 12], “This is followed by the further sliding between” [page 15], “FMT textile performed the conductivity” [page 17], “which the design contained a ground layer made of FMT” [page 19], “APP interfaces” [page 21], “due to the thermal generated” [page 21], “The hundreds of meters” [page 22], “rate of 0.3 mm min⁻¹” [page 31], “receiving & analysis” [34].*

Response:

We thank the reviewer for your comment.

The sentence “the presence of the transverse wrinkles because of weak interfacial interactions led to greater porosity with more voids” has been revised to be “the presence of the transverse wrinkles because of weak interfacial interactions led to more voids” (page 5, line 21).

The “~3453.2 cm⁻¹” has been revised to be “~3453.2 cm⁻¹” (page 8, line 13).

The “Simultaneously, in MCP fiber, the binding energy of Ti-O shifted up to 459.4 eV and 464.1 eV compared to that in MC fiber” has been revised to be “Simultaneously, the binding energy of Ti-O in the MCP fiber shifted up to 459.4 eV and 464.1 eV compared to those in MC fiber” (page 8, line 20).

The “scale-up MC fiber” has been revised to be “scaled-up MC fiber” (page 8, line 24).

The “increase the spinning nozzle” has been revised to be “increase the spinning nozzle diameter” (page 11, line 32).

The “testified by in-situ XRD patterns heated from” has been revised to be “testified by in-situ XRD patterns obtained during fiber heating from” (page 12, line 12).

The “This is followed by the further sliding between MXene nanosheets and among CNTs” has been revised to be “Subsequently, further sliding takes place between MXene nanosheets as well as among CNTs” (page 15, line 29).

The “MCP textile performed the conductivity” has been revised to be “the MCP fiber in the textile performed the conductivity” (page 18, line 5).

We have deleted the sentence “which the design contained a ground layer made of MCP” according to Reviewer 1’s comment.

The “Simultaneously, the smartphone APP interfaces recorded data through wireless communication” has been revised to be “Simultaneously, the smartphone with the customized application recorded data through wireless communication” (page 21, line 3).

The “due to the thermal generated” has been revised to be “due to the generated body heat” (page 21, line 12).

The “The hundreds of meters” has been revised to be “Hundreds of meters” (page 21, line 29).

The “rate of 0.3 mm min⁻¹” has been revised to be “ rate of 0.3 mm min⁻¹” (page 29, line 25).

The “receiving & analysis” has been revised to be “reception and analysis” (page 33, line 23).

We have also carefully polished the language in the revised Manuscript.

Point-by-Point Responses to Reviewers' Comments

We thank all the reviewers for their in-depth review of our manuscript and for enriching us with their valuable comments and suggestions, which have helped us further improve the quality of the manuscript. The reviewers' comments are in *blue font*, and the authors' responses are in **black font**. All the changes in the revised Manuscript and Supplementary Information are marked in **red font**.

Reviewer #1:

The authors have addressed all the related issues. Publication of this manuscript is recommended.

Response:

We really appreciate your in-depth review of our manuscript.

Reviewer #4:

The manuscript demonstrates a strong composite-fiber strategy (static filling + thermal drawing with PLA) and convincing battery-free NFC textile demos with clear scalability and durability. The authors have addressed most of Reviewer 3's questions satisfactorily. However, one consistency issue remains: you compute tensile strength from the maximum load using the total composite cross-section (including PLA), whereas electrical conductivity is reported on a core-only basis. This mismatch may unintentionally overestimate performance. Given that the system is a composite fiber and the PLA encapsulation materially affects performance and demonstrations, the authors should report electrical conductivity on a composite-fiber basis—i.e., σ_{comp} using the total cross-section (core + PLA)—and avoid mixing core-only conductivity with composite-area mechanics in headline claims. This will prevent overstatement and remove ambiguity in MX/MC \rightarrow MCP comparisons.

Response:

We thank the reviewer for your comments. According to your comments and suggestions, we have recalculated electrical conductivity on a composite-fiber basis by using the total cross-section including both the inner MXene core and the encapsulating PLA, as shown in **Table R1**. When calculating the electrical conductivity, the total cross-section including the inner MXene core and the encapsulating PLA is used. We have already provided both values and updated Figure 1e, Figure 3b, Supplementary Figure 39c, and Supplementary Tables 17-19, 23 accordingly in the revised Manuscript and Supplementary Information.

Table R1. The cross-sectional area of the inner MC fibers and the electrical conductivity of the fabricated MCP fibers with increasing draw-down ratios.

Samples	S (μm^2) Inner MC fiber	Conductivity (Inner MXene fiber, MCP _{core}) (S cm ⁻¹)	Conductivity (Whole composite fiber) (S cm ⁻¹)
MCP-55	5,096.2 \pm 69.9	9,245.8 \pm 54.9	1,681.9 \pm 17.9
MCP-63	4,976.1 \pm 86.5	10,986.8 \pm 67.5	2,509.3 \pm 23.9
MCP-67	4,607.4 \pm 124.6	11,498.5 \pm 125.7	3,214.5 \pm 21.4
MCP-71	4,451.2 \pm 62.5	12,836.4 \pm 108.7	3,899.0 \pm 14.8